# Bifurcations and loss jumps in RNN training

**Lukas Eisenmann[1,2,*], Zahra Monfared[1,*], Niclas Göring[1,2], and Daniel Durstewitz[1,2,3]**

{lukas.eisenmann,zahra.monfared,daniel.durstewitz}@zi-mannheim.de
[1]Department of Theoretical Neuroscience, Central Institute of Mental Health,
Medical Faculty Mannheim, Heidelberg University, Mannheim, Germany
[2]Faculty of Physics and Astronomy, Heidelberg University, Heidelberg, Germany
[3]Interdisciplinary Center for Scientific Computing, Heidelberg University
[*]These authors contributed equally

## Abstract

Recurrent neural networks (RNNs) are popular machine learning tools for modeling and forecasting sequential data and for inferring dynamical systems (DS) from observed time series. Concepts from DS theory (DST) have variously been used to further our understanding of both, how trained RNNs solve complex tasks, and the training process itself. Bifurcations are particularly important phenomena in DS, including RNNs, that refer to topological (qualitative) changes in a system's dynamical behavior as one or more of its parameters are varied. Knowing the bifurcation structure of an RNN will thus allow to deduce many of its computational and dynamical properties, like its sensitivity to parameter variations or its behavior during training. In particular, bifurcations may account for sudden loss jumps observed in RNN training that could severely impede the training process. Here we first mathematically prove for a particular class of ReLU-based RNNs that certain bifurcations are indeed associated with loss gradients tending toward infinity or zero. We then introduce a novel heuristic algorithm for detecting all fixed points and $k$-cycles in ReLU-based RNNs and their existence and stability regions, hence bifurcation manifolds in parameter space. In contrast to previous numerical algorithms for finding fixed points and common continuation methods, our algorithm provides *exact* results and returns fixed points and cycles up to high orders with surprisingly good scaling behavior. We exemplify the algorithm on the analysis of the training process of RNNs, and find that the recently introduced technique of generalized teacher forcing completely avoids certain types of bifurcations in training. Thus, besides facilitating the DST analysis of trained RNNs, our algorithm provides a powerful instrument for analyzing the training process itself.

## 1 Introduction

Recurrent neural networks (RNNs) are common and powerful tools for learning sequential tasks or modeling and forecasting time series data [31, 65, 2, 43, 9, 26]. Typically, RNNs are "black boxes," whose inner workings are hard to dissect. Techniques from dynamical system theory (DST) can significantly aid in this effort, as RNNs are formally discrete-time dynamical systems (DS) [37]. A better understanding of how RNNs solve their tasks is important for detecting failure modes and designing better architectures and training algorithms. In scientific machine learning, on the other hand, RNNs are often employed for reconstructing unknown DS from sets of time series observations [10, 30, 64, 63, 23, 55], e.g. in climate or disease modeling. In this case, the RNN is supposed to provide an approximation to the flow of the observed system that reproduces all its dynamical properties, e.g., cyclic behavior in climate or epidemiological systems. In such scientific or medical

settings, a detailed understanding of the RNN's dynamics and its sensitivity to parameter variations is in fact often crucial.

Beyond understanding the dynamical properties of a once trained RNN, it may also be of interest to know how its dynamical repertoire changes with changes in its parameters. The parameter space of any DS is partitioned into regions (or sets) with topologically different dynamical behaviors by bifurcation curves (or, more generally, manifolds). Such bifurcations, qualitative changes in the system dynamics due to parameter variations, may not only be crucial in scientific applications where we use RNNs, for instance, to predict tipping points in climate systems or medical scenarios like sepsis detection [45]. They also pose severe challenges for the training process itself as qualitative changes in RNN dynamics may go in hand with sudden jumps in the loss landscape [16, 44]. Although methods from DST have significantly advanced the field in recent years, especially with regards to algorithms for reconstructing nonlinear DS from data [55, 46, 67], progress is still hampered by the lack of efficient tools for analysing the dynamics of higher-dimensional RNNs and their bifurcations. In particular, methods are needed for exactly locating geometrical objects like fixed points or cycles in an RNN's state space, but current numerical techniques do not scale well to higher-dimensional scenarios and provide only approximate solutions [29, 21, 61].

The contributions of this work are threefold: After first providing an introduction into bifurcations of piecewise-linear (ReLU-based) RNNs (PLRNNs), which have been extensively used for reconstructing DS from empirical data [10, 30], we mathematically prove that certain bifurcations during the training process will indeed cause loss gradients to diverge to infinity, resulting in abrupt jumps in the loss, while others will cause them to vanish. As RNNs are likely to undergo several bifurcations during training on their way from some initial parameter configuration toward a dynamics that successfully implements any given task, this poses severe challenges for RNN training and may be one of the reasons for exploding and vanishing gradients [16, 44, 8, 25]. We then create a novel, efficient heuristic algorithm for *exactly* locating all fixed points and $k$-cycles in PLRNNs, which can be used to delineate bifurcation manifolds in higher-dimensional systems. Our algorithm finds these dynamical objects in many orders of magnitude less time than an exhaustive search would take. Using this algorithm, we demonstrate empirically that steep cliffs in loss landscapes and bifurcation curves indeed tightly overlap, and that bifurcations in the system dynamics are accompanied by sudden loss jumps. Finally, we prove and demonstrate that the recently introduced technique of generalized teacher forcing (GTF) [24] completely eliminates certain types of bifurcation in training, providing an explanation for its efficiency.

## 2 Related Work

**DS analysis of trained RNNs**   In many areas of science one is interested in identifying the nonlinear DS that explains a set of observed time series [12, 23]. A variety of purely data-driven machine learning approaches have been developed for this purpose [11, 51, 15], but mostly RNNs are used for the goal of reconstructing DS from measured time series [10, 55, 30, 64, 63, 46, 45]. In scientific settings in particular, but also often in engineering applications, we seek a detailed understanding of the system dynamics captured – or the dynamical repertoire produced – by a trained RNN [62, 33, 61, 14, 13]. To analyze an RNN's dynamics, typically its fixed points are determined numerically, for instance by optimizing some cost function [21, 34, 61], by numerically traversing directional fibers [29], or by co-training a switching linear dynamical system [56]. These techniques, however, often scale poorly with dimensionality, provide only approximate solutions, and are not designed for detecting other dynamical objects like $k$-cycles. This is, however, crucial for understanding the complete dynamical repertoire of a trained RNN. PLRNNs are of particular interest in this context [10, 55, 30, 38], because their piecewise linear nature makes some of their properties analytically accessible [55, 10], which is a tremendous advantage from the perspective of DST. In this work, we develop an efficient heuristic algorithm for locating a PLRNN's fixed points exactly, as well as its $k$-cycles.

**Bifurcations and loss jumps in RNN training**   The idea that bifurcations in RNN dynamics could impede the training process is not new [16, 44]. Doya [16], to our knowledge, was the first to point out that even in simple single-unit RNNs with sigmoid activation function (saddle-node) bifurcations may occur as an RNN parameter is adapted during training. This may not only cause an abrupt jump in training loss, but could lead to situations where it is impossible, even in principle, to reach the training objective (the desired target output), as across the bifurcation point there is a *discrete* change

in network behavior [16, 68, 4]. Pascanu et al. [44] discussed similar associations between steep cliffs in RNN loss functions and bifurcations. Although profound for training success, this topic received surprisingly little attention over the years. Haschke and Steil [22] extended previous work by a more formal treatment of bifurcation boundaries in RNNs, and Marichal et al. [35] examined fold bifurcations in RNNs. The effects of bifurcations and their relation to exploding gradients in gated recurrent units (GRUs) was investigated in Kanai et al. [27]. Ribeiro et al. [50] looked at the connection between dynamics and smoothness of the cost function, but failed to find a link between bifurcations and jumps in performance. In contrast, Rehmer and Kroll [49] observed large gradients at bifurcation boundaries and concluded that bifurcations can indeed cause problems in gradient-based optimization. To the best of our knowledge, we are, however, the first to *formally prove* a direct link between bifurcations and the behavior of loss gradients, and to derive a systematic and efficient algorithmic procedure for identifying bifurcation manifolds for a class of ReLU-based RNNs.

## 3 Theoretical analysis

In this paper we will focus on PLRNNs as one representative of the wider class of ReLU-based RNNs, but similar derivations and algorithmic procedures could be devised for any type of ReLU-based RNN (in fact, many other types of ReLU-based RNNs could be brought into the same functional form as PLRNNs; e.g. [10]). We will first, in sect. 3.1, provide some theoretical background on bifurcations in PLRNNs, and illustrate how existence and stability regions of fixed points and cycles could be analytically computed for low dimensional ($2d$) PLRNNs. In sect. 3.2 we will then state two theorems regarding the association of bifurcations and loss gradients in training. It turns out that for certain types of bifurcations exploding or vanishing gradients are inevitable in gradient-based training procedures like Back-Propagation Through Time (BPTT).

### 3.1 Bifurcation curves in PLRNN parameter space

The PLRNN, originally introduced as a kind of discrete time neural population model [18], has the general form

$$z_t = F_\theta(z_{t-1}, s_t) = A\,z_{t-1} + W\phi(z_{t-1}) + C s_t + h, \tag{1}$$

where $z_t \in \mathbb{R}^M$ is the latent state vector and $\theta$ are system parameters consisting of diagonal matrix $A \in \mathbb{R}^{M \times M}$ (auto-regression weights), off-diagonal matrix $W \in \mathbb{R}^{M \times M}$ (coupling weights), $\phi(z_{t-1}) = \max(z_{t-1}, 0)$ is the element-wise rectified linear unit ($ReLU$) function, $h \in \mathbb{R}^M$ a constant bias term, and $s_t \in \mathbb{R}^K$ represents external inputs weighted by $C \in \mathbb{R}^{M \times K}$. The original formulation of the PLRNN is stochastic [18, 30], with a Gaussian noise term added to eq. (1), but here we will consider the deterministic variant.

Formally, like other ReLU-based RNNs, PLRNNs constitute piecewise linear (PWL) maps, a subclass of piecewise smooth (PWS) discrete-time DS. Define $D_{\Omega(t)} := \mathrm{diag}(d_{\Omega(t)})$ as a diagonal matrix with indicator vector $d_{\Omega(t)} := (d_1, d_2, \cdots, d_M)$ such that $d_m(z_{m,t}) =: d_m = 1$ whenever $z_{m,t} > 0$, and zero otherwise. Then (1) can be rewritten as

$$z_t = F_\theta(z_{t-1}) = (A + W D_{\Omega(t-1)}) z_{t-1} + h =: W_{\Omega(t-1)}\, z_{t-1} + h, \tag{2}$$

where we have ignored external inputs $s_t$ for simplicity. There are in general $2^M$ different configurations for matrix $D_{\Omega(t-1)}$ and hence for matrix $W_{\Omega(t-1)}$, dividing the phase space into $2^M$ sub-regions separated by switching manifolds (see Appx. A.1.1 for more details).

Recall that fixed points of a map $z_t = F_\theta(z_{t-1})$ are defined as the set of points for which we have $z^* = F_\theta(z^*)$, and that the type (node, saddle, spiral) and stability of a fixed point can be read off from the eigenvalues of the Jacobian $J_t := \frac{\partial F_\theta(z_{t-1})}{\partial z_{t-1}} = \frac{\partial z_t}{\partial z_{t-1}}$ evaluated at $z^*$ [3, 47]. Similarly, a $k$-cycle of map $F_\theta$ is a periodic orbit $\{z_1^*, z_2^*, \ldots, z_k^*\}$ such that each of the periodic points $z_i^*, i = 1 \ldots k$, is distinct, and is a solution to the equation $z_i^* = F_\theta^k(z_i^*)$, i.e. the $k$ times iterated map $F_\theta$. Type and stability of a $k$-cycle are then determined via the Jacobian $\prod_{r=1}^k J_{t+k-r} = \prod_{r=1}^k \frac{\partial z_{t+k-r}}{\partial z_{t+k-r-1}} = \frac{\partial z_{t+k-1}}{\partial z_{t-1}}$. Solving these equations and computing the corresponding Jacobians thus allows to determine all existence and stability regions of fixed points and cycles, where the latter are a subset of the former, bounded by bifurcation curves (see Appx. A.1 for more formal details).

To provide a specific example, assume $M = 2$ and fix – for the purpose of this exposition – parameters $w_{12} = w_{22} = 0$, such that we have

$$\boldsymbol{W}_{\Omega^1} = \boldsymbol{W}_{\Omega^3} = \begin{pmatrix} a_{11} & 0 \\ 0 & a_{22} \end{pmatrix}, \qquad \boldsymbol{W}_{\Omega^2} = \boldsymbol{W}_{\Omega^4} = \begin{pmatrix} a_{11} + w_{11} & 0 \\ w_{21} & a_{22} \end{pmatrix}, \qquad (3)$$

i.e. only one border which divides the phase space into two distinct sub-regions (see Appx. A.1.1). For this setup, Fig. 1A provides examples of analytically determined stability regions for two low order cycles in the $(a_{11}, a_{11} + w_{11})$-parameter plane (see Appx. A.1). Note that there are regions in parameter space where two or more stability regions overlap: In these regions we have *multi-stability*, the co-existence of different attractor states in the PLRNN's state space.

As noted above, bifurcation curves delimit the different stability regions in parameter space and are hence associated with abrupt changes in the topological structure of a system's state space. In general, there are many different types of bifurcations through which dynamical objects can come into existence, disappear, or change stability (see, e.g., [3, 40, 47]), the most common ones being saddle node, transcritical, pitchfork, homoclinic, and Hopf bifurcations. In comparison with smooth systems, bifurcation theory of PWS (or PWL) maps includes additional dynamical phenomena related to the existence of borders in the phase space [6]. *Border-collision bifurcations (BCBs)* arise when for a PWS map a specific point of an invariant set collides with a border and this collision leads to a qualitative change of dynamics [5, 6, 42]. More specifically, a BCB occurs, if for a PWS map $\boldsymbol{z}_t = F_{\boldsymbol{\theta}}(\boldsymbol{z}_{t-1})$ a fixed point or $k$-cycle either *crosses* the switching manifold $\sum_i := \{\boldsymbol{z} \in \mathbb{R}^n : \boldsymbol{e}_i^\mathsf{T} \boldsymbol{z} = 0\}$ transversely at $\theta = \theta^*$ and its qualitative behavior changes in the event, or if it *collides* on the border with another fixed point or $k$-cycle and both objects disappear [7, 41]. *Degenerate transcritical bifurcations (DTBs)* occur when a fixed point or a periodic point of a cycle tends to infinity and one of its eigenvalues tends to 1 by variation of a parameter. Specifically, let $\Gamma_k, k \geq 1$, be a fixed point or a $k$-cycle with the periodic points $\{\boldsymbol{z}_1^*, \boldsymbol{z}_2^*, \ldots, \boldsymbol{z}_k^*\}$, and assume $\lambda^i$ denotes an eigenvalue of the Jacobian matrix at the periodic point $\boldsymbol{z}_i^*, i \in \{1, 2, \cdots, k\}$. Then $\Gamma_k$ undergoes a DTB at $\theta = \theta^*$, if $\lambda^i(\theta^*) \to +1$ and $\|\boldsymbol{z}_i^*\| \to \infty$. $\Gamma_k$ undergoes a *degenerate flip bifurcation (DFB)*, iff $\lambda^i(\theta^*) = -1$ and the map $F^k$ has locally, in some neighborhood of $\boldsymbol{z}_i^*$, infinitely many 2-cycles at $\theta = \theta^*$. A *center bifurcation (CB)* occurs, if $\Gamma_k$ has a pair of complex conjugate eigenvalues $\lambda_{1,2}$ and locally becomes a center at the bifurcation value $\theta = \theta^*$, i.e. if its eigenvalues are complex and lie on the unit circle ($|\lambda_{1,2}(\theta^*)| = 1$). Another important class of bifurcations are *multiple attractor bifurcations (MABs)*, discussed in more detail in Appx. A.1.5 (see Fig. S3). In addition to existence and stability regions of fixed points and cycles, in Appx. A.1.2-A.1.4 we also illustrate how to analytically determine the types of bifurcation curves bounding the existence and stability regions of $k$-cycles (DTB, DFB, CB and BCB curves; see also Fig. S6 for a $1d$ example).

## 3.2 Bifurcations and loss jumps in training

Here we will prove that major types of bifurcations discussed above are always associated with exploding or vanishing gradients in PLRNNs during training, and hence often with abrupt jumps in the loss. For this we may assume any generic loss function $\mathcal{L}(\boldsymbol{\theta})$, like a negative log-likelihood or a mean-squared-error (MSE) loss, and a gradient-based training technique like BPTT [52] or Real-Time-Recurrent-Learning (RTRL) that involves a recursion (via chain rule) through loss terms across time. The first theorem establishes that a DTB inevitably causes exploding gradients.

**Theorem 1.** *Consider a PLRNN of the form (2) with parameters $\boldsymbol{\theta} = \{\boldsymbol{A}, \boldsymbol{W}, \boldsymbol{h}\}$. Assume that it has a stable fixed point or $k$-cycle $\Gamma_k$ ($k \geq 1$) with $\mathcal{B}_{\Gamma_k}$ as its basin of attraction. If $\Gamma_k$ undergoes a degenerate transcritical bifurcation (DTB) for some parameter value $\theta = \theta_0 \in \boldsymbol{\theta}$, then the norm of the PLRNN loss gradient, $\left\| \frac{\partial \mathcal{L}_t}{\partial \theta} \right\|$, tends to infinity at $\theta = \theta_0$ for every $\boldsymbol{z}_1 \in \mathcal{B}_{\Gamma_k}$, i.e. $\lim_{\theta \to \theta_0} \left\| \frac{\partial \mathcal{L}_t}{\partial \theta} \right\| = \infty$.*

*Proof.* See Appx. A.2.1 □

However, bifurcations may also cause gradients to suddenly vanish, as it is the case for a BCB as established by our second theorem:

**Theorem 2.** *Consider a PLRNN of the form (2) with parameters $\boldsymbol{\theta} = \{\boldsymbol{A}, \boldsymbol{W}, \boldsymbol{h}\}$. Assume that it has a stable fixed point or $k$-cycle $\Gamma_k$ ($k \geq 1$) with $\mathcal{B}_{\Gamma_k}$ as its basin of attraction. If $\Gamma_k$ undergoes a border collision bifurcation (BCB) for some parameter value $\theta = \theta_0 \in \boldsymbol{\theta}$, then the gradient of the loss function, $\frac{\partial \mathcal{L}_t}{\partial \theta}$, vanishes at $\theta = \theta_0$ for every $\boldsymbol{z}_1 \in \mathcal{B}_{\Gamma_k}$, i.e. $\lim_{\theta \to \theta_0} \left\| \frac{\partial \mathcal{L}_t}{\partial \theta} \right\| = 0$.*

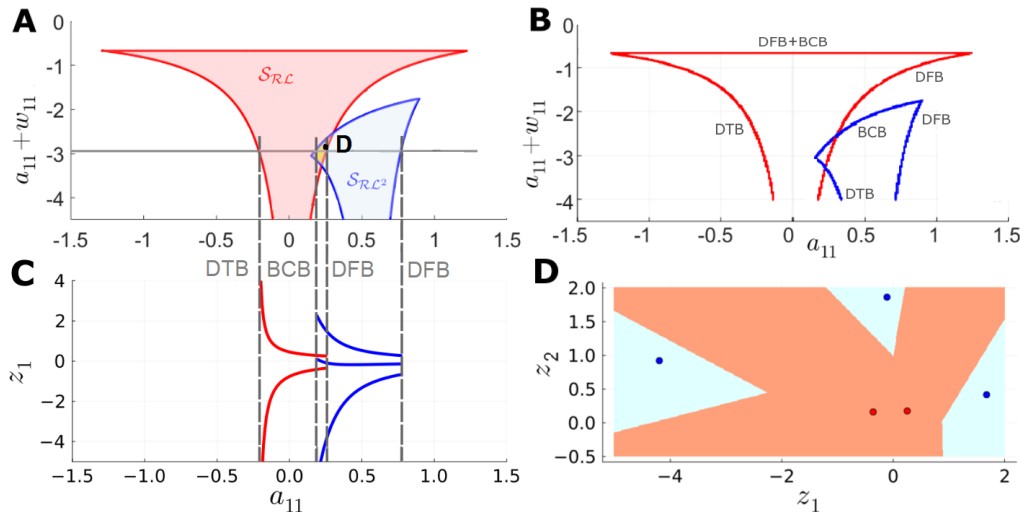

Figure 1: A) Analytically calculated stability regions for a 2-cycle ($\mathcal{S}_{\mathcal{RL}}$, red), a 3-cycle ($\mathcal{S}_{\mathcal{RL}^2}$, blue), and their intersection (yellow) in the $(a_{11}, a_{11} + w_{11})$-parameter plane for the system eq. (3) with $a_{22} = 0.2$, $w_{21} = 0.5$. B) Same as determined by SCYFI, with bifurcation curves bordering the stability regions labeled by the type of bifurcation (DTB = Degenerate Transcritical Bifurcation, BCB= Border Collision Bifurcation, DFB = Degenerate Flip Bifurcation). C) Bifurcation graph (showing the stable cyclic points in the $z_1$ coordinate) along the cross-section in A indicated by the gray line, illustrating the different types of bifurcation encountered when moving in and out of the various stability regions in A. D) State space at the point denoted 'D' in A (for $a_{11} = 0.253$, $a_{11} + w_{11} = -2.83$), where the 2-cycle (red) and 3-cycle (blue) co-exist for the same parameter settings, with their corresponding basins of attraction indicated by lighter colors.

*Proof.* See Appx. A.2.2. ☐

**Corollary 1.** *Assume that the PLRNN (2) has a stable fixed point $\Gamma_1$ with $\mathcal{B}_{\Gamma_1}$ as its basin of attraction. If $\Gamma_1$ undergoes a degenerate flip bifurcation (DFB) for some parameter value $\theta = \theta_0 \in \boldsymbol{\theta}$, then this will always coincide with a BCB of a 2-cycle, and as a result $\lim_{\theta \to \theta_0} \left\| \frac{\partial \mathcal{L}_t}{\partial \theta} \right\| = 0$ for every $z_1 \in \mathcal{B}_{\Gamma_1}$.*

*Proof.* See Appx. A.2.3. ☐

Hence, certain bifurcations will inevitably cause gradients to suddenly explode or vanish, and often induce abrupt jumps in the loss (see Appx. A.3.2 for when this will happen for a BCB). We emphasize that these results are general and hold *for systems of any dimension*, as well as *in the presence of inputs*. Since inputs do not affect the Jacobians in eqn. (70), (71) and (76), they do not change the theorems (even if they would affect the Jacobians, Theorem 2 would be unaltered, and Theorem 1 could be amended in a straightforward way). Furthermore, since we are addressing bifurcations that occur during model training, from this angle inputs may simply be treated as either additional parameters (if piecewise constant) or states of the system (without changing any of the mathematical derivations). In fact, mathematically, any non-autonomous dynamical system (RNN with inputs) can always and strictly be reformulated as an autonomous system (RNN without inputs), see [3, 47, 66].

## 4 Heuristic algorithm for finding PLRNN bifurcation manifolds

### 4.1 Searcher for fixed points and cycles (SCYFI): motivation and validation

In sect. 3.1 and Appx. A.1.2-A.1.4 we derived existence and stability regions for fixed points and low order ($k \leq 3$) cycles in $2d$ PLRNNs with specific parameter constraints analytically. For higher-order cycles and higher-dimensional PLRNNs (or any other ReLU-type RNN) this is no longer feasible due to the combinatorial explosion in the number of subregions that need to be considered as $M$ and $k$ increase. Here we therefore introduce an efficient search algorithm for finding all $k$-cycles of a given PLRNN, which we call *Searcher for Cycles and Fixed points*: **SCYFI** (Algorithm 1). Once all $k$-cycles ($k \geq 1$) have been detected on some parameter grid, the stability-/existence regions of these objects and thereby the bifurcation manifolds can be determined. $k$-cycles were defined in sect. 3.1,

**Algorithm 1** SCYFI

The algorithm is iteratively run with $k = 1 \cdots K_{max}$, with $K_{max}$ the max. order of cycles tested

**Input:** PLRNN parameters $\boldsymbol{A}$, $\boldsymbol{W}$, $\boldsymbol{h}$;

$\mathcal{L} = \{\mathcal{L}_n\}_{n=1}^{k-1}$: collection of all sets $\mathcal{L}_n = \{\{\boldsymbol{z}_l^{(m)}\}_{l=1}^n\}_{m=1}^{M_n}$ of all lower order $n$-cycles discovered so far, where $M_n$ is the number of found cycles $\{\boldsymbol{z}_l^{(m)}\}_{l=1}^n$ of order $n$, with corresponding ReLU derivative matrices $\{\boldsymbol{D}_l^{(m)}\}_{l=1}^n$.

**Parameters:**

$N_{out}$: max. number of random initialisations;

$N_{in}$: max. number of iterations

**Output:** $\mathcal{L} \cup \mathcal{L}_k$; $\mathcal{L}_k$: set of all discovered $k$-cycles

```
 1: ℒ_k = {}
 2: i → 0
 3: while i < N_out ... do
 4:     Select k subregions D_init = {D_1, D_2, ..., D_k} at random with replacement
 5:     c → 0
 6:     while c < N_in ... do
 7:         Solve eq. (4) for a cycle candidate {z*_l}^k_{l=1} with W_{Ω(k−r)} as defined in eq. (2) based
            on D_init
 8:         Determine {D*_l}^k_{l=1} based on the signs of the corresponding components of {z*_l}^k_{l=1}
 9:         if D_init = {D*_l}^k_{l=1} (self-consistency) and ∀1 ≤ s ≤ k, ∀{z^(m)_l}^{k−s+1}_{l=1} ∈ ℒ_{k−s+1} :
            {z^(m)_l}^{k−s+1}_{l=1} ⊄ {z*_l}^k_{l=1} then
10:             ℒ_k → ℒ_k ∪ {{z*_l}^k_{l=1}}
11:             i → c → 0
12:         else
13:             D_init → {D*_l}^k_{l=1}
14:         end if
15:         c → c + 1
16:     end while
17:     i → i + 1
18: end while
```

and for the PLRNN, eq. (2), are given by the set of $k$-periodic points $\{\boldsymbol{z}_1^*, \ldots, \boldsymbol{z}_l^*, \ldots, \boldsymbol{z}_k^*\}$, where

$$\boldsymbol{z}_k^* = \left( \mathbb{1} - \prod_{r=0}^{k-1} \boldsymbol{W}_{\Omega(k-r)} \right)^{-1} \left[ \sum_{j=2}^{k-1} \prod_{r=0}^{k-j} \boldsymbol{W}_{\Omega(k-r)} + \mathbb{1} \right] \boldsymbol{h}, \tag{4}$$

if $\left( \mathbb{1} - \prod_{r=0}^{k-1} \boldsymbol{W}_{\Omega(k-r)} \right)$ is invertible (if not, we are dealing with a bifurcation or a continuous set of fixed points). The other periodic points are $\boldsymbol{z}_l = F^l(\boldsymbol{z}_k^*)$, $l = 1, \cdots, k-1$, with corresponding matrices $\boldsymbol{W}_{\Omega(l)} = \boldsymbol{A} + \boldsymbol{W}\boldsymbol{D}_l$. Now, if the diagonal entries in $\boldsymbol{D}_l$ are consistent with the signs of the corresponding states $z_{ml}^*$, i.e. if $d_{mm}^{(l)} = 1$ if $z_{ml}^* > 0$ and $d_{mm}^{(l)} = 0$ otherwise for all $l$, $\{\boldsymbol{z}_1^*, \ldots, \boldsymbol{z}_k^*\}$ is a true cycle of eq. (2), otherwise we call it *virtual*. To find a $k$-cycle, since an $M$-dimensional PLRNN harbors $2^M$ different linear sub-regions, there are approximately $2^{Mk}$ different combinations of configurations of the matrices $\boldsymbol{D}_l$, $l = 1 \ldots k$, to consider (strictly, mathematical constraints rule out some of these possibilities, e.g. not all periodic points can lie within the same sub-region/orthant).

Clearly, for higher-dimensional PLRNNs and higher cycle orders exhaustively searching this space becomes unfeasible. Instead, we found that the following heuristic works surprisingly well: First, for some order $k$ and a random initialization of the matrices $\boldsymbol{D}_l$, $l = 1 \ldots k$, generate a cycle candidate by solving eq. (4). If each of the points $\boldsymbol{z}_l^*$, $l = 1 \ldots k$, is consistent with the diagonal entries in the corresponding matrix $\boldsymbol{D}_l$, $l = 1 \ldots k$, and none of them is already in the current library of cyclic points, then a true $k$-cycle has been identified, otherwise the cycle is virtual (or a super-set of lower-order cycles). We discovered that the search becomes extremely efficient, without the need to exhaustively consider all configurations, if a new search loop is re-initialized at the last visited virtual cyclic point (i.e., all *inconsistent* entries $d_{mm}^{(l)}$ in the matrices $\boldsymbol{D}_l$, $l = 1 \ldots k$, are flipped, $d_{mm}^{(l)} \to 1 - d_{mm}^{(l)}$, to bring them into agreement with the signs of the solution points, $\boldsymbol{z}_l^* = [z_{ml}^*]$, of eq.

(4), thus yielding the next initial configuration). It is straightforward to see that this procedure almost surely converges if $N_{out}$ is chosen large enough, see Appx. A.2.4. The whole procedure is formalized in Algorithm 1, and the code is available at `https://github.com/DurstewitzLab/SCYFI`.

To validate the algorithm, we can compare analytical solutions as derived in sect. 3.1 to the output of the algorithm. To delineate all existence and stability regions, the algorithm searches for all $k$-cycles up to some maximum order $K$ along a fine grid across the $(a_{11}, \ a_{11} + w_{11})$-parameter plane. A bifurcation happens whenever between two grid points a cycle appears, disappears, or changes stability (as determined from the eigenvalue spectrum of the respective $k^{th}$-order Jacobian). The results of this procedure are shown in Fig. 1B, illustrating that the analytical solutions for existence and stability regions precisely overlap with those identified by Algorithm 1 (see also Fig. S7).

## 4.2  Numerical and theoretical results on SCYFI's scaling behavior

Because of the combinatorial nature of the problem, it is generally not feasible to obtain ground truth settings in higher dimensions for SCYFI to compare to. To nevertheless assess its scaling behavior, we therefore studied two specific scenarios. For an exhaustive search, the expected and median numbers of linear subregions $n$ until an object of interest (fixed point or cycle) is found, i.e. the number of $\{\boldsymbol{D}_{1:k}\}$ constellations that need to be inspected until the first hit, are given by

$$E[n] = \frac{N+1}{m+1} = \frac{2^{Mk}+1}{m+1}, \quad \overline{n} = \min\left\{ n \in \mathbb{N} \left| \binom{2^{Mk}-n}{m} \leq \frac{1}{2}\binom{2^{Mk}}{m} \right. \right\} \tag{5}$$

with $m$ being the number of existing $k$-cycles and $N$ the total number of combinations, as shown in Ahlgren [1] (assuming no prior knowledge about the mathematical limitations when drawing regions). The median $\overline{n}$ as well as the actual median number of to-be-searched combinations required by SCYFI to find at least one $k$-cycle is given for low-dimensional systems in Fig. 2A as a function of cycle order $k$, and can be seen to be surprisingly linear as confirmed by linear regression fits to the data (see Fig. 2 legend for details). To assess scaling as a function of dimensionality $M$, we explicitly constructed systems with one known fixed point (see Appx. A.3.1 for details) and determined the number $n$ of subregions required by SCYFI to detect this embedded fixed point (Fig. 2B). In general, the scaling depended on the system's eigenspectrum, but for reasonable scenarios was polynomial or even sublinear (Fig. 2B, see also Fig. S8). In either case, the number of required SCYFI iterations scaled much more favorably than would be expected from an exhaustive search.

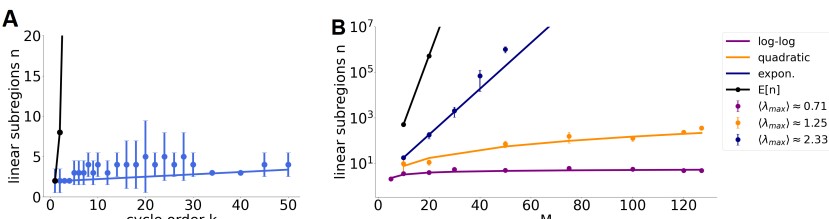

Figure 2: A) Number of linear subregions $n$ searched until at least one cycle of order $k$ was found by SCYFI (blue) vs. the median number $\overline{n}$ an exhaustive search would take by randomly drawing combinations without replacement (black) as a function of cycle order ($M = 2$ fixed). Each data point represents the median of 50 different initializations across 5 different PLRNN models. Error bars = median absolute deviation. Linear regression fit using weighted least-squares ($R^2 \approx 0.998, p < 10^{-30}$). B) Number of linear subregions $n$ searched until a specific fixed point was found as function of dimensionality $M$ for different eigenvalue spectra (see Appx. A.3.1 for details).

How could this surprisingly good scaling behavior be explained? As shown numerically in Fig. S9, when we initiate SCYFI in different randomly selected linear subregions, it converges to the subregions including the dynamical objects of interest exponentially fast, offsetting the combinatorial explosion. A more specific and stronger theoretical result about SCYFI's convergence speed can be obtained under certain conditions on the parameters (which agrees nicely with the numerical results in Fig. 2). It rests on the observation that SCYFI is designed to move *only among subregions containing virtual or actual fixed points or cycles*, based on the fact that it is always reinitialized with the next virtual fixed (cyclic) point in case the consistency check fails. The result can be stated as follows:

**Theorem 3.** *Consider a PLRNN of the form (2) with parameters $\boldsymbol{\theta} = \{\boldsymbol{A}, \boldsymbol{W}, \boldsymbol{h}\}$. Under certain conditions on $\boldsymbol{\theta}$ (for which $\|\boldsymbol{A}\| + \|\boldsymbol{W}\| < 1$), SCYFI will converge in at most linear time.*

*Proof.* See Appx. A.2.5 □

## 5 Loss landscapes and bifurcation curves

### 5.1 Bifurcations and loss jumps in training

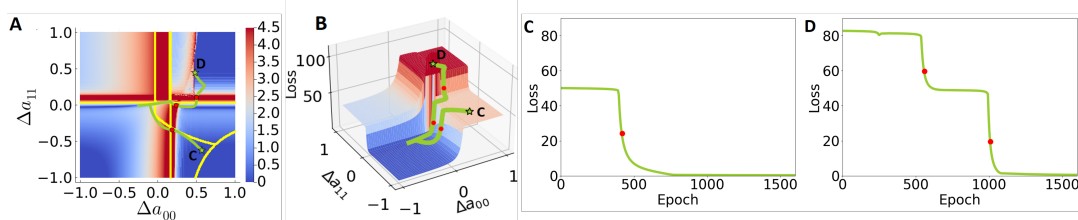

Figure 3: A) Logarithm of gradient norm of the loss in PLRNN parameter space, with ground truth parameters centered at $(0, 0)$. Superimposed in yellow are bifurcation curves computed by SCYFI, and in green two examples of training trajectories from different parameter initial conditions (indicated by the stars). Red dots indicate the bifurcation crossing time points shown in C & D. B) Relief plot of the loss landscape from A to highlight the differences in loss altitude associated with the bifurcations. C) Loss during the training run represented by the green trajectory labeled C in A and B. Red dot indicates the time point of bifurcation crossing corresponding to the red dot in A and B. D) Same for trajectory labeled D in A and B.

Fig. 3 provides a $2d$ toy example illustrating the tight association between the loss landscape and bifurcation curves, as determined through SCYFI, for a PLRNN trained by BPTT on reproducing a specific 16-cycle. Fig. 3A depicts a contour plot of the gradient norms with overlaid bifurcation curves in yellow, while Fig. 3B shows the MSE loss landscape as a relief for better appreciation of the sharp changes in loss height associated with the bifurcation curves. Shown in green are two trajectories from two different parameter initializations traced out during PLRNN training in parameter space, where training was confined to only those two parameters given in the graphs (i.e., all other PLRNN parameters were kept fixed during training for the purpose of this illustration). As confirmed in Fig. 3C & D, as soon as the training trajectory crosses the bifurcation curves in parameter space, a huge jump in the loss associated with a sudden increase in the gradient norm occurs. This illustrates empirically and graphically the theoretical results derived in sect. 3.

Next we illustrate the application of SCYFI on a real-world example, learning the behavior of a rodent spiking cortical neuron observed through time series measurements of its membrane potential (note that spiking is a highly nonlinear behavior involving fast within-spike and much slower between-spike time scales). For this, we constructed a 6-dimensional delay embedding of the membrane voltage [53, 28], and trained a PLRNN with one hidden layer (cf. eq. 6) using BPTT with sparse teacher forcing (STF) [37] to approximate the dynamics of the spiking neuron (see Appx. A.3.2 for a similar analysis on a biophysical neuron model). With $M = 6$ latent states and $H = 20$ hidden dimensions, the trained PLRNN comprises $2^{20}$ different linear subregions and $|\boldsymbol{\theta}| = 272$ parameters, much higher-dimensional than the toy example considered above. Fig. 4A gives the MAE loss as a function of training epoch (i.e., single SGD updates), while Figs. 4B & C illustrate the well-trained behavior in time (Fig. 4B) and in a 2-dimensional projection of the model's state space obtained by PCA (Fig. 4C). The loss curve exhibits several steep jumps. Zooming into one of these regions (Fig. 4A; indicated by the red box) and examining the transitions in parameter space using SCYFI, we find they are indeed produced by bifurcations, with an example given in Fig. 4D. Note that we are now dealing with high-dimensional state and parameter spaces, such that visualization of results becomes tricky. For the bifurcation diagram in Fig. 4D we therefore projected all extracted $k$-cycles ($k \geq 1$) onto a line given by the PCA-derived maximum eigenvalue component, and plotted this as a function of training epoch.[1] Since SCYFI extracts all $k$-cycles and their eigenvalue spectrum, we can also determine the type of bifurcation that caused the jump. While before the loss jump the PLRNN already produced time series quite similar to those of the physiologically recorded cell (Fig. 4E), a

---

[1] Of course, very many of the PLRNN parameters may change from one epoch to the next.

DTB (cf. Theorem 1) produced catastrophic forgetting of the learned behavior with the PLRNN's states suddenly diverging to minus infinity (Fig. 4F; Fig. S10 also provides an example of a BCB during PLRNN training, and Fig. S11 an example of a DFB). This illustrates how SCYFI can be used to analyze the training process with respect to bifurcation events also for high-dimensional real-world examples, as well as the behavior of the trained model (Fig. 4C).

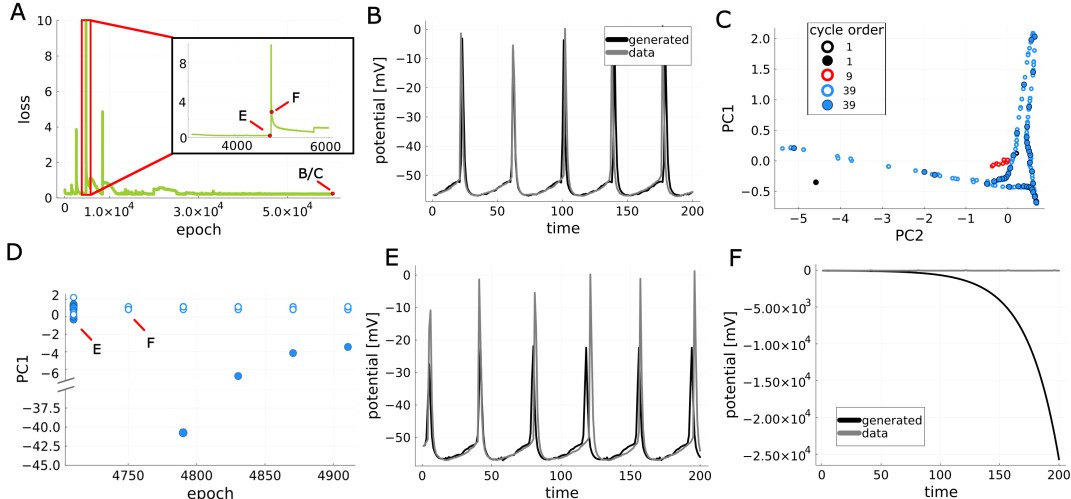

Figure 4: A) Loss across training epochs for a PLRNN with one hidden layer trained on electro-physiological recordings from a cortical neuron. Red box zooms in on one of the training phases with a huge loss jump, caused by a DTB. Letters refer to selected training epochs in other subpanels. B) Time series of true (gray) and PLRNN-simulated (black) membrane potential in the well trained regime (see A). C) All fixed points and cycles discovered by SCYFI for the well-trained model in state space projected onto the first two principle components using PCA. Filled circles represent stable and open circles unstable objects. The stable 39-cycle corresponds to the spiking behavior. D) Bifurcation diagram of the PLRNN as a function of training epoch around the loss peak in A. Locations of stable (filled circles) and unstable (open circles) objects projected onto the first principle component. E) Model behavior as in B shortly before the DTB and associated loss jump (from the epoch indicated in A, D). F) Model behavior as in B right around the DTB (diverging to $-\infty$).

## 5.2 Implications for designing training algorithms

What are potential take-homes of the results in sects. 3.2 & 5.1 for designing RNN training algorithms? One possibility is to design smart initialization or training procedures that aim to place or push an RNN into the right topological regime by taking big leaps in parameter space whenever the current regime is not fit for the data, rather than dwelling within a wrong regime for too long. These ideas are discussed in a bit more depth in Appx.A.3.3, with a proof of concept in Fig. S12.

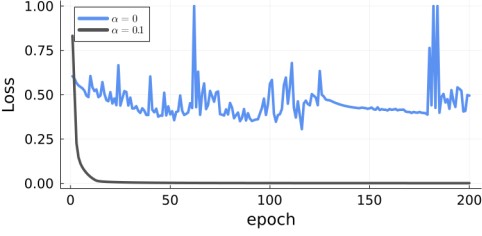

Figure 5: Example loss curves during training a PLRNN ($M = 10$) on a $2d$ cycle using gradient descent, once without GTF ($\alpha = 0$, blue curve) but gradient clipping, and once with GTF ($\alpha = 0.1$). Note that without GTF there are several sharp loss jumps associated with bifurcations in the PLRNN parameters, while activating GTF leads to a smooth loss curve avoiding bifurcations. Note: For direct comparability both loss curves were cut off at 4 and then scaled to $[0, 1]$. The absolute loss is much lower for GTF.

More importantly, however, we discovered that the recently proposed technique of 'generalized teacher forcing (GTF)' [24] tends to circumvent bifurcations in RNN training altogether, leading to much faster convergence as illustrated in Fig. 5. The way this works is that GTF, by trading off forward-iterated RNN latent states with data-inferred states according to a specific annealing schedule during training (see Appx. A.2.6), tends to pull the RNN directly into the right dynamical regime. In fact, for DTBs we can strictly prove these will never occur in PLRNN training with the right adjustment of the GTF parameter:

**Theorem 4.** *Consider a PLRNN of the form (2) with parameters $\theta = \{A, W, h\}$. Assume that it has a stable fixed point or $k$-cycle $\Gamma_k$ ($k \geq 1$) that undergoes a degenerate transcritical bifurcation (DTB) for some parameter value $\theta = \theta_0 \in \theta$.*

*(i) If $\|A\| + \|W\| \leq 1$, then for any GTF parameter $0 < \alpha < 1$, GTF controls the system, avoiding a DTB and, hence, gradient divergence at $\theta_0$.*

*(ii) If $\|A\| + \|W\| = r > 1$, then for any $1 - \frac{1}{r} < \alpha < 1$, GTF prevents a DTB and, hence, gradient divergence at $\theta_0$.*

*Proof.* See Appx. A.2.6. □

As this example illustrates, we may be able to amend training procedures such as to avoid specific types of bifurcations.

## 6   Discussion

DS theory [3, 47, 58] is increasingly appreciated in the ML/AI community as a powerful mathematical framework for understanding both the training process of ML models [49, 54, 44, 16] as well as the behavior of trained models [62, 33, 61]. While the latter is generally useful for understanding how a trained RNN performs a given ML task, with prospects of improving found solutions, it is in fact imperative in areas like science or medicine where excavating the dynamical behavior and repertoire of trained models yields direct insight into the underlying physical, biological, or medical processes the model is supposed to capture. However, application of DS theory is often not straightforward, especially when dealing with higher-dimensional systems, and commonly requires numerical routines that may only find some of the dynamical objects of interest, and also only approximate solutions. One central contribution of the present work therefore was the design of a novel algorithm, SCYFI, that can exactly locate fixed points and cycles of a wide class of ReLU-based RNNs. This provides an efficient instrument for the DS analysis of trained models, supporting their interpretability and explainability.

A surprising observation was that SCYFI often finds cycles in only linear time, despite the combinatorial nature of the problem, a feature shared with the famous Simplex algorithm for solving linear programming tasks [36, 32, 57]. While we discovered numerically that SCYFI for empirically relevant scenarios converges surprisingly fast, deriving strict theoretical guarantees is hard, and so far we could establish stronger theoretical results on its convergence properties only under specific assumptions on the RNN parameters. Further theoretical work is therefore necessary to precisely understand why the algorithm works so effectively.

In this work we applied SCYFI to illuminate the training process itself. Since RNNs are themselves DS, they are subject to different forms of bifurcations during training as their parameters are varied under the action of a training algorithm (similar considerations may apply to very deep NNs). It has been recognized for some time that bifurcations in RNN training may give rise to sudden jumps in the loss [44, 16], but the phenomenon has rarely been treated more systematically and mathematically. Another major contribution of this work thus was to formally prove a strict connection between three types of bifurcations and abrupt changes in the gradient norms, and to use SCYFI to further reveal such events during PLRNN training on various example systems. There are numerous other types of bifurcations (e.g., center bifurcations, Hopf bifurcations etc.) that are likely to impact gradients during training, only for a subset of which we could provide formal proofs here. As we have demonstrated, understanding the topological and bifurcation landscape of RNNs could help improve training algorithms and provide insights into their working. Hence, a more general understanding of how various types of bifurcation affect the training process in a diverse range of RNN architectures is a promising future avenue not only for our theoretical understandings of RNNs, but also for guiding future algorithm design.

**Acknowledgements** This work was supported by the German Research Foundation (DFG) through individual grants Du 354/10-1 & Du 354/15-1 to DD, within research cluster FOR-5159 ("Resolving prefrontal flexibility"; Du 354/14-1), and through the Excellence Strategy EXC 2181/1 – 390900948 (STRUCTURES). We also thank Mahasheta Patra for lending us code for graphing analytically derived bifurcation diagrams and state spaces.

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

# A  Appendix

## A.1  Analysis of bifurcation curves

### A.1.1  PLRNNs

The standard PLRNN [18], given in eq. (1) in sect. 3.1, was defined by

$$\boldsymbol{z}_t = F_{\boldsymbol{\theta}}(\boldsymbol{z}_{t-1}, \boldsymbol{s}_t) = \boldsymbol{A}\,\boldsymbol{z}_{t-1} + \boldsymbol{W}\phi(\boldsymbol{z}_{t-1}) + \boldsymbol{C}\boldsymbol{s}_t + \boldsymbol{h},$$

where $\phi(\boldsymbol{z}_{t-1}) = \max(\boldsymbol{z}_{t-1}, 0)$. There are various extensions of this basic architecture like the dendPLRNN [10] or the 'shallow PLRNN' (shPLRNN) [24], as used in sect. 5.1 for training on single cell membrane potentials. The latter is essentially a 1-hidden-layer version of the form

$$\boldsymbol{z}_t = F_{\boldsymbol{\theta}}(\boldsymbol{z}_{t-1}, \boldsymbol{s}_t) = \boldsymbol{A}\,\boldsymbol{z}_{t-1} + \boldsymbol{W}_1\phi(\boldsymbol{W}_2\boldsymbol{z}_{t-1} + \boldsymbol{h}_2) + \boldsymbol{C}\boldsymbol{s}_t + \boldsymbol{h}_1, \tag{6}$$

with $\boldsymbol{W}_1 \in \mathbb{R}^{M \times L}$ and $\boldsymbol{W}_2 \in \mathbb{R}^{L \times M}$, $L \geq M$, connectivity matrices, $\boldsymbol{h}_1 \in \mathbb{R}^M$, $\boldsymbol{h}_2 \in \mathbb{R}^L$ bias terms, and all other parameters and variables as in eq. (1). While this formulation is beneficial for training, the shPLRNN can essentially be rewritten in standard PLRNN form (see [24]).

Assume that $\boldsymbol{D}_{\Omega(t)} := \mathrm{diag}(\boldsymbol{d}_{\Omega(t)})$ is a diagonal matrix with an indicator vector $\boldsymbol{d}_{\Omega(t)} := (d_1, d_2, \cdots, d_M)$ such that $d_m(z_{m,t}) =: d_m = 1$ whenever $z_{m,t} > 0$, and zero otherwise. Then eq. (1) can be rewritten as

$$\boldsymbol{z}_t = (\boldsymbol{A} + \boldsymbol{W}\boldsymbol{D}_{\Omega(t-1)})\boldsymbol{z}_{t-1} + \boldsymbol{C}\boldsymbol{s}_t + \boldsymbol{h} =: \boldsymbol{W}_{\Omega(t-1)}\,\boldsymbol{z}_{t-1} + \boldsymbol{C}\boldsymbol{s}_t + \boldsymbol{h}.$$

Let us ignore the inputs for simplicity. There are $2^M$ different configurations for matrix $\boldsymbol{D}_{\Omega(t-1)}$ and so $2^M$ different forms for matrix $\boldsymbol{W}_{\Omega(t-1)}$ in the system

$$\boldsymbol{z}_t = F_{\boldsymbol{\theta}}(\boldsymbol{z}_{t-1}) = \boldsymbol{W}_{\Omega(t-1)}\,\boldsymbol{z}_{t-1} + \boldsymbol{h}. \tag{7}$$

Thus, the phase space of the system is divided into $2^M$ sub-regions corresponding to the indexed matrices

$$\boldsymbol{W}_{\Omega^k} := \boldsymbol{A} + \boldsymbol{W}\boldsymbol{D}_{\Omega^k}, \quad k = 1, 2, \cdots, 2^M, \tag{8}$$

see [38, 39] for more details. For $M = 2$, assuming

$$\boldsymbol{W} = \begin{pmatrix} w_{11} & 0 \\ w_{21} & 0 \end{pmatrix}, \tag{9}$$

in (8), we have

$$\boldsymbol{W}_{\Omega^1} = \boldsymbol{W}_{\Omega^3} = \begin{pmatrix} a_{11} & 0 \\ 0 & a_{22} \end{pmatrix} = \boldsymbol{A},$$

$$\boldsymbol{W}_{\Omega^2} = \boldsymbol{W}_{\Omega^4} = \begin{pmatrix} a_{11} + w_{11} & 0 \\ w_{21} & a_{22} \end{pmatrix}. \tag{10}$$

Hence, for this parameter constellation, the map simplifies as there exists only one border which divides the phase space into two distinct sub-regions, such that (7) can be rewritten as a map of the form

$$\begin{pmatrix} z_{1,t} \\ z_{2,t} \end{pmatrix} = T(z_{1,t-1}, z_{2,t-1})$$

$$= \begin{cases} T_{\mathcal{L}}(z_{1,t-1}, z_{2,t-1}) = \underbrace{\begin{pmatrix} a_l & c \\ b_l & d \end{pmatrix}}_{\boldsymbol{A}_{\mathcal{L}}} \begin{pmatrix} z_{1,t-1} \\ z_{2,t-1} \end{pmatrix} + \begin{pmatrix} h_1 \\ h_2 \end{pmatrix}; & z_{1,t-1} \leq 0 \\[4mm] T_{\mathcal{R}}(z_{1,t-1}, z_{2,t-1}) = \underbrace{\begin{pmatrix} a_r & c \\ b_r & d \end{pmatrix}}_{\boldsymbol{A}_{\mathcal{R}}} \begin{pmatrix} z_{1,t-1} \\ z_{2,t-1} \end{pmatrix} + \begin{pmatrix} h_1 \\ h_2 \end{pmatrix}; & z_{1,t-1} \geq 0 \end{cases}, \tag{11}$$

with $a_l = a_{11}$, $a_r = a_{11} + w_{11}$, $b_r = w_{21}$, $d = a_{22}$, $b_l = c = 0$. The map (11) is a PWL dynamical system whose phase space is split into left and right half-planes (sub-regions) by the borderline $\Sigma$ ($z_2$-axis). Note that bifurcation curves of the $2d$ PLRNN (7) in the $(a_{11}, a_{11} + w_{11})$-parameter space can be determined analogous to those of the PWL map (11) in the $(a_l, a_r)$-parameter space.

Another way to simplify the PLRNN to a $2d$ ($M = 2$) PWL map with just a single border is to remove one of the ReLU nonlinearities and define $\phi(\boldsymbol{z}_{t-1}) = (\phi_1(z_{1,t-1}), \beta\, z_{2,t-1})^\mathsf{T}$, where $\beta \in \mathbf{R}$ and $\phi_1$ is some variant of the $ReLU$ function such as the leaky or parametric $ReLU$ given by

$$\phi_1(z) = \begin{cases} z; & z > 0 \\ \alpha\, z; & z \leq 0 \end{cases} \qquad (\alpha \in \mathbf{R}). \tag{12}$$

Then $\boldsymbol{D}_{\Omega(t)} := \operatorname{diag}(d_1, \beta)$ such that

$$d_1(z_{1,t}) =: d_1 = \begin{cases} 1; & z_{1,t} > 0 \\ \alpha; & z_{1,t} \leq 0 \end{cases}, \tag{13}$$

and so

$$\boldsymbol{W}_{\Omega^1} = \boldsymbol{W}_{\Omega^3} = \begin{pmatrix} a_{11} + \alpha w_{11} & \beta w_{12} \\ \alpha w_{21} & a_{22} + \beta w_{22} \end{pmatrix} =: \begin{pmatrix} a_l & c \\ b_l & d \end{pmatrix},$$

$$\boldsymbol{W}_{\Omega^2} = \boldsymbol{W}_{\Omega^4} = \begin{pmatrix} a_{11} + w_{11} & \beta w_{12} \\ w_{21} & a_{22} + \beta w_{22} \end{pmatrix} =: \begin{pmatrix} a_r & c \\ b_r & d \end{pmatrix}. \tag{14}$$

This gives another example of a representative of $2d$ PWL maps with only one border defined in eq. (11). We are pointing this out because eq. (11) is a generic system considered more widely in the discrete dynamical systems literature [5, 6], and also was the basis for the analyses below and in Fig. 1.

### A.1.2 Fixed points of the map (11) and their bifurcations

For $a_l, a_r, b_l, b_r, c, d, h_1, h_2 \in \mathbb{R}$, the map (11) has the following two fixed points

$$\mathcal{O}_{\mathcal{L}/\mathcal{R}} = \left(z_1^{\mathcal{L}/\mathcal{R}}, z_2^{\mathcal{L}/\mathcal{R}}\right)^\mathsf{T} = \left( \frac{(1-d)\,h_1 + c\,h_2}{(1-d)(1-a_{l/r}) - b_{l/r}\,c}, \frac{b_{l/r}\,h_1 + (1-a_{l/r})\,h_2}{(1-d)(1-a_{l/r}) - b_{l/r}\,c} \right)^\mathsf{T}. \tag{15}$$

The fixed points $\mathcal{O}_{\mathcal{L}}$ and $\mathcal{O}_{\mathcal{R}}$ exist iff $z_1^{\mathcal{L}} < 0$ and $z_1^{\mathcal{R}} > 0$ respectively; otherwise they are virtual. Hence, the existence regions of admissible fixed points are

$$E_{\mathcal{O}_{\mathcal{L}}} = \left\{ (h_1, h_2, a_l, b_l, c, d) \,\Big|\, \frac{(1-d)\,h_1 + c\,h_2}{(1-d)(1-a_l) - b_l\,c} < 0 \right\},$$

$$E_{\mathcal{O}_{\mathcal{R}}} = \left\{ (h_1, h_2, a_r, b_r, c, d) \,\Big|\, \frac{(1-d)\,h_1 + c\,h_2}{(1-d)(1-a_r) - b_r\,c} > 0 \right\}. \tag{16}$$

Let $\mathcal{D}_{\mathcal{L}/\mathcal{R}}$ be the determinant and $\mathcal{T}_{\mathcal{L}/\mathcal{R}}$ the trace of $\boldsymbol{A}_{\mathcal{L}/\mathcal{R}}$, and

$$\mathcal{P}_{\mathcal{L}/\mathcal{R}}(\lambda) = \lambda^2 - (a_{l/r} + d)\lambda + a_{l/r}\,d - b_{l/r}\,c = \lambda^2 - \mathcal{T}_{\mathcal{L}/\mathcal{R}}\,\lambda + \mathcal{D}_{\mathcal{L}/\mathcal{R}}, \tag{17}$$

its characteristic polynomial. The corresponding eigenvalues are given by

$$\lambda_{1,2}(\mathcal{O}_{\mathcal{L}/\mathcal{R}}) = \frac{a_{l/r} + d}{2} \pm \frac{\sqrt{(a_{l/r} - d)^2 + 4\,b_{l/r}\,c}}{2} = \frac{\mathcal{T}_{\mathcal{L}/\mathcal{R}}}{2} \pm \frac{\sqrt{\mathcal{T}_{\mathcal{L}/\mathcal{R}}^2 - 4\mathcal{D}_{\mathcal{L}/\mathcal{R}}}}{2}, \tag{18}$$

which are always real for $b_{l/r}\,c \geq 0$, while for $b_{l/r}\,c < 0$ they are real provided that $|a_{l/r} - d| > 2\sqrt{-b_{l/r}\,c}$. For complex conjugate eigenvalues of $\boldsymbol{A}_{\mathcal{L}/\mathcal{R}}$ obviously $|\lambda|^2 = \mathcal{D}_{\mathcal{L}/\mathcal{R}}$. Thus computing the real eigenvalues, the stability condition for the fixed points is determined as

$$-(1 + \mathcal{D}_{\mathcal{L}/\mathcal{R}}) < \mathcal{T}_{\mathcal{L}/\mathcal{R}} < 1 + \mathcal{D}_{\mathcal{L}/\mathcal{R}}. \tag{19}$$

Accordingly, the stability region of the fixed points $\mathcal{O}_{\mathcal{L}}$ and $\mathcal{O}_{\mathcal{R}}$ can be obtained by $\mathcal{P}_{\mathcal{L}/\mathcal{R}}(\pm 1) = 1 \mp (a_{l/r} + d) + a_{l/r}\,d - b_{l/r}\,c > 0$ and $\mathcal{D}_{\mathcal{L}/\mathcal{R}} < 1$ as

$$\mathcal{S}_{\mathcal{L}/\mathcal{R}} = \left\{ (h_1, h_2, a_{l/r}, b_{l/r}, c, d) \in E_{\mathcal{O}_{\mathcal{L}/\mathcal{R}}} \,\Big|\, a_{l/r}\,d - b_{l/r}\,c < 1, \right.$$

$$1 \pm (a_{l/r} + d) + a_{l/r}\, d - b_{l/r}\, c > 0 \Big\}. \tag{20}$$

Note that when $\mathcal{D}_{\mathcal{L}/\mathcal{R}} < 0$, all the eigenvalues are real and so there cannot be any spiralling orbit.

**Remark 1.** *Consider the PLRNN (2) with $M = 2$. For the parameter setting (3), i.e.*

$$\boldsymbol{W}_{\Omega^1} = \boldsymbol{W}_{\Omega^3} = \begin{pmatrix} a_{11} & 0 \\ 0 & a_{22} \end{pmatrix} =: \boldsymbol{A}_{\mathcal{L}}, \qquad \boldsymbol{W}_{\Omega^2} = \boldsymbol{W}_{\Omega^4} = \begin{pmatrix} a_{11} + w_{11} & 0 \\ w_{21} & a_{22} \end{pmatrix} =: \boldsymbol{A}_{\mathcal{R}},$$
$$\tag{21}$$

*the two fixed points $\mathcal{O}_{\mathcal{L}/\mathcal{R}} = \big(z_1^{\mathcal{L}/\mathcal{R}}, z_2^{\mathcal{L}/\mathcal{R}}\big)^{\mathsf{T}}$ are given by*

$$\mathcal{O}_{\mathcal{L}} = \left( \frac{h_1}{1 - a_{11}}, \frac{h_2}{1 - a_{22}} \right)^{\mathsf{T}}, \quad \mathcal{O}_{\mathcal{R}} = \left( \frac{h_1}{1 - a_{11} - w_{11}}, \frac{w_{21}\, h_1 + (1 - a_{11} - w_{11})\, h_2}{(1 - a_{22})(1 - a_{11} - w_{11})} \right)^{\mathsf{T}}. \tag{22}$$

*Hence, the existence regions of admissible fixed points are*

$$E_{\mathcal{O}_{\mathcal{L}}} = \left\{ (h_1, a_{11}, a_{22}) \Big| \frac{h_1}{1 - a_{11}} < 0 \right\}, \qquad E_{\mathcal{O}_{\mathcal{R}}} = \left\{ (h_1, a_{11}, a_{22}, w_{11}) \Big| \frac{h_1}{1 - a_{11} - w_{11}} > 0 \right\},$$

*and their stability regions can be obtained as*

$$\mathcal{S}_{\mathcal{L}} = \left\{ (h_1, a_{11}, a_{22}) \in E_{\mathcal{O}_{\mathcal{L}}} \,\middle|\, a_{11}\, a_{22} < 1, \ 1 \pm (a_{11} + a_{22}) + a_{11}\, a_{22} > 0 \right\}, \tag{23}$$

$$\mathcal{S}_{\mathcal{R}} = \left\{ (h_1, a_{11}, a_{22}, w_{11}) \in E_{\mathcal{O}_{\mathcal{R}}} \,\middle|\, (a_{11} + w_{11})a_{22} < 1, \ 1 \pm (a_{11} + w_{11} + a_{22}) + (a_{11} + w_{11})a_{22} > 0 \right\}.$$

**Remark 2.** *If $b_{l/r}\, c = 0$, then $\lambda_{1,2}(\mathcal{O}_{\mathcal{L}/\mathcal{R}})$ are real and the stability regions $\mathcal{S}_{\mathcal{L}/\mathcal{R}}$ become*

$$\mathcal{S}_{\mathcal{L}/\mathcal{R}} = \left\{ (h_1, h_2, a_{l/r}, b_{l/r}, c, d) \in E_{\mathcal{O}_{\mathcal{L}/\mathcal{R}}} \,\middle|\, b_{l/r}\, c = 0, \ -1 \le a_{l/r} \le 1, \ -1 \le d \le 1 \right\}. \tag{24}$$

The fixed points are regular saddles for all parameters that belong to

$$\left\{ (h_1, h_2, a_{l/r}, b_{l/r}, c, d) \in E_{\mathcal{O}_{\mathcal{L}/\mathcal{R}}} \,\middle|\, a_{l/r} + d > 1, \ a_{l/r}\, d - a_{l/r} - d + 1 \ < \ b_{l/r}\, c < a_{l/r}\, d \right\}, \tag{25}$$

and in this case $\lambda_1(\mathcal{O}_{\mathcal{L}/\mathcal{R}}) > 1$, $0 < \lambda_2(\mathcal{O}_{\mathcal{L}/\mathcal{R}}) < 1$. Furthermore, they are flip saddles (i.e., with one negative eigenvalue) if parameters are in

$$\left\{ (h_1, h_2, a_{l/r}, b_{l/r}, c, d) \in E_{\mathcal{O}_{\mathcal{L}/\mathcal{R}}} \,\middle|\, a_{l/r} + d > 1, \ a_{l/r}\, d < b_{l/r}\, c < a_{l/r}\, d + a_{l/r} + d + 1 \right\} \bigcup$$

$$\left\{ (h_1, h_2, a_{l/r}, b_{l/r}, c, d) \in E_{\mathcal{O}_{\mathcal{L}/\mathcal{R}}} \,\middle|\, d - a_{l/r} - d + 1 < b_{l/r}\, c < a_{l/r}\, d + a_{l/r} + d + 1, \right.$$

$$\left. 0 < a_{l/r} + d \le 1, \ a_{l/r} \right\}, \tag{26}$$

for which $\lambda_1(\mathcal{O}_{\mathcal{L}/\mathcal{R}}) > 1$, $-1 < \lambda_2(\mathcal{O}_{\mathcal{L}/\mathcal{R}}) < 0$, as well as in

$$\left\{ (h_1, h_2, a_{l/r}, b_{l/r}, c, d) \in E_{\mathcal{O}_{\mathcal{L}/\mathcal{R}}} \,\middle|\, a_{l/r} + d \le -1, \ a_{l/r}\, d < b_{l/r}\, c < a_{l/r}\, d - a_{l/r} - d + 1 \right\} \bigcup$$

$$\left\{ (h_1, h_2, a_{l/r}, b_{l/r}, c, d) \in E_{\mathcal{O}_{\mathcal{L}/\mathcal{R}}} \,\middle|\, a_{l/r}\, d + a_{l/r} + d + 1 < b_{l/r}\, c < a_{l/r}\, d - a_{l/r} + d - 1, \right.$$

$$\left. -1 < a_{l/r} + d < 0 \right\} \tag{27}$$

such that $0 < \lambda_1(\mathcal{O}_{\mathcal{L}/\mathcal{R}}) < 1$, $\lambda_2(\mathcal{O}_{\mathcal{L}/\mathcal{R}}) < -1$.

When $b_{l/r}\, c < 0$ and $|a_{l/r} - d| < 2\sqrt{-b_{l/r}\, c}$, the eigenvalues are complex conjugates and both $\mathcal{O}_{\mathcal{L}}$ and $\mathcal{O}_{\mathcal{R}}$ are spirally attracting (attracting focus) if $a_{l/r}\, d - b_{l/r}\, c < 1$. In this case, if $a_{l/r} + d > 0$ then they are clockwise spiral, while for $a_{l/r} + d < 0$ the spiralling motion will be counterclockwise. Moreover, for $a_{l/r}\, d - b_{l/r}\, c > 1$ they are repelling foci. Finally, for $a_{l/r}\, d - b_{l/r}\, c = 1$, the fixed points are locally centers and they undergo a CB at the following boundaries:

$$\mathcal{C}_{\mathcal{L}} = \left\{ (a_l, b_l, c, d) \,\middle|\, b_l\, c < 0, \ |a_l - d| < 2\sqrt{-b_l\, c}, \ a_l\, d - b_l\, c = 1 \right\},$$

$$\mathcal{C}_{\mathcal{R}} = \left\{ (a_r, b_r, c, d) \,\middle|\, b_r\, c < 0, \ |a_r - d| < 2\sqrt{-b_r\, c}, \ a_r\, d - b_r\, c = 1 \right\}. \tag{28}$$

At these boundaries, the fixed points lose their stability with a pair of complex conjugate eigenvalues crossing the unit circle. For the parameters belonging to $\mathcal{C}_{\mathcal{L}/\mathcal{R}}$, the Jacobian $J_{\mathcal{L}/\mathcal{R}}$ is a rotation matrix whose determinant is equal to 1. In this case, $J_{\mathcal{L}/\mathcal{R}}$ can be determined by a rotation number which is either rational ($\frac{p}{q}$) or irrational ($\rho$). Therefore, in some neighborhood of $\mathcal{O}_{\mathcal{L}/\mathcal{R}}$, there is a region filled with invariant ellipses such that they are periodic with period $p$ (if the rotation number is a rational number $\frac{p}{q}$) or quasiperiodic (if the rotation number is an irrational number $\rho$); for more information see [60, 59]. For $(1 - d)\, h_1 + c\, h_2 \neq 0$, at the boundary

$$\tau_{\mathcal{L}} = \left\{ (h_1, h_2, a_l, b_l, c, d) \,\middle|\, 1 - a_l - d + a_l\, d - b_l\, c = 0 \right\}, \tag{29}$$

the fixed point $\mathcal{O}_{\mathcal{L}}$ undergoes a DTB, since, if the parameters tend to $\tau_{\mathcal{L}}$, then $\mathcal{O}_{\mathcal{L}} \to \pm\infty$ and $\lambda(\mathcal{O}_{\mathcal{L}}) \to 1$. Similarly, for $(1 - d)\, h_1 + c\, h_2 \neq 0$, a DTB occurs for the fixed point $\mathcal{O}_{\mathcal{R}}$ at the boundary

$$\tau_{\mathcal{R}} = \left\{ (h_1, h_2, a_r, b_r, c, d) \,\middle|\, 1 - a_r - d + a_r\, d - b_r\, c = 0 \right\}. \tag{30}$$

A DTB of a fixed point results in its disappearance, as in this case the fixed point becomes virtual which may lead to changes in the global dynamics [6]. Furthermore, the BCB curves are given by

$$\xi_{\mathcal{L}} = \left\{ (h_1, h_2, a_l, b_l, c, d) \,\middle|\, (1 - d)(1 - a_l) - b_l\, c \neq 0, \ (1 - d)\, h_1 + c\, h_2 = 0 \right\}, \tag{31}$$

and

$$\xi_{\mathcal{R}} = \left\{ (h_1, h_2, a_r, b_r, c, d) \,\middle|\, (1 - d)(1 - a_r) - b_r\, c \neq 0, \ (1 - d)\, h_1 + c\, h_2 = 0 \right\}. \tag{32}$$

In addition, the DFB curves for the fixed points $\mathcal{O}_{\mathcal{L}}$ and $\mathcal{O}_{\mathcal{R}}$ are

$$\mathcal{F}_{\mathcal{L}} = \left\{ (h_1, h_2, a_l, b_l, c, d) \,\middle|\, 1 + a_l + d + a_l\, d - b_l\, c = 0 \right\},$$

$$\mathcal{F}_{\mathcal{R}} = \left\{ (h_1, h_2, a_r, b_r, c, d) \,\middle|\, 1 + a_r + d + a_r\, d - b_r\, c = 0 \right\}. \tag{33}$$

**Remark 3.** *The existence regions $E_{\mathcal{O}_{\mathcal{L}}}$ and $E_{\mathcal{O}_{\mathcal{R}}}$ are bounded by the BCB curves $\xi_{\mathcal{L}}$ and $\xi_{\mathcal{R}}$.*

**Remark 4.** *The stability regions $\mathcal{S}_{\mathcal{L}/\mathcal{R}}$ of fixed points (eq. (20)) are bounded by the DTB curves $\tau_{\mathcal{L}/\mathcal{R}}$ (eqn. (29) and (30)), the DFB curves $\mathcal{F}_{\mathcal{L}/\mathcal{R}}$ (eq. (33)), and the CB curves $a_{l/r}\, d - b_{l/r}\, c = 1$. For instance, for $d = 1$, $\mathcal{S}_{\mathcal{L}/\mathcal{R}}$ are illustrated in Fig. S1(a). In this case, the stability regions only exist for $b_{l/r}\, c < 0$ and $a_{l/r} - d < -2\sqrt{-b_{l/r}\, c}$. Moreover, as shown in Fig. S1(b), for $d = 0.01$, these stability regions can exist for both cases $b_{l/r}\, c < 0$, $a_{l/r} - d < -2\sqrt{-b_{l/r}\, c}$ (in blue), and $b_{l/r}\, c < 0$, $a_{l/r} - d > 2\sqrt{-b_{l/r}\, c}$ (in green), but not for $b_{l/r}\, c \geq 0$. Furthermore, if $c = 1$, there are stability regions $\mathcal{S}_{\mathcal{L}/\mathcal{R}}$ for the two cases $b_{l/r}\, c < 0$, $a_{l/r} - d < -2\sqrt{-b_{l/r}\, c}$ (in blue), and $b_{l/r}\, c > 0$ (in purple); see Fig. S2(a). Finally, when $c = 0$, as explained in Remark 2, the stability regions have the form (24), i.e.*

$$\mathcal{S}_{\mathcal{L}/\mathcal{R}} = \left\{ (h_1, h_2, a_{l/r}, b_{l/r}, c, d) \in E_{\mathcal{O}_{\mathcal{L}/\mathcal{R}}} \,\middle|\, c = 0, \ b_{l/r} \in \mathbb{R}, \ -1 \leq a_{l/r}, d \leq 1 \right\}. \tag{34}$$

*which are displayed in Fig. S2(b).*

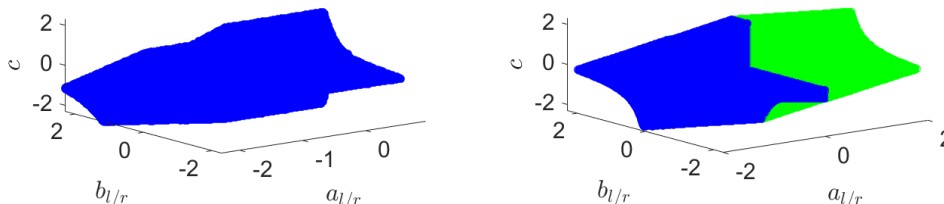

Figure S1: Stability regions $\mathcal{S}_{\mathcal{L}/\mathcal{R}}$. Left: for $d = 1$; right: for $d = 0.01$. The case $b_{l/r}\, c < 0$, $a_{l/r} - d < -2\sqrt{-b_{l/r}\, c}$ is plotted in blue, and the case $b_{l/r}\, c < 0$, $a_{l/r} - d > 2\sqrt{-b_{l/r}\, c}$ is drawn in green.

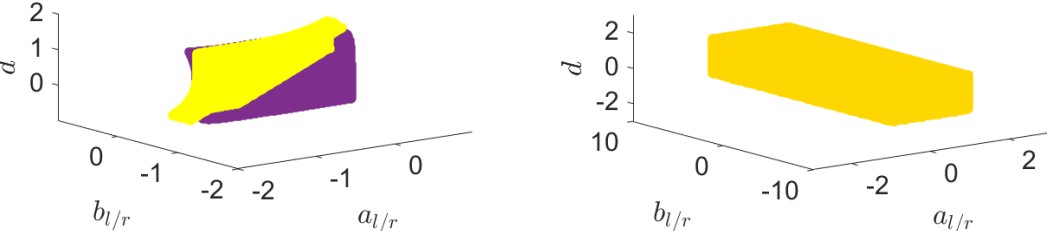

Figure S2: Stability regions $\mathcal{S}_{\mathcal{L}/\mathcal{R}}$. Left: for $c = 1$; the case $b_{l/r}\, c < 0$, $a_{l/r} - d < -2\sqrt{-b_{l/r}\, c}$ is plotted in yellow, and the case $b_{l/r}\, c > 0$ is drawn in purple. Right: for $c = 0$.

Since the system (11) is a linear map in each sub-region $\mathcal{L}$ and $\mathcal{R}$, there cannot be any $n$-cycle, $n \geq 2$, with all periodic points on only one linear side. So, all period-$n$ orbits have both letters $\mathcal{L}$ and $\mathcal{R}$ in their symbolic sequence.

### A.1.3   2-cycles of the map (11) and their bifurcations

The 2-cycle $\mathcal{O}_{\mathcal{R}\mathcal{L}}$ of the map (11) is determined by solving the equation $T_{\mathcal{L}} \circ T_{\mathcal{R}}(z_1, z_2) = (z_1, z_2)^{\mathsf{T}}$ where

$$T_{\mathcal{L}} \circ T_{\mathcal{R}}(z_1, z_2) = \begin{pmatrix} a_l\, a_r + b_r\, c & a_l\, c + c\, d \\ a_r\, b_l + b_r\, d & d^2 + b_l\, c \end{pmatrix} \begin{pmatrix} z_1 \\ z_2 \end{pmatrix} + \begin{pmatrix} c\, h_2 + h_1\,(a_l + 1) \\ b_l\, h_1 + h_2\,(d + 1) \end{pmatrix}. \tag{35}$$

In this case if $\left(I - J_{\mathcal{L}}\, J_{\mathcal{R}}\right)$ is invertible, then the solution $(z_1, z_2)^{\mathsf{T}} = \left(z_1^{(1)}, z_2^{(1)}\right)^{\mathsf{T}}$ is given by

$$\left(z_1^{(1)}, z_2^{(1)}\right)^{\mathsf{T}} = \Bigg( \frac{\left((1-d)h_1 + c\, h_2\right)\left(a_l + d + a_l\, d - b_l\, c + 1\right)}{(a_r\, d - b_r\, c)(a_l\, d - b_l\, c) - c(b_l + b_r) - d^2 - a_l\, a_r + 1},$$

$$\frac{h_2\left(1 + d - a_l\, a_r - b_r\, c - a_l\, a_r\, d + a_r\, b_l\, c\right) + h_1\left(b_l + a_r\, b_l + b_r\, d + a_l\, b_r\, d - b_l\, b_r\, c\right)}{(a_r\, d - b_r\, c)(a_l\, d - b_l\, c) - c(b_l + b_r) - d^2 - a_l\, a_r + 1} \Bigg). \tag{36}$$

Also $T_{\mathcal{R}}\left(z_1^{(1)}, z_2^{(1)}\right) = \left(z_1^{(2)}, z_2^{(2)}\right)^{\mathsf{T}}$ yields

$$\left(z_1^{(2)}, z_2^{(2)}\right)^{\mathsf{T}} = \Bigg( \frac{\left((1-d)h_1 + c\, h_2\right)\left(a_r + d + a_r\, d - b_r\, c + 1\right)}{(a_r\, d - b_r\, c)(a_l\, d - b_l\, c) - c(b_l + b_r) - d^2 - a_l\, a_r + 1},$$

$$\frac{h_2\left(1 + d - a_l\, a_r - b_l\, c - a_l\, a_r\, d + a_l\, b_r\, c\right) + h_1\left(b_r + a_l\, b_r + b_l\, d + a_r\, b_l\, d - b_l\, b_r\, c\right)}{(a_r\, d - b_r\, c)(a_l\, d - b_l\, c) - c(b_l + b_r) - d^2 - a_l\, a_r + 1} \Bigg). \tag{37}$$

Hence, the existence region of the 2-cycle $\mathcal{O}_{\mathcal{R}\mathcal{L}}$ is

$$E_{\mathcal{O}_{\mathcal{R}\mathcal{L}}} = \Bigg\{ (h_1, h_2, a_l, b_l, c, d) \,\Big|\, \frac{\left((1-d)h_1 + c\, h_2\right)\left(a_l + d + a_l\, d - b_l\, c + 1\right)}{(a_r\, d - b_r\, c)(a_l\, d - b_l\, c) - c(b_l + b_r) - d^2 - a_l\, a_r + 1} > 0,$$

$$\frac{\big((1-d)h_1 + c\,h_2\big)\big(a_r + d + a_r\,d - b_r\,c + 1\big)}{(a_r\,d - b_r\,c)(a_l\,d - b_l\,c) - c(b_l + b_r) - d^2 - a_l\,a_r + 1} < 0 \bigg\}. \tag{38}$$

The characteristic polynomial of $J_{\mathcal{RL}} = J_{\mathcal{L}}\,J_{\mathcal{R}} = \begin{pmatrix} a_l\,a_r + b_r\,c & a_l\,c + c\,d \\ a_r\,b_l + b_r\,d & d^2 + b_l\,c \end{pmatrix}$ is given by

$$\mathcal{P}_{\mathcal{O}_{\mathcal{RL}}}(\lambda) = \lambda^2 - (d^2 + a_l\,a_r + b_l\,c + b_r\,c)\lambda + (a_r\,d - b_r\,c)(a_l\,d - b_l\,c), \tag{39}$$

and

$$\mathcal{D}_{\mathcal{RL}} = (a_r\,d - b_r\,c)(a_l\,d - b_l\,c),$$

$$\mathcal{P}_{\mathcal{O}_{\mathcal{RL}}}(1) = (a_r\,d - b_r\,c)(a_l\,d - b_l\,c) - c(b_l + b_r) - d^2 - a_l\,a_r + 1,$$

$$\mathcal{P}_{\mathcal{O}_{\mathcal{RL}}}(-1) = (a_r\,d - b_r\,c)(a_l\,d - b_l\,c) + c(b_l + b_r) + d^2 + a_l\,a_r + 1,$$

$$\lambda_{1,2}(\mathcal{O}_{\mathcal{RL}}) = \frac{a_l\,a_r + c(b_l + b_r) + d^2}{2}$$
$$\pm \frac{\sqrt{(a_l\,a_r + b_l\,c)^2 + (b_r\,c + d^2)^2 + 2(a_l\,a_r - b_l\,c)(b_r\,c - d^2) + 4\,c\,d(a_l\,b_r + a_r\,b_l)}}{2}. \tag{40}$$

Thus, the stability region of $\mathcal{O}_{\mathcal{RL}}$ is

$$\mathcal{S}_{\mathcal{RL}} = \bigg\{ (h_1, h_2, a_{l/r}, b_{l/r}, c, d) \in E_{\mathcal{O}_{\mathcal{RL}}} \,\big|\, -1 < (a_r\,d - b_r\,c)(a_l\,d - b_l\,c) < 1,$$

$$- (a_r\,d - b_r\,c)(a_l\,d - b_l\,c) - 1 < c(b_l + b_r) + d^2 + a_l\,a_r < (a_r\,d - b_r\,c)(a_l\,d - b_l\,c) + 1 \bigg\}. \tag{41}$$

In addition, for $\big((1-d)h_1 + c\,h_2\big)\big(a_l + d + a_l\,d - b_l\,c + 1\big) \neq 0$, the set

$$\tau_{\mathcal{RL}} = \bigg\{ (h_1, h_2, a_l, a_r, b_l, b_r, c, d) \,\big|\, a_l\,a_r + b_l\,c + b_r\,c + d^2 - a_l\,a_r\,d^2 - b_l\,b_r\,c^2 + a_l\,b_r\,c\,d$$

$$+ a_r\,b_l\,c\,d - 1 = 0 \bigg\}, \tag{42}$$

is the DTB curve for the 2-cycle $\mathcal{O}_{\mathcal{RL}}$. As in this case, for the parameter values belonging to $\tau_{\mathcal{RL}}$, the points of the 2-cycle $\mathcal{O}_{\mathcal{RL}}$ tend to $\pm\infty$, and the corresponding eigenvalue tends to one. Moreover, for $(1-d)h_1 + c\,h_2 \neq 0$, the BCB curves of $\mathcal{O}_{\mathcal{RL}}$ can be computed as

$$\xi_{\mathcal{RL}}^1 = \bigg\{ (h_1, h_2, a_l, b_l, c, d) \,\big|\, a_l + d + a_l\,d - b_l\,c + 1 = 0 \bigg\},$$

$$\xi_{\mathcal{RL}}^2 = \bigg\{ (h_1, h_2, a_r, b_r, c, d) \,\big|\, a_r + d + a_r\,d - b_r\,c + 1 = 0 \bigg\}. \tag{43}$$

Note that here the condition $(1-d)h_1 + c\,h_2 \neq 0$ guarantees a regular BCB in the sense that only one periodic point of $\mathcal{O}_{\mathcal{RL}}$ collides with the switching boundary; for more details see [6]. Besides,

$$\mathcal{F}_{\mathcal{RL}} = \bigg\{ (h_1, h_2, a_l, a_r, b_l, b_r, c, d) \,\big|\, a_l a_r + b_l c + b_r c + d^2 + a_l\,a_r d^2 + b_l\,b_r c^2$$

$$- a_l\,b_r\,c\,d - a_r\,b_l c d + 1 = 0 \bigg\}, \tag{44}$$

is the DFB curve of the 2-cycle $\mathcal{O}_{\mathcal{RL}}$.

**Remark 5.** *One can see that for $(1-d)h_1 + c\,h_2 \neq 0$ the DFB curves of the fixed points $\mathcal{O}_{\mathcal{L}}$ and $\mathcal{O}_{\mathcal{R}}$ ($\mathcal{F}_{\mathcal{L}}$ and $\mathcal{F}_{\mathcal{R}}$) and the BCB boundaries of the 2-cycle $\mathcal{O}_{\mathcal{RL}}$ ($\xi_{\mathcal{RL}}^1$ and $\xi_{\mathcal{RL}}^2$) are the same. In this case, the DFB of the fixed points can lead to the (attracting) 2-cycle $\mathcal{O}_{\mathcal{RL}}$.*

### A.1.4  3-cycles of the map (11) and their bifurcations

Here, we investigate the existence, stability and bifurcation structure of maximal or basic 3-cycles. Note that for the continuous map (11), basic $n$-cycles $\mathcal{O}_{\mathcal{RL}^{n-1}}$ ($n \geq 3$) exist in pairs with their complementary cycles ($\mathcal{O}_{\mathcal{RL}^{n-2}\mathcal{R}}$), and they appear via BCBs such that one of them may be attracting and the other repelling [6, 20, 38]. In this case, a BCB of basic cycles demonstrates a non-smooth fold bifurcation which includes a stable basic orbit and an unstable nonbasic orbit [6, 5, 19]. Furthermore, the complementary orbits can have nonempty stability regions such that, similar to the basic orbits, they are bounded by curves of BCBs, DTBs and DFBs [6, 5].

**Basic 3-cycles $\mathcal{O}_{\mathcal{RL}^2}$ and their complementary cycles $\mathcal{O}_{\mathcal{R}^2\mathcal{L}}$.**   The basic 3-cycle $\mathcal{O}_{\mathcal{RL}^2}$ can be obtained from the equation $T_{\mathcal{L}} \circ T_{\mathcal{L}} \circ T_{\mathcal{R}}(z_1, z_2) = (z_1, z_2)^{\mathsf{T}}$ where

$$
T_{\mathcal{L}} \circ T_{\mathcal{L}} \circ T_{\mathcal{R}}(z_1, z_2) = \begin{pmatrix} a_r(a_l^2 + b_l\,c) + b_r(a_l\,c + c\,d) & c(a_l^2 + b_l\,c) + d(a_l\,c + c\,d) \\ b_r(d^2 + b_l\,c) + a_r(a_l\,b_l + b_l\,d) & d(d^2 + b_l\,c) + c(a_l\,b_l + b_l\,d) \end{pmatrix} \begin{pmatrix} z_1 \\ z_2 \end{pmatrix}
$$
$$
+ \begin{pmatrix} h_1\big(b_l\,c + a_l(a_l+1)+1\big) + h_2\big(a_l\,c + c(d+1)\big) \\ h_1\big(b_l\,d + b_l(a_l+1)\big) + h_2\big(b_l\,c + d(d+1)+1\big) \end{pmatrix}. \tag{45}
$$

If $\left(I - J_{\mathcal{L}}^2\,J_{\mathcal{R}}\right)$ is invertible, then the solution $(z_1, z_2)^{\mathsf{T}} = \big(z_1^{(1)}, z_2^{(1)}\big)^{\mathsf{T}}$ is

$$
(z_1^{(1)}, z_2^{(1)})^T = \left( \frac{\big((1-d)h_1 + c\,h_2\big)G_1}{G}, \frac{G_2}{G} \right)^{\mathsf{T}}, \tag{46}
$$

where

$$
G_1 = a_l^2 d^2 + a_l^2 d + a_l^2 - 2a_l\,b_l cd - a_l b_l c + a_l d^2 + a_l d + a_l + b_l^2 c^2 - b_l cd + b_l c + d^2 + d + 1,
$$
$$
G = -a_l^2\,a_r - d^3 - c\big(a_l\,b_l + a_l\,b_r + a_r\,b_l + d(2\,b_l + b_r)\big) + (a_r\,d - b_r\,c)(a_l\,d - b_l\,c)^2 + 1,
$$
$$
G_2 = h_2 + b_l h_1 + dh_2 + d^2 h_2 + a_l b_l h_1 + b_l ch_2 + b_l dh_1 - a_l^2 a_r h_2 + b_r d^2 h_1 - a_r b_l^2 ch_1
$$
$$
- a_l^2 a_r dh_2 + a_l b_r d^2 h_1 + b_l b_r c^2 h_2 - a_r b_l^2 c^2\,h_2 - a_l^2 a_r d^2 h_2 + a_l^2 b_r\,d^2 h_1 + b_l^2\,b_r\,c^2 h_1
$$
$$
+ a_l\,a_r\,b_l\,h_1 - a_l\,b_r\,c\,h_2 - a_r\,b_l\,c\,h_2 + a_r\,b_l\,d\,h_1 + b_l\,b_r\,ch_1 - b_r\,c\,dh_2 + a_l\,a_r\,b_l ch_2
$$
$$
+ a_l\,a_r\,b_l\,dh_1 - a_l\,b_r\,c\,d\,h_2 - b_l\,b_r\,c\,dh_1 + 2a_l\,a_r\,b_l\,c\,dh_2 - 2a_l\,b_l\,b_r\,c\,dh_1. \tag{47}
$$

Further

$$
T_{\mathcal{R}}(z_1^{(1)}, z_2^{(1)}) = \big(z_1^{(2)}, z_2^{(2)}\big)^{\mathsf{T}} = \left( \frac{\big((1-d)h_1 + c\,h_2\big)K_1}{G}, \frac{K_2}{G} \right)^{T},
$$
$$
T_{\mathcal{L}}(z_1^{(2)}, z_2^{(2)}) = \big(z_1^{(3)}, z_2^{(3)}\big)^{\mathsf{T}} = \left( \frac{\big((1-d)h_1 + c\,h_2\big)H_1}{G}, \frac{H_2}{G} \right)^{T}, \tag{48}
$$

where

$$
K_1 = a_r + d + a_l a_r + b_l c + a_r d + a_r d^2 + d^2 + a_l a_r d - a_l b_r c - b_r cd + a_l a_r d^2 + b_l b_r c^2
$$
$$
- a_l b_r cd - a_r b_l cd + 1,
$$
$$
K_2 = h_2 + b_r h_1 + dh_2 + d^2 h_2 + a_l b_r h_1 + b_r ch_2 + b_l dh_1 - a_l^2 a_r h_2 + a_l^2 b_r h_1 + b_l d^2 h_1
$$
$$
+ a_l^2 b_r ch_2 + a_r b_l d^2 h_1 + b_l b_r c^2 h_2 - a_l^2 a_r d^2 h_2 + b_l^2 b_r c^2 h_1 - a_l b_l ch_2 - a_r b_l ch_2 + a_l b_l dh_1
$$
$$
+ b_l\,b_r\,ch_1 - b_l cd\,h_2 + a_l a_r b_l dh_1 - a_l\,b_l b_r ch_1 - a_r\,b_l\,c\,dh_2 - b_l b_r c\,dh_1 + a_l\,a_r b_l d^2 h_1
$$
$$
- a_l\,b_l b_r c^2 h_2 - a_r b_l^2 cdh_1 + a_l^2 b_r cdh_2 + a_l\,a_r\,b_l\,c\,d\,h_2 - a_l\,b_l\,b_r\,c\,d\,h_1 - a_l^2 a_r dh_2,
$$
$$
H_1 = a_l + d + a_l a_r + a_l d + b_r c + a_l d^2 + d^2 + a_l a_r d - a_r b_l c - b_l cd + a_l a_r d^2 + b_l b_r c^2
$$
$$
- a_l b_r cd - a_r b_l cd + 1,
$$
$$
H_2 = h_2 + b_l h_1 + dh_2 + d^2 h_2 + b_l^2 c^2 h_2 + a_r b_l h_1 + b_l ch_2 + b_r dh_1 - a_l^2 a_r h_2 + b_l^2 ch_1
$$

$$+ \; b_l d^2 h_1 - a_l^2 a_r dh_2 - b_l cdh_2 + a_l b_l d^2 h_1 + a_l^2 b_r dh_1 - b_l^2 cdh_1 - a_l^2 a_r d^2 h_2 + b_l^2 b_r c^2 h_1$$

$$+ \; a_l a_r b_l h_1 - a_l b_l ch_2 - a_l b_r ch_2 + a_l b_r dh_1 + a_l a_r b_l ch_2 - a_l b_l b_r ch_1 - a_l b_l cd\, h_2$$

$$+ \; a_l a_r b_l d^2 h_1 - a_l b_l b_r c^2 h_2 - a_r b_l^2 cdh_1 + a_l^2 b_r cdh_2 + a_l a_r b_l cdh_2 - a_l b_l b_r cdh_1. \tag{49}$$

Therefore, the existence region of the 3-cycle $\mathcal{O}_{\mathcal{RL}^2}$ is given by

$$E_{\mathcal{O}_{\mathcal{RL}^2}} = \left\{ (h_1, h_2, a_l, b_l, c, d) \, \middle| \, \frac{((1-d)h_1 + c\, h_2)G_1}{G} > 0, \quad \frac{((1-d)h_1 + c\, h_2)K_1}{G} < 0, \right.$$
$$\left. \frac{((1-d)h_1 + c\, h_2)H_1}{G} < 0 \right\}, \tag{50}$$

where $G, G_1, K_1$ and $H_1$ are defined in (47) and (49). On the other hand, the characteristic polynomial of

$$J_{\mathcal{RL}^2} = J_{\mathcal{L}}^2 \, J_{\mathcal{R}} = \begin{pmatrix} a_r(a_l^2 + b_l\, c) + b_r(a_l\, c + c\, d) & c(a_l^2 + b_l\, c) + d(a_l\, c + c\, d) \\ b_r(d^2 + b_l\, c) + a_r(a_l\, b_l + b_l\, d) & d(d^2 + b_l\, c) + c(a_l\, b_l + b_l\, d) \end{pmatrix},$$

is

$$\mathcal{P}_{\mathcal{O}_{\mathcal{RL}^2}}(\lambda) = \lambda^2 - \left( a_l^2 a_r + d^3 + c(a_l b_l + a_l b_r + a_r b_l + d(2b_l + b_r)) \right)\lambda + (a_r d - b_r c)(a_l d - b_l c)^2. \tag{51}$$

According to

$$\mathcal{D}_{\mathcal{RL}^2} = (a_l\, d - b_l\, c)^2 (a_r\, d - b_r\, c),$$

$$\mathcal{P}_{\mathcal{O}_{\mathcal{RL}^2}}(1) = -a_l^2 a_r - d^3 - c(a_l b_l + a_l b_r + a_r b_l + d(2b_l + b_r))$$

$$\mathcal{P}_{\mathcal{O}_{\mathcal{RL}^2}}(-1) = a_l^2 a_r + d^3 + c(a_l b_l + a_l b_r + a_r b_l + d(2b_l + b_r)) + (a_r d - b_r c)(a_l d - b_l c)^2 + 1, \tag{52}$$

the stability region of the 3-cycle $\mathcal{O}_{\mathcal{RL}^2}$ is given by

$$\mathcal{S}_{\mathcal{RL}^2} = \left\{ (h_1, h_2, a_{l/r}, b_{l/r}, c, d) \in E_{\mathcal{O}_{\mathcal{RL}^2}} \, \middle| \, -1 < (a_l\, d - b_l\, c)^2 (a_r\, d - b_r\, c) < 1, \right.$$
$$- (a_l d - b_l c)^2 (a_r\, d - b_r c) - 1 < a_l^2 a_r + d^3 + c(a_l b_l + a_l b_r + a_r b_l + d(2b_l + b_r))$$
$$\left. < (a_l\, d - b_l\, c)^2 (a_r\, d - b_r\, c) + 1 \right\}. \tag{53}$$

Furthermore, for $(1-d)h_1 + c\, h_2 \neq 0$ and $G_1, G_2, K_1, K_2, H_1, H_2 \neq 0$, the DTB curve for the 3-cycle $\mathcal{O}_{\mathcal{RL}^2}$ is

$$\tau_{\mathcal{RL}^2} = \left\{ (h_1, h_2, a_l, a_r, b_l, b_r, c, d) \, \middle| \, b_r\, a_l^2 cd^2 - a_r\, a_l^2\, d^3 + a_r a_l^2 - 2\, b_r\, a_l b_l c^2 d + 2a_r\, a_l\, b_l\, c\, d^2 \right.$$
$$\left. + \; a_l\, b_l\, c + b_r\, a_l\, c + b_r\, b_l^2 c^3 - a_r\, b_l^2 c^2 d + 2\, b_l cd + a_r b_l\, c + b_r\, c\, d + d^3 - 1 = 0 \right\}. \tag{54}$$

For $(1-d)h_1 + c\, h_2 \neq 0$

$$\xi_{\mathcal{RL}^2}^1 = \left\{ (h_1, h_2, a_r, b_r, c, d) \, \middle| \, K_1 = a_r + d + a_l a_r + b_l\, c + a_r d + a_r d^2 + d^2 + a_l a_r d \right.$$
$$\left. - \; a_l\, b_r\, c - b_r\, c\, d + a_l\, a_r\, d^2 + b_l\, b_r\, c^2 - a_l\, b_r\, c\, d - a_r\, b_l\, c\, d + 1 = 0 \right\},$$

$$\xi_{\mathcal{RL}^2}^2 = \left\{ (h_1, h_2, a_r, b_r, c, d) \, \middle| \, H_1 = a_l + d + a_l\, a_r + a_l\, d + b_r\, c + a_l\, d^2 + d^2 + a_l\, a_r\, d \right.$$

$$- a_r\,b_l\,c - b_l\,c\,d + a_l\,a_r\,d^2 + b_l\,b_r\,c^2 - a_l\,b_r\,c\,d - a_r\,b_l\,c\,d + 1 = 0\Big\}, \qquad (55)$$

are (regular) BCB curves of $\mathcal{O}_{\mathcal{RL}^2}$. Furthermore, the set

$$\mathcal{F}_{\mathcal{RL}^2} = \Big\{(h_1, h_2, a_l, a_r, b_l, b_r, c, d)\big| - b_r\,a_l^2 cd^2 + a_r a_l^2 d^3 + a_r a_l^2 + 2b_r a_l b_l c^2 d - 2a_r a_l b_l cd^2$$
$$+ a_l b_l c + b_r a_l c - b_r b_l^2 c^3 + a_r\,b_l^2\,c^2 d + 2b_l cd + a_r b_l c + b_r\,c\,d + d^3 + 1 = 0\Big\},$$
$$(56)$$

is the DFB curve of $\mathcal{O}_{\mathcal{RL}^2}$. As noted, the basic 3-cycle $\mathcal{O}_{\mathcal{RL}^2}$ exists in a pair with its complementary cycle $\mathcal{O}_{\mathcal{R}^2\mathcal{L}}$. Moreover, the existence region of $\mathcal{O}_{\mathcal{R}^2\mathcal{L}}$ can easily be found by interchanging the letters $\mathcal{L}$ and $\mathcal{R}$ in all notations of the equations (45)-(49) and considering

$$z_1^{(1)} < 0, \quad z_1^{(2)} > 0, \quad z_1^{(3)} > 0. \qquad (57)$$

Further, the stability region of the 3-cycle $\mathcal{O}_{\mathcal{R}^2\mathcal{L}}$ for the parameter values satisfying (57) is given by

$$\mathcal{S}_{\mathcal{R}^2\mathcal{L}} = \Big\{(h_1, h_2, a_{l/r}, b_{l/r}, c, d)\,\big|\, -1 < (a_r\,d - b_r\,c)^2(a_l\,d - b_l\,c) < 1,$$
$$- (a_r d - b_r c)^2(a_l d - b_l c) - 1 < a_r^2 a_l + d^3 + c\big(a_r b_r + a_r b_l + a_l b_r + d(2b_r + b_l)\big)$$
$$< (a_r\,d - b_r\,c)^2(a_l\,d - b_l\,c) + 1\Big\}. \qquad (58)$$

Notice that whenever the stable 3-cycle $\mathcal{O}_{\mathcal{RL}^2}$ exists, its complementary orbit $\mathcal{O}_{\mathcal{R}^2\mathcal{L}}$ also exists, but it is unstable. Furthermore, for $(1 - d)h_1 + c\,h_2 \neq 0$ both the 3-cycles $\mathcal{O}_{\mathcal{RL}^2}$ and $\mathcal{O}_{\mathcal{R}^2\mathcal{L}}$ appear at the same BCB curves (55). On the other hand, the DTB and DFB curves of the 3-cycle $\mathcal{O}_{\mathcal{R}^2\mathcal{L}}$ are given by

$$\tau_{\mathcal{R}^2\mathcal{L}} = \Big\{(h_1, h_2, a_l, a_r, b_l, b_r, c, d)\,\big|\, b_l a_r^2 cd^2 - a_l a_r^2 d^3 + a_l a_r^2 - 2b_l a_r b_r c^2 d + 2a_l a_r b_r cd^2$$
$$+ a_r\,b_r\,c + b_l a_r c + b_l\,b_r^2 c^3 - a_l\,b_r^2\,c^2 d + 2b_r cd + a_l b_r c + b_l\,c\,d + d^3 - 1 = 0\Big\},$$
$$(59)$$

and

$$\mathcal{F}_{\mathcal{R}^2\mathcal{L}} = \Big\{(h_1, h_2, a_l, a_r, b_l, b_r, c, d)\,\big|\, -b_l a_r^2 cd^2 + a_l a_r^2 d^3 + a_l a_r^2 + 2b_l a_r b_r c^2 d - 2a_l a_r b_r cd^2$$
$$+ a_r b_r c + b_l\,a_r c - b_l\,b_r^2 c^3 + a_l\,b_r^2 c^2\,d + 2b_r cd + a_l\,b_r\,c + b_l\,c\,d + d^3 + 1 = 0\Big\},$$
$$(60)$$

respectively.

### A.1.5 Multiple attractor bifurcations (MABs) of the map (11)

To detect multiple attractor bifurcations for the map (11), a straightforward way is to determine the overlapping stability regions of different periodic orbits. For instance, as shown in Fig. 1A, for $c = 0.8$, $d = 0.2$, $b_l = -0.4$, $b_r = 0.5$, two stability regions $\mathcal{S}_{\mathcal{RL}}$ and $\mathcal{S}_{\mathcal{RL}^2}$ overlap in the $(a_l, a_r)$-parameter plane (or in the $(a_{11}, a_{11} + w_{11})$-parameter space for the 2d PLRNN (7)). Their overlapping region, displayed in yellow, reveals the structure of the $(a_l, a_r)$-parameter plane. This helps us to find various MABs. Assuming $h_2 = 0$ and varying $h_1$ from a negative value to a positive one, an MAB of the form

$$\mathcal{O}_{\mathcal{L}}^s \overset{h_1}{\longleftrightarrow} \mathcal{O}_{\mathcal{RL}}^s + \mathcal{O}_{\mathcal{RL}^2}^s, \qquad (61)$$

occurs in the overlapping region. An example of this kind of bifurcation is illustrated in Fig. S3A.

Moreover, Fig. S4 indicates, for $c = 0.9$, $d = 0.3$, $b_l = -0.6$, $b_r = -1.54$, there are nonempty overlapping regions $\mathcal{S}_{\mathcal{RL}} \cap \mathcal{S}_{\mathcal{RL}^2}$ and $\mathcal{S}_{\mathcal{RL}^2} \cap \mathcal{S}_{\mathcal{R}}$. This leads to the occurrence of two different MABs given by (61) and

$$\mathcal{O}_\mathcal{L}^s \overset{h_1}{\longleftrightarrow} \mathcal{O}_\mathcal{R}^s + \mathcal{O}_{\mathcal{RL}^2}, \tag{62}$$

for $h_2 = 0$ and $h_1$ changing from negative to positive values. Both of these bifurcations are shown in Fig. S3A and Fig. S3B, associated with the points $P_1$ and $P_2$ in Fig. S4. Note that in Fig. S4, all the points $P_2$, $P_3$ and $P_4$ belong to the overlapping region $\mathcal{S}_{\mathcal{RL}^2} \cap \mathcal{S}_{\mathcal{R}}$ (in sky blue). These points are related to the parameter values $c = 0.9$, $d = 0.3$, $b_l = -0.6$, $b_r = -1.54$, $a_r = -1.8$, $h_2 = 0$, and they only differ in the parameter $a_l$. In this case, one can see that fixing all parameters and changing only the parameter $a_l$, from $P_2$ to $P_4$, the basins of attraction change. The corresponding basins of attraction for these three points are demonstrated in Fig. S3B (right) and Fig. S5 for $h_1 = 0.5$ (after the bifurcation).

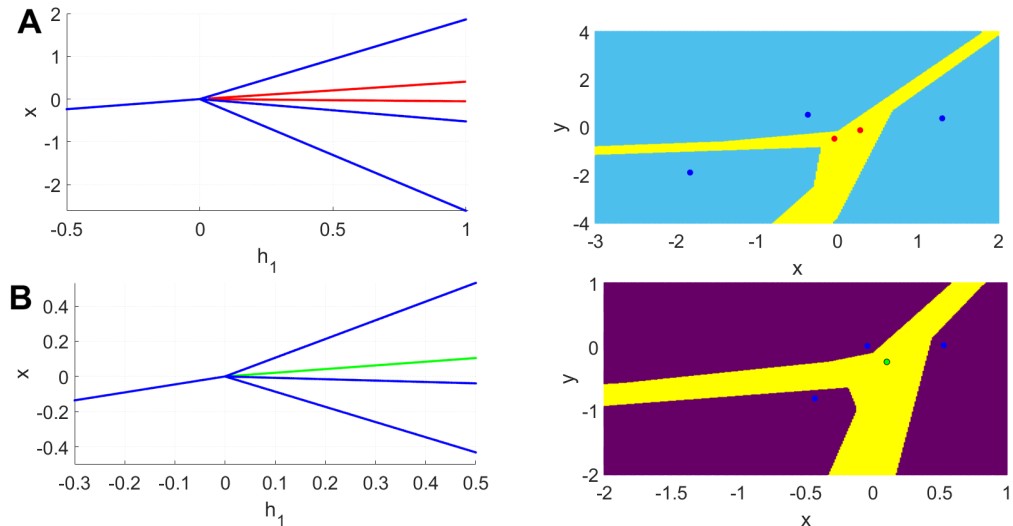

Figure S3: MAB at $c = 0.9$, $d = 0.3$, $b_l = -0.6$, $b_r = -1.54$, $h_2 = 0$. A) Left: Bifurcation diagram for $a_l = -0.44$ and $a_r = -1.8$ corresponding to the point $P_1$ in Fig. S4. Right: Multistability of the fixed point $\mathcal{O}_\mathcal{R}^s$ and the 3-cycle $\mathcal{O}_{\mathcal{RL}^2}^s$ after the bifurcation and their basins of attraction at $h_1 = 0.5$. B) Left: Bifurcation diagram for $a_l = -0.35$ and $a_r = -2.2$ corresponding to the point $P_2$ in Fig. S4. Right: Multistability of the 2-cycle $\mathcal{O}_{\mathcal{RL}}^s$ and the 3-cycle $\mathcal{O}_{\mathcal{RL}^2}^s$ after the bifurcation and their basins of attraction at $h_1 = 0.7$.

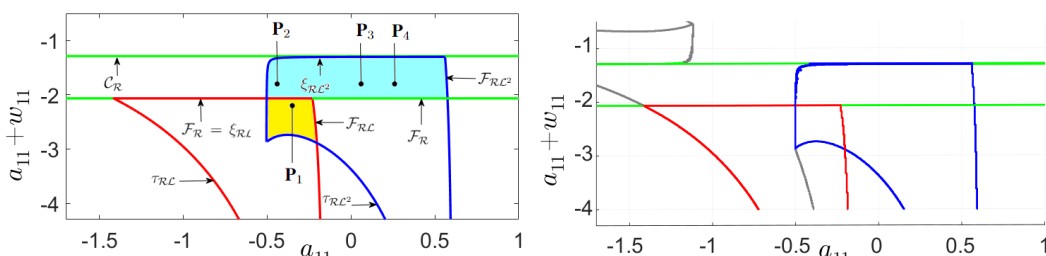

Figure S4: Analytically calculated stability regions for a different parameter setting than used in Fig. 1. Left: Analytically calculated stability regions $\mathcal{S}_\mathcal{R}$, $\mathcal{S}_{\mathcal{RL}}$ and $\mathcal{S}_{\mathcal{RL}^2}$, shown in green, red and blue, respectively, in the $(a_{11}, a_{11} + w_{11})$-parameter plane for $a_{22} = 0.3$, $w_{21} = -1.54$. The overlapping regions $\mathcal{S}_{\mathcal{RL}} \cap \mathcal{S}_{\mathcal{RL}^2}$ and $\mathcal{S}_{\mathcal{RL}^2} \cap \mathcal{S}_\mathcal{R}$, representing multi-stable regimes, are given in yellow and sky blue. Right: Bifurcation curves for the same parameter settings as determined by SCYFI. Note that SCYFI identifies additional structure (regions demarcated by gray curves) not included in our analytical derivations.

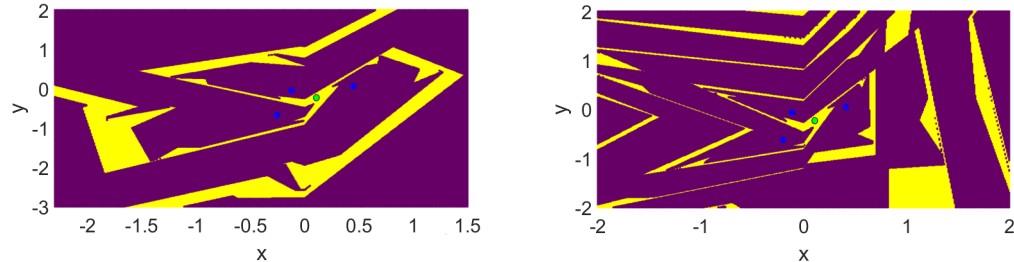

Figure S5: Multistability of the fixed point $\mathcal{O}^s_{\mathcal{R}}$ and the 3-cycle $\mathcal{O}^s_{\mathcal{R}\mathcal{L}^2}$ at $c = 0.9$, $d = 0.3$, $b_l = -0.6$, $b_r = -1.54$, $a_r = -1.8$, $h_2 = 0$ after the bifurcation and their basins of attraction at $h_1 = 0.5$. Left: $a_l = 0.06$ (point $P_3$ in Fig. S4); right: $a_l = 0.26$ (point $P_4$ in Fig. S4).

### A.2  Proofs of theorems

#### A.2.1  Proof of theorem 1

*Proof.* Let $\mathcal{L}(\boldsymbol{\theta})$ be some loss function employed for PLRNN training that decomposes in time as $\mathcal{L} = \sum_{t=1}^{T} \mathcal{L}_t$. Then

$$\frac{\partial \mathcal{L}}{\partial \theta} = \sum_{t=1}^{T} \frac{\partial \mathcal{L}_t}{\partial \theta},$$

$$\frac{\partial \mathcal{L}_t}{\partial \theta} = \frac{\partial \mathcal{L}_t}{\partial \boldsymbol{z}_t} \frac{\partial \boldsymbol{z}_t}{\partial \theta}. \tag{63}$$

Denoting the Jacobian of system (2) at time $t$ by

$$\boldsymbol{J}_t := \frac{\partial F_{\boldsymbol{\theta}}(\boldsymbol{z}_{t-1})}{\partial \boldsymbol{z}_{t-1}} = \frac{\partial \boldsymbol{z}_t}{\partial \boldsymbol{z}_{t-1}}, \tag{64}$$

we have

$$\frac{\partial \boldsymbol{z}_t}{\partial \theta} = \frac{\partial \boldsymbol{z}_t}{\partial \boldsymbol{z}_{t-1}} \frac{\partial \boldsymbol{z}_{t-1}}{\partial \theta} + \frac{\partial^+ \boldsymbol{z}_t}{\partial \theta} = \boldsymbol{J}_t \frac{\partial \boldsymbol{z}_{t-1}}{\partial \theta} + \frac{\partial^+ \boldsymbol{z}_t}{\partial \theta}, \tag{65}$$

where $\partial^+$ denotes the immediate partial derivative (see [48] for more details). Assume that $\Gamma_k$ is a $k$-cycle ($k \geq 1$) of (2). Thus, $\Gamma_k$ is a set of temporally successive periodic points

$$P_k := \{\boldsymbol{z}_{t*k}, \boldsymbol{z}_{t*k-1}, \cdots, \boldsymbol{z}_{t*k-(k-1)}\} = \{\boldsymbol{z}_{t*k}, F(\boldsymbol{z}_{t*k}), \ldots, F^{k-1}_{\boldsymbol{\theta}}(\boldsymbol{z}_{t*k})\}, \tag{66}$$

such that all of them are fixed points of

$$\boldsymbol{z}_{t+k} = F^k_{\boldsymbol{\theta}}(\boldsymbol{z}_t) = F_{\boldsymbol{\theta}}(F_{\boldsymbol{\theta}}(F_{\boldsymbol{\theta}}(...F_{\boldsymbol{\theta}}(\boldsymbol{z}_t)...))), \tag{67}$$

and $k$ is the smallest such positive integer (for $k = 1$, $\Gamma_1$ is a fixed point of $F_{\boldsymbol{\theta}}$). Similar to (65), the tangent vector $\frac{\partial \boldsymbol{z}_{t+k}}{\partial \theta}$ can be computed as

$$\frac{\partial \boldsymbol{z}_{t+k}}{\partial \theta} = \frac{\partial \boldsymbol{z}_{t+k}}{\partial \boldsymbol{z}_t} \frac{\partial \boldsymbol{z}_t}{\partial \theta} + \frac{\partial^+ \boldsymbol{z}_{t+k}}{\partial \theta} = \prod_{r=0}^{k-1} \boldsymbol{J}_{t+k-r} \frac{\partial \boldsymbol{z}_t}{\partial \theta} + \frac{\partial^+ \boldsymbol{z}_{t+k}}{\partial \theta}. \tag{68}$$

Thus, for $\boldsymbol{z}_{t*k} = F^k_{\boldsymbol{\theta}}(\boldsymbol{z}_{t*k})$ we have

$$\frac{\partial \boldsymbol{z}_{t*k}}{\partial \theta} = \prod_{r=0}^{k-1} \boldsymbol{J}_{t*k-r} \frac{\partial \boldsymbol{z}_{t*k}}{\partial \theta} + \frac{\partial^+ \boldsymbol{z}_{t*k}}{\partial \theta}. \tag{69}$$

Accordingly

$$\frac{\partial \boldsymbol{z}_{t*k}}{\partial \theta} = \left(\boldsymbol{I} - \prod_{r=0}^{k-1} \boldsymbol{J}_{t*k-r}\right)^{-1} \frac{\partial^+ \boldsymbol{z}_{t*k}}{\partial \theta} = \frac{adj\left(\boldsymbol{I} - \prod_{r=0}^{k-1} \boldsymbol{J}_{t*k-r}\right)}{P_{\prod_{r=0}^{k-1} \boldsymbol{J}_{t*k-r}}(1)} \frac{\partial^+ \boldsymbol{z}_{t*k}}{\partial \theta}, \tag{70}$$

where $P_{\prod_{r=0}^{k-1} J_{t^{*k}-r}}(1) = \det\left(I - \prod_{r=0}^{k-1} J_{t^{*k}-r}\right)$. Moreover, from (63) and (70) we have

$$\left\|\frac{\partial \mathcal{L}_t}{\partial \theta}\right\| = \frac{1}{P_{\prod_{r=0}^{k-1} J_{t^{*k}-r}}(1)} \left\|\frac{\partial \mathcal{L}_t}{\partial z_{t^{*k}}} \, adj\left(I - \prod_{r=0}^{k-1} J_{t^{*k}-r}\right) \frac{\partial^+ z_{t^{*k}}}{\partial \theta}\right\|. \tag{71}$$

Now, suppose that $\Gamma_k$ undergoes a DTB, such that the fixed or cyclic points given by (66) tend to infinity and one of their eigenvalues tends to 1 for some parameter value $\theta = \theta_0$. This implies $P_{\prod_{r=0}^{k-1} J_{t^{*k}-r}}(1)$ becomes zero at $\theta = \theta_0$ and so, due to (70), $\left\|\frac{\partial z_{t^{*k}}}{\partial \theta}\right\|$ goes to infinity. Therefore the norm of the loss gradient, $\left\|\frac{\partial \mathcal{L}_t}{\partial \theta}\right\|$, tends to infinity at $\theta = \theta_0$ which results in a abrupt jump in the loss function.

Let $\{z_{t_1}, z_{t_2}, z_{t_3}, \ldots\}$ be an orbit which converges to $\Gamma_k$, i.e.

$$\lim_{n \to \infty} d(z_{t_n}, \Gamma_k) = 0. \tag{72}$$

Then there exists a neighborhood $U$ of $\Gamma_k$ and $k$ sub-sequences $\{z_{t_{km}}\}_{m=1}^{\infty}, \{z_{t_{km+1}}\}_{m=1}^{\infty}, \cdots,$ $\{z_{t_{km+(k-1)}}\}_{m=1}^{\infty}$ of the sequence $\{z_{t_n}\}_{n=1}^{\infty}$ such that all these sub-sequences belong to $U$ and

a) $z_{t_{km+s}} = F^k(z_{t_{k(m-1)+s}})$, $s = 0, 1, 2, \cdots, k-1$,

b) $\lim_{m \to \infty} z_{t_{km+s}} = z_{t^{*k}-s}$, $s = 0, 1, 2, \cdots, k-1$,

c) for every $z_{t_n} \in U$ there is some $s \in \{0, 1, 2, \cdots, k-1\}$ such that $z_{t_n} \in \{z_{t_{km+s}}\}_{m=1}^{\infty}$.

This implies for every $z_{t_n} \in U$ with $z_{t_n} \in \{z_{t_{km+s}}\}_{m=1}^{\infty}$, there exists some $\tilde{n} \in \mathbb{N}$ such that $z_{t_n} = z_{t_{k\tilde{n}+s}}$ and $\lim_{\tilde{n} \to \infty} z_{t_{k\tilde{n}+s}} = z_{t^{*k}-s}$. Consequently, there exists some $\tilde{N} \in \mathbb{N}$ such that for every $\tilde{n} \geq \tilde{N}$ both $z_{t_{k\tilde{n}+s}}$ and $z_{t^{*k}-s}$ belong to the same sub-region and so the matrices $W_{\Omega(t_{k\tilde{n}+s})}$ and $W_{\Omega(t^{*k}-s)}$ ($s \in \{0, 1, 2, \cdots, k-1\}$) are identical. Without loss of generality, let $s = 0$. Since $z_{t_{k(\tilde{n}+1)}} = F^k(z_{t_{k\tilde{n}}})$, so

$$\frac{\partial z_{t_{k(\tilde{n}+1)}}}{\partial \theta} = \prod_{r=0}^{k-1} J_{t^{*k}-r} \frac{\partial z_{t_{k\tilde{n}}}}{\partial \theta} + \frac{\partial^+ z_{t_{k(\tilde{n}+1)}}}{\partial \theta}. \tag{73}$$

On the other hand, $\lim_{\tilde{n} \to \infty} \frac{\partial z_{t_{k(\tilde{n}+1)}}}{\partial \theta} = \lim_{\tilde{n} \to \infty} \frac{\partial z_{t_{k\tilde{n}}}}{\partial \theta}$, which results in

$$lim_{\tilde{n} \to \infty} \frac{\partial z_{t_{k(\tilde{n}+1)}}}{\partial \theta} = \left(I - \prod_{r=0}^{k-1} J_{t^{*k}-r}\right)^{-1} lim_{\tilde{n} \to \infty} \frac{\partial^+ z_{t_{k(\tilde{n}+1)}}}{\partial \theta}$$

$$= \frac{adj\left(I - \prod_{r=0}^{k-1} J_{t^{*k}-r}\right)}{P_{\prod_{r=0}^{k-1} J_{t^{*k}-r}}(1)} lim_{\tilde{n} \to \infty} \frac{\partial^+ z_{t_{k(\tilde{n}+1)}}}{\partial \theta}. \tag{74}$$

This means as $\bar{n} \to \infty$, for any orbit converging to $\Gamma_k$ the norm of the loss gradient tends to infinity at $\theta = \theta_0$ which completes the proof. $\square$

### A.2.2 Proof of theorem 2

*Proof.* Let $\Gamma_k$ be a $k$-cycle ($k \geq 1$) of (2) defined by periodic points (66). Suppose further that $\Gamma_k$ undergoes a BCB for some parameter value $\theta = \theta_0$. Hence, one of its periodic points, e.g.

$z_{t^{*k}}$, collides with one border. Therefore, $z_{m\,t^{*k}} = 0$ for some $1 \leq m \leq M$ by the definition of discontinuity boundaries in [38, 39]. Similar to the proof of Theorem 1, for $\boldsymbol{\theta} = \boldsymbol{A}, \boldsymbol{W}$ we have

$$\frac{\partial \boldsymbol{z}_{t^{*k}-1}}{\partial \theta} = \frac{adj\left(\boldsymbol{I} - \prod_{r=0}^{k-1} \boldsymbol{J}_{t^{*k}-1-r}\right)}{P_{\prod_{r=0}^{k-1} \boldsymbol{J}_{t^{*k}-1-r}}(1)} \frac{\partial^+ \boldsymbol{z}_{t^{*k}-1}}{\partial \theta}, \tag{75}$$

in which

$$\frac{\partial^+ \boldsymbol{z}_{t^{*k}-1}}{\partial w_{nm}} = \boldsymbol{1}_{(n,m)} \, \boldsymbol{D}_{\Omega(t^{*k})} \, \boldsymbol{z}_{t^{*k}},$$

$$\frac{\partial^+ \boldsymbol{z}_{t^{*k}-1}}{\partial a_{mm}} = \boldsymbol{1}_{(m,m)} \, \boldsymbol{z}_{t^{*k}}, \tag{76}$$

where $\boldsymbol{1}_{(n,m)}$ is an $M \times M$ indicator matrix with a 1 for the $(n,m)$'th entry and 0 everywhere else. Since $z_{m\,t^{*k}} = 0$ at $\theta = \theta_0$, due to (76) $\frac{\partial^+ \boldsymbol{z}_{t^{*k}-1}}{\partial \theta}$ becomes the zero vector at $\theta = \theta_0$. Consequently, $\left\|\frac{\partial \boldsymbol{z}_{t^{*k}-1}}{\partial \theta}\right\|$ and so $\left\|\frac{\partial \mathcal{L}_t}{\partial \theta}\right\|$ vanishes at $\theta = \theta_0$. Now it can be shown that at $\theta = \theta_0$ the loss gradient goes to zero for every $\boldsymbol{z}_1 \in \mathcal{B}_{\Gamma_k}$ (the proof is similar to the last part of the proof of Theorem 1). $\qquad\square$

### A.2.3 Proof of corollary 1

*Proof.* For $M = 2$, let $h_1 \neq 0$. Then the DFB curves of the fixed point $\Gamma_1$ coincide with the BCB curves of the 2-cycle $\mathcal{O}_{\mathcal{RL}}$ of the form

$$\mathcal{F}_1 = \xi_{\mathcal{RL}}^1 = \{(h_1, h_2, a_{11}, a_{22}) | 1 + a_{11} + a_{22} + a_{11} a_{22} = 0\}, \tag{77}$$

or

$$\mathcal{F}_2 = \xi_{\mathcal{RL}}^2 = \{(h_1, h_2, a_{11}, w_{11}, w_{21}, a_{22}) \big| 1 + a_{11} + w_{11} + a_{22} + (a_{11} + w_{11}) a_{22} = 0\}. \tag{78}$$

For $M > 2$, assume that $\Gamma_1 = \{\boldsymbol{z}_1^*\}$ is a fixed point of the system, i.e.

$$\boldsymbol{z}_1^* = (\boldsymbol{I} - \boldsymbol{W}_{\Omega(t_1^*)})^{-1} \boldsymbol{h} = \frac{adj(\boldsymbol{I} - \boldsymbol{W}_{\Omega(t_1^*)})}{P_{\boldsymbol{I} - \boldsymbol{W}_{\Omega(t_1^*)}}(1)} \boldsymbol{h}, \tag{79}$$

where $P_{\boldsymbol{I} - \boldsymbol{W}_{\Omega(t_1^*)}}(1)$ is the characteristic polynomial of $\boldsymbol{I} - \boldsymbol{W}_{\Omega(t_1^*)}$ at 1. Let us denote the first row of the adjoint matrix of $\boldsymbol{I} - \boldsymbol{W}_{\Omega(t_1^*)}$ by $adj(\boldsymbol{I} - \boldsymbol{W}_{\Omega(t_1^*)})_1$. If $adj(\boldsymbol{I} - \boldsymbol{W}_{\Omega(t_1^*)})_1 \, \boldsymbol{h} \neq 0$, then we can analogously demonstrate that the DFB curves of the fixed point align with the BCB curves of the 2-cycles. This implies that, in accordance with Theorem 2, DFBs of fixed points will also lead to vanishing gradients in the loss function. $\qquad\square$

### A.2.4 Convergence of SCYFI

To ensure that SCYFI almost surely converges, we can simply choose the number of random initializations (i.e., $N_{out}$ in algorithm 1) large enough such that every linear subregion will have been sampled with probability almost 1. More precisely, drawing uniformly from the $2^M$ different $\boldsymbol{D}_\Omega$-matrices (linear subregions) for initialization, the probability that a particular subregion has not been drawn after $r$ repetitions is $p = (1 - \frac{1}{2^M})^r$. Hence, in order to ensure that all $2^M$ subregions have been visited with probability $1 - \epsilon$ we need $r \geq \lceil \frac{\ln(\epsilon)}{\ln(1 - \frac{1}{2^M})} \rceil$ iterations. Choosing $N_{out} = r$, we can thus ensure that SCYFI was initialized in each subregion with probability almost 1, and thus, in the limit, will have probed all subregions for dynamical objects. This argument extends to $k$-cycles by replacing $2^M$ by $2^{kM}$ above (strictly, a more precise bound for $k \geq 2$ is given by $2^{M(k-1)} \times (2^M - 1) = 2^{Mk} - 2^{M(k-1)}$, due to the fact that the PLRNN (2) is a linear map in each subregion and, hence, cannot have any $k$-cycles with all periodic points in only one subregion).

### A.2.5 Proof of theorem 3

*Proof.* We examine the convergence and scaling behavior of SCYFI for fixed points. A similar argument applies to $k$-cycles where $k > 1$.

Let $\boldsymbol{z}_1^*$ be a fixed point of the system, i.e.

$$\boldsymbol{z}_1^* = \left(\boldsymbol{I} - \boldsymbol{W}_{\Omega(t_1^*)}\right)^{-1}\boldsymbol{h}. \tag{80}$$

$\boldsymbol{z}_1^*$ is a true fixed point iff

$$(d_m(t_1^*) - a) \cdot z_{m,t_1^*} > 0, \qquad \forall\, m \in \{1, 2, \cdots, M\}, \tag{81}$$

where $\boldsymbol{D}_{\Omega(t_1^*)} = diag(d_1(t_1^*), d_2(t_1^*), \cdots, d_M(t_1^*))$ and $0 < a < 1$ is a positive real constant.

For examining SCYFI's efficiency, here we focus on two scenarios that impose specific constraints on parameters $\boldsymbol{\theta}$; other cases remain to be investigated.

**Case (I)** : Let $\boldsymbol{R}$ be a randomly generated matrix with uniformly distributed entries in the interval $[0, 1)$, and $\boldsymbol{h} \neq 0$ be a random vector with all its components being non-negative. For an arbitrary $\epsilon > 0$, we set

$$\boldsymbol{A} = \frac{1}{2 + \|\boldsymbol{R}\| + \epsilon}\, diag(\boldsymbol{R}),$$

$$\boldsymbol{W} = \frac{1}{2 + \|\boldsymbol{R}\| + \epsilon}\left(\boldsymbol{R} - diag(\boldsymbol{R})\right). \tag{82}$$

Then

$$\|\boldsymbol{A}\| = \frac{\|diag(\boldsymbol{R})\|}{2 + \|\boldsymbol{R}\| + \epsilon} < \frac{1}{2 + \|\boldsymbol{R}\| + \epsilon},$$

$$\|\boldsymbol{W}\| = \frac{\|\boldsymbol{R} - diag(\boldsymbol{R})\|}{2 + \|\boldsymbol{R}\| + \epsilon} \leq \frac{\|\boldsymbol{R}\| + \|diag(\boldsymbol{R})\|}{2 + \|\boldsymbol{R}\| + \epsilon} < \frac{1 + \|\boldsymbol{R}\|}{2 + \|\boldsymbol{R}\| + \epsilon}, \tag{83}$$

and so $\|\boldsymbol{A}\| + \|\boldsymbol{W}\| < 1$. Therefore

$$\forall t \quad \|\boldsymbol{W}_{\Omega(t)}\| = \|\boldsymbol{A} + \boldsymbol{W}\boldsymbol{D}_{\Omega(t)}\| \leq \|A\| + \|W\|\,\|\boldsymbol{D}_{\Omega(t)}\| \leq \|A\| + \|W\| < 1, \tag{84}$$

and so

$$\forall t \quad \rho(\boldsymbol{W}_{\Omega(t)}) \leq \|\boldsymbol{W}_{\Omega(t)}\| < 1. \tag{85}$$

In this case, for any $n \in \mathbb{N}$, we also have

$$\left\|\prod_{i=1}^n \boldsymbol{W}_{\Omega(t_i)}\right\| \leq \prod_{i=1}^n \|\boldsymbol{W}_{\Omega(t_i)}\| \leq \left(\|A\| + \|W\|\right)^n < 1. \tag{86}$$

This ensures the stability of all fixed points and $k$-cycles of the system.

According to (85), we have

$$\left(\boldsymbol{I} - \boldsymbol{W}_{\Omega(t_1^*)}\right)^{-1} = \sum_{n=0}^{\infty} \boldsymbol{W}_{\Omega(t_1^*)}^n = \boldsymbol{I} + \boldsymbol{W}_{\Omega(t_1^*)} + \boldsymbol{W}_{\Omega(t_1^*)}^2 + \cdots. \tag{87}$$

Hence, all the elements of $\left(\boldsymbol{I} - \boldsymbol{W}_{\Omega(t_1^*)}\right)^{-1}$ are positive, and so $z_{m,t_1^*} > 0$ for every $t_1^*$. This implies that all true and virtual fixed points exist within a singular sub-region. Additionally, only one fixed point is true, while all the other fixed points are virtual.

**Case (II)** : Let $\boldsymbol{h} = (h_1, h_2, \cdots, h_M)^{\mathsf{T}}$ be a random vector with all $h_m$ uniformly distributed in $(0, 1]$ and

$$\beta_{min} = \min\{h_m : h_m \in \boldsymbol{h},\, 1 \leq m \leq M\} > 0, \qquad 0 < \beta_{min} \leq 1,$$

$$\beta_{max} = \max\{h_m : h_m \in \boldsymbol{h},\, 1 \leq m \leq M\} > 0, \qquad 0 < \beta_{max} \leq 1. \tag{88}$$

Assume further that $\boldsymbol{R}_1$ is a randomly generated matrix with uniformly distributed entries in the interval $(-1, 0]$, and for $M \geq 2$

$$\boldsymbol{W} = \frac{\beta_{min}}{M + \|R_1\| + \epsilon} \Big( \boldsymbol{R}_1 - \mathrm{diag}(\boldsymbol{R}_1) \Big). \tag{89}$$

Consider

$$\alpha_{max} = \max \big\{ |w_{ij}| : w_{ij} \in \boldsymbol{W} \big\}, \qquad 0 \leq \alpha_{max} < \frac{\beta_{min}}{M + \|R\| + \epsilon} \tag{90}$$

and $S \subset \{1, 2, \cdots, M\} = I$ such that $K = 2^{M - card(S)} \ll 2^M$. Suppose that $\boldsymbol{R}_2 = diag(r_1, \cdots, r_M)$ is a randomly chosen diagonal matrix with $r_m$ uniformly distributed in $(-1, 1)$ for $m \in I \setminus S$, and the other elements ($m \in S$) uniformly distributed in $(r^* - 1, 0)$ where $r^* = \frac{(M-1)\alpha_{max}\beta_{max}}{\beta_{min}}$. Since

$$0 \leq \frac{(M-1)\alpha_{max}\beta_{max}}{\beta_{min}} < \frac{(M-1)\beta_{max}}{M + \|R\| + \epsilon} \leq \frac{(M-1)}{M + \|R\| + \epsilon} < \frac{(M-1)}{M} < 1, \tag{91}$$

so $-1 \leq r^* - 1 < 0$.

If

$$\boldsymbol{A} = \frac{1}{2 + \|R_1\| + \epsilon} \boldsymbol{R}_2, \tag{92}$$

then

$$\|\boldsymbol{A}\| = \frac{\|\boldsymbol{R}_2\|}{2 + \|\boldsymbol{R}_1\| + \epsilon} < \frac{1}{2 + \|\boldsymbol{R}_1\| + \epsilon},$$

$$\|\boldsymbol{W}\| = \frac{\beta_{min} \|\boldsymbol{R}_1 - \mathrm{diag}(\boldsymbol{R}_1)\|}{M + \|\boldsymbol{R}_1\| + \epsilon} \leq \frac{\|\boldsymbol{R}_1\| + \|\mathrm{diag}(\boldsymbol{R}_1)\|}{M + \|\boldsymbol{R}_1\| + \epsilon} < \frac{1 + \|\boldsymbol{R}_1\|}{2 + \|\boldsymbol{R}_1\| + \epsilon}, \tag{93}$$

which implies $\|\boldsymbol{A}\| + \|\boldsymbol{W}\| < 1$. We set $\epsilon > 0$ large enough to satisfy the condition

$$\big( I - \boldsymbol{W}_{\Omega(t_1^*)} \big)^{-1} = \sum_{n=0}^{\infty} \boldsymbol{W}_{\Omega(t_1^*)}^n \approx \boldsymbol{I} + \boldsymbol{W}_{\Omega(t_1^*)} \qquad \forall t_1^*. \tag{94}$$

On the other hand, for any $t$ we have

$$\boldsymbol{W}_{\Omega(t)} = \boldsymbol{A} + \boldsymbol{W}\boldsymbol{D}_{\Omega(t)} = \begin{pmatrix} a_{11} & w_{12}d_2(t) & w_{13}d_3(t) & \cdots & w_{1M}d_M(t) \\ w_{21}d_1(t) & a_{22} & w_{23}d_3(t) & \cdots & w_{2M}d_M(t) \\ w_{31}d_1(t) & w_{32}d_2(t) & a_{33} & \cdots & w_{3M}d_M(t) \\ \vdots & \vdots & \vdots & \ddots & \vdots \\ w_{M1}d_1(t) & w_{M2}d_2(t) & w_{M3}d_3(t) & \cdots & a_{MM} \end{pmatrix}. \tag{95}$$

Hence

$$z_{m,t_1^*} = (1 + a_{mm})\, h_m + \sum_{\substack{j=1 \\ j \neq m}}^{M} w_{mj}\, d_j(t_1^*) h_j = (1 + a_{mm})\, h_m - \sum_{\substack{j=1 \\ j \neq m}}^{M} |w_{mj}|\, d_j(t_1^*) h_j. \tag{96}$$

Since for every $t_1^*$

$$\sum_{\substack{j=1 \\ j \neq m}}^{M} |w_{mj}|\, d_j(t_1^*) h_j \leq \sum_{\substack{j=1 \\ j \neq m}}^{M} |w_{mj}|\, h_j, \tag{97}$$

so

$$z_{m,t_1^*} \geq (1 + a_{mm})\, h_m - \sum_{\substack{j=1 \\ j \neq m}}^{M} |w_{mj}|\, h_j \qquad \forall m \in I. \tag{98}$$

Moreover $a_{ss} \in (r^* - 1, 0)$, for every $s \in S$, and thus

$$a_{ss} + 1 > \frac{(M-1)\alpha_{max}\beta_{max}}{\beta_{min}} = \frac{\sum_{\substack{j=1 \\ j \neq s}}^{M} \alpha_{max}\beta_{max}}{\beta_{min}} \geq \frac{\sum_{\substack{j=1 \\ j \neq s}}^{M} |w_{sj}| h_j}{h_s}. \tag{99}$$

Therefore, due to (98) and (99), $z_{s,t_1^*} > 0$ for every $t_1^*$ and $s \in S$. This means that all true and virtual fixed points only exist within a relatively small number of sub-regions, denoted as $K = 2^{M-card(S)} \ll 2^M$. Given our specific initalization of $\boldsymbol{\theta}$, in both cases (I) and (II) there is a set of $K$ different sub-regions, each associated with a unique $\boldsymbol{D}_{\Omega(t)}$ matrix. We refer to the entire set of these matrices as

$$\mathcal{D}_K = \{\boldsymbol{D}_1, ..., \boldsymbol{D}_K\}. \tag{100}$$

SCYFI, by its definition, only moves within the sub-regions that have virtual and true fixed points, continuing until it discovers a true fixed point (or gets stuck in a virtual cycle). Thus, it can iterate between $J \leq K$ sub-regions

$$\mathcal{D}_J = \{\boldsymbol{D}_1, ..., \boldsymbol{D}_J\} \subseteq \mathcal{D}_K, \tag{101}$$

or within the set of virtual fixed points

$$\mathcal{Z}_L = \{\boldsymbol{z}_1, ..., \boldsymbol{z}_L\}. \tag{102}$$

In case (I), there is only one true fixed point. Since all virtual fixed points are located within the same single sub-region, SCYFI's initialization will naturally position it within the correct linear region, requiring no more than 1 iteration. Hence, it needs at most 2 iterations to find the true fixed point. Consequently, SCYFI's scaling is constant.

For case (II), if we suppose that SCYFI follows the virtual/true fixed point structure of the underlying system in these $K$ sub-regions, the necessity for the probability of discovering the fixed point to be close to 1, specifically $1 - \epsilon$, is to have

$$N \geq \lceil \frac{\ln(\epsilon)}{\ln(1 - \frac{1}{2^{M-card(S)}})} \rceil = \lceil \frac{\ln(\epsilon)}{\ln(1 - \frac{1}{K})} \rceil, \tag{103}$$

iterations. Since $1 \leq card(S) \leq M - 1$, so $K \geq 2$ and $\ln(1 - \frac{1}{K}) \approx \frac{-1}{K}$. For $\epsilon^* \geq \epsilon$, let $N = \lceil \frac{\ln(\epsilon^*)}{\ln(1 - \frac{1}{K})} \rceil \geq \lceil \frac{\ln(\epsilon)}{\ln(1 - \frac{1}{K})} \rceil$, then

$$N = \lceil \frac{\ln(\epsilon^*)}{\ln(1 - \frac{1}{K})} \rceil \leq \frac{\ln(\epsilon^*)}{\ln(1 - \frac{1}{K})} \approx \ln(\frac{1}{\epsilon^*}) K := cK, \tag{104}$$

which implies the number of iterations is bounded from above. If, for every $M$, we choose $K$ small enough, then the upper bound will stay within a linear growth. $\qquad \square$

### A.2.6 Proof of theorem 4

In GTF [24], during training RNN latent states are replaced by a weighted sum of forward propagated states $\boldsymbol{z}_t = F_{\boldsymbol{\theta}}(\boldsymbol{z}_{t-1})$ and data-inferred states $\bar{\boldsymbol{z}}_t = G_{\boldsymbol{\phi}}^{-1}(\boldsymbol{x}_t)$ (obtained by inversion of the decoder model $G_{\boldsymbol{\phi}}$):

$$\tilde{\boldsymbol{z}}_t := (1 - \alpha)\boldsymbol{z}_t + \alpha\bar{\boldsymbol{z}}_t, \tag{105}$$

where $0 \leq \alpha \leq 1$ is the GTF parameter (usually adaptively regulated in training, see [24]). This leads to the following factorization of Jacobians in PLRNN (2) training:

$$\boldsymbol{J}_t^{GTF} = \frac{\partial \boldsymbol{z}_t}{\partial \boldsymbol{z}_{t-1}} = \frac{\partial \boldsymbol{z}_t}{\partial \tilde{\boldsymbol{z}}_{t-1}} \frac{\partial \tilde{\boldsymbol{z}}_{t-1}}{\partial \boldsymbol{z}_{t-1}} = \frac{\partial \boldsymbol{F}_{\boldsymbol{\theta}}(\tilde{\boldsymbol{z}}_{t-1})}{\partial \tilde{\boldsymbol{z}}_{t-1}} \frac{\partial \tilde{\boldsymbol{z}}_{t-1}}{\partial \boldsymbol{z}_{t-1}} = (1 - \alpha)\tilde{\boldsymbol{J}}_t = (1 - \alpha)\boldsymbol{W}_{\Omega(t)}. \tag{106}$$

*Proof.* ($i$) Since $\|\boldsymbol{A}\| + \|\boldsymbol{W}\| \leq 1$, we have

$$\forall t \quad \|\boldsymbol{W}_{\Omega(t)}\| = \|\boldsymbol{A} + \boldsymbol{W}\boldsymbol{D}_{\Omega(t)}\| \leq \|A\| + \|W\| \|\boldsymbol{D}_{\Omega(t)}\| \leq \|A\| + \|W\| \leq 1, \tag{107}$$

and so

$$\forall t \ \rho(\boldsymbol{W}_{\Omega(t)}) \leq \|\boldsymbol{W}_{\Omega(t)}\| \leq 1, \tag{108}$$

where $\rho$ denotes the spectral radius of a matrix. In this case, for any $n \in \mathbb{N}$, we also have

$$\rho(\prod_{i=1}^{n} \boldsymbol{W}_{\Omega(t_i)}) \leq \left\|\prod_{i=1}^{n} \boldsymbol{W}_{\Omega(t_i)}\right\| \leq \prod_{i=1}^{n} \|\boldsymbol{W}_{\Omega(t_i)}\| \leq \left(\|A\| + \|W\|\right)^n \leq 1. \tag{109}$$

Now, for any $0 < \alpha < 1$, the product of Jacobians under GTF is

$$\prod_{i=1}^{n} \boldsymbol{J}_{t_i}^{GTF} = (1-\alpha)^n \prod_{i=1}^{n} \tilde{\boldsymbol{J}}_{t_i} = (1-\alpha)^n \prod_{i=1}^{n} \boldsymbol{W}_{\Omega(t_i)}, \tag{110}$$

and

$$\rho\left(\prod_{i=1}^{n} \boldsymbol{J}_{t_i}^{GTF}\right) = \rho\left((1-\alpha)^n \prod_{i=1}^{n} \boldsymbol{W}_{\Omega(t_i)}\right) = (1-\alpha)^n \rho\left(\prod_{i=1}^{n} \boldsymbol{W}_{\Omega(t_i)}\right)$$

$$\leq (1-\alpha)^n < 1. \tag{111}$$

Hence $\rho\left(\prod_{i=1}^{n} \boldsymbol{J}_{t_i}^{GTF}\right) < 1$ which implies for any $n \in \mathbb{N}$ and $0 < \alpha < 1$, the product $\prod_{i=1}^{n} \boldsymbol{J}_{t_i}^{GTF}$ has no eigenvalue equal to 1 and so no DTB can occur (see definition of DTB in sect. 3).

$(ii)$ Let $\|\boldsymbol{A}\| + \|\boldsymbol{W}\| = r > 1$, then for any $n \in \mathbb{N}$ we have

$$\rho\left(\prod_{i=1}^{n} \tilde{\boldsymbol{J}}_{t_i}\right) \leq r^n, \tag{112}$$

and thus

$$\rho\left(\prod_{i=1}^{n} \boldsymbol{J}_{t_i}^{GTF}\right) = (1-\alpha)^n \rho\left(\prod_{i=1}^{n} \boldsymbol{W}_{\Omega(t_i)}\right) \leq [(1-\alpha)\, r]^n. \tag{113}$$

Since $0 < 1 - \frac{1}{r} < 1$, inserting $1 - \frac{1}{r} < \alpha = \alpha^* < 1$ into the r.h.s. of (113) again gives $\rho\left(\prod_{i=1}^{n} \boldsymbol{J}_{t_i}^{GTF}\right) < 1$ for any $n \in \mathbb{N}$, implying that no DTB can occur. $\qquad\square$

### A.3 Additional results

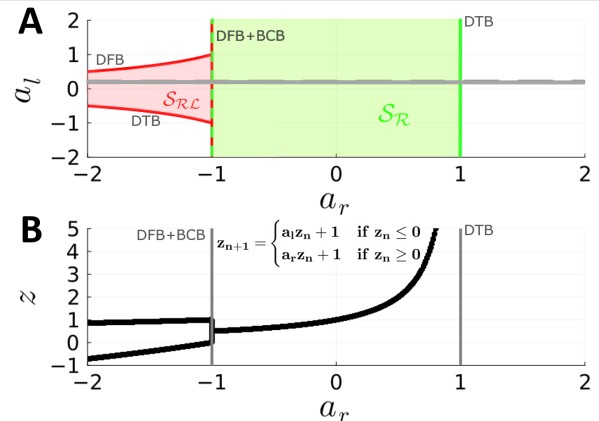

Figure S6: A) Analytically calculated stability regions for a 2-cycle $\mathcal{S}_{\mathcal{RL}}$ (red), and a fixed point $\mathcal{S}_{\mathcal{R}}$ (green) for the $1d$ skew tent map, as defined in the figure, in the parameter plane given by $(a_r, a_l)$. B) Bifurcation diagram along the cross section indicated by the gray line in A, showing a BCB and DFB occurring simultaneously at $a_r = -1$ and a DTB occurring at $a_r = 1$.

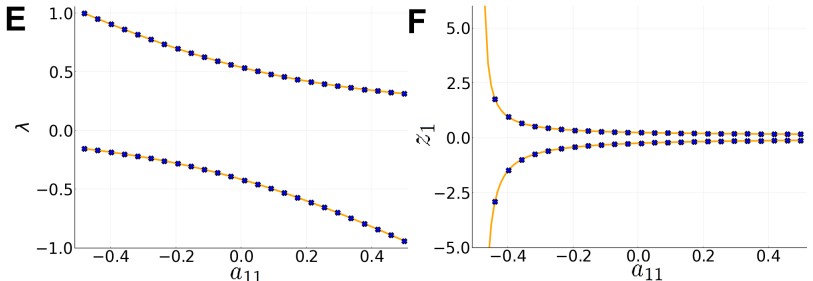

Figure S7: Results from SCYFI (blue) versus analytical results (orange) for the 2-cycle in Fig. 1 ($a_{11} + w_{11} = -2$). Eigenvalues (left) and location in state space (right) for one of the cyclic points. This confirms that fixed point locations and eigenvalues computed in closed-form and via SCYFI exactly agree, as they should.

#### A.3.1 Scaling analysis

Although the results presented in Fig. 2 suggest that SCYFI's scaling behavior is much better than theoretically expected, the fact that it is hard to obtain ground truth comparisons for high-dimensional systems (because of the combinatorial explosion) generally makes an extensive empirical analysis difficult. For Fig. 2 we therefore focused on scenarios for which we can also provide analytical curves for an exhaustive search strategy (eq. (5)) and where we then either examined scaling with cycle-order $k$ for rather low-dimensional systems, or where we explicitly embedded fixed points to search for which allowed us to move to very high dimensionality $M$. In general we observed that the scaling behavior also depended on the PLRNN's matrix norms and the eigenspectrum of the embedded fixed points, so we constructed different scenarios where we varied these factors as well.

To construct a fairly well behaved case with low matrix norms, we randomly generated matrices $\mathbf{R}$ with uniformly distributed entries in the interval $[-1, 1]$ and then normalized by its maximal eigenvalue: We set PLRNN parameters $\mathbf{A} = \frac{1}{\lambda_{max}}\mathrm{diag}(\mathbf{R})$ and $\mathbf{W} = \frac{1}{\lambda_{max}}(\mathbf{R} - \mathrm{diag}(\mathbf{R}))$, and chose $\mathbf{h}$ uniformly in the interval $[-50, 50]$. For each of 10 different such systems, we fixed the number of outer loops and inner loops ($N_{out}$, $N_{in}$ in Algorithm 1) such that a fixed point would be detected in at least $50/75$ independent runs of the algorithm, and then determined the total number $n$ of linear regions (i.e., across all $N_{out}$ different initializations) the algorithm needed to cycle through

to detect a stable fixed point. We also ensured that across all different runs this stable fixed point would be the same, in accordance with our assumptions. The resulting scaling behavior was well fitted by a doubly-logarithmic curve of the form $c_1 \ln(\ln(M)) + c_2$ ($R^2 \approx 0.913, p < 10^{-4}$). This low-matrix norm scenario with a stable fixed point may be seen as a kind of lower bound on the scaling.

To embed a specific fixed point $z^*$, we again start with a matrix $R$ as described above and take $A = \mathrm{diag}(R)$ and $W = (R - \mathrm{diag}(R))$. We then minimize

$$\min_{A,W,h} |\, z^* - ((A + W \cdot D_{\Omega(t^*)}) \cdot z^* + h) \,|, \tag{114}$$

subject to $A$ staying diagonal and $W$ off-diagonal (we observed that adding a small Gaussian noise term to the right appearance of $z^*$ in eq. 114 which decayed proportionally to the learning rate improved numerical stability in the optimization process). The such constructed PLRNNs generally have several fixed points, but to compute $n$ we only search for the inserted fixed point $z^*$ (making eq. (5) directly applicable). This way we produced $5 - 10$ systems, initializing $R$ with values in $[-0.2, 0.2]$ (orange curve in Fig. 2B) or $[-1.0, 1.0]$ (blue curve in Fig. 2B), thus effectively restricting the eigenspectrum of the fixed point as well as the matrix norms of the PLRNN to a certain range. However, since matrix norms may change during optimization, eq. (114), our procedure is not strictly guaranteed to produce eigenspectra and matrix norms within a desired range, which is crucial especially for the first scenario where we wanted to keep norms within a 'typical range' (see below). So here, to ensure consistency among drawn systems and with our assumptions, the mean absolute eigenvalue of the embedded fixed points was kept close to $0.31 \pm 0.05$ and the mean maximum absolute eigenvalue close to $1.25 \pm 0.13$. For $> 75\%$ of the resulting systems spectral matrix norms were within the range $[1.0, 3.0]$. While this produced matrix spectra typical for trained PLRNNs ($> 95\%$ out of 361 PLRNNs trained on various benchmarks and data had spectral matrix norms within $[1.0, 3.0]$), the second initialization range resulted in unnaturally large matrix norms and hence may be seen as providing a kind of upper bound on SCYFI's scaling behavior. Fig. S8 shows the best case (left; purple curve in Fig. 2B) and typical (right; orange curve in Fig. 2B) scaling scenarios on linear scale to better expose the scaling behavior and function fits.

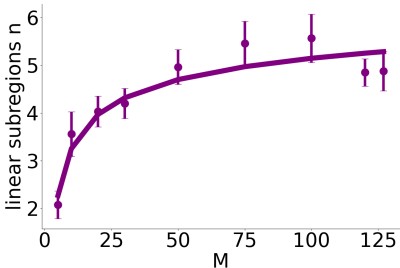 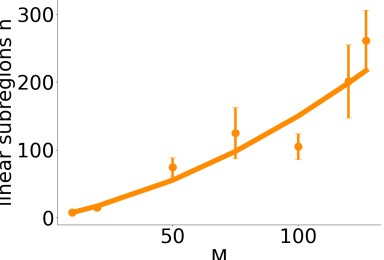

Figure S8: Zoom-ins on linear axes of the scenarios with doubly-logarithmic (left; $R^2 \approx 0.913, p < 10^{-4}$) and quadratic (right; $R^2 \approx 0.925, p < 10^{-5}$) scaling behaviors from Fig. 2B.

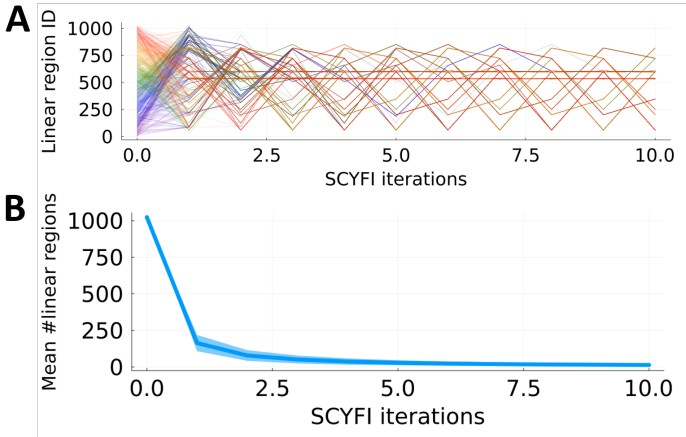

Figure S9: A) Initializing SCYFI in a wide array of different subregions (different colors), it quickly converges – within just a few iterations – to the same set of linear subregions which contain the dynamical objects of interest (fixed points in this case). B) The number of different subregions explored by SCYFI when started from different initializations shrinks exponentially fast with the number of iterations. Shown are means ($\pm$ stdv) from 10 different systems with $M = 10$.

### A.3.2 Loss jumps & bifurcations in PLRNN training on biophysical model simulations

Here we provide an additional illustration of how SCYFI can be used to dissect bifurcations in model training. For this, we produced time series of membrane voltage and a gating variable from a biophysical neuron model [17], on which we trained a dendPLRNN [10] using BPTT [52] with sparse teacher forcing (STF) [37]. The dendPLRNN used ($M = 9$ latent states, $B = 2$ bases) has $2^{18}$ different linear subregions and $|\boldsymbol{\theta}| = 124$ parameters. Fig. S10A gives the MSE loss as a function of training epoch (i.e., single SGD updates). The loss curve exhibits several steep jumps. Zooming into these points and examining the transitions in parameter space using SCYFI, we find they are indeed produced by bifurcations, with an example given in Fig. S10B. As we had done for Fig. 4 in the main text, since the state and parameter spaces are very high dimensional, for the bifurcation diagram in Fig. S10B all extracted $k$-cycles ($k \geq 1$), including fixed points, were projected onto a line given by the PCA-derived maximum eigenvalue component, and plotted as a function of training epoch. For the example in Fig. S10B, we found that a BCB (Theorem 2) underlies the transition in the qualitative dynamics of the PLRNN as training progresses. Fig. S10C illustrates the dendPLRNN dynamics just before (left) and right after (right) the bifurcation point highlighted in Fig. S10B, together with time series from the true system.

More generally, whether a bifurcation associated with *vanishing* gradients produces a loss jump depends on the system's dynamics before and after the bifurcation point. In the case of BCBs, one possible scenario involves a change in stability, as illustrated in Fig . S10. During a BCB, a stable fixed point (or cycle) can loose stability as it passes through the bifurcation point. The maximum Lyapunov exponent of an unstable fixed point (or cycle) is positive, resulting in exploding gradients right after the bifurcation point [37], and consequently to a very steep slope in the loss function near the bifurcation point as in Fig . S10.

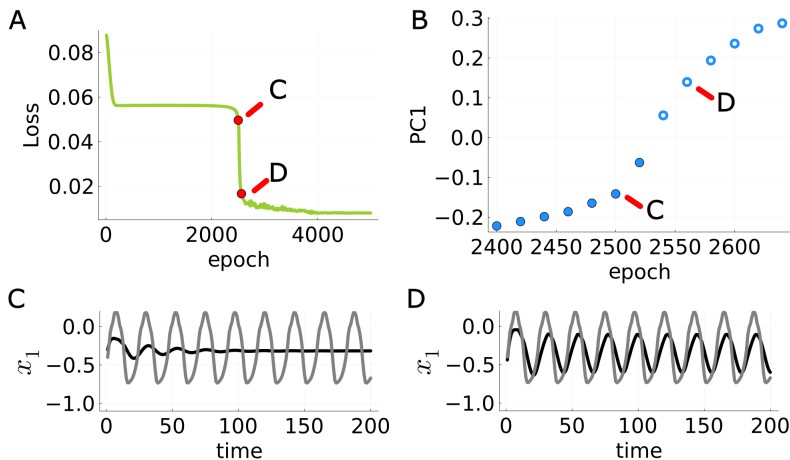

Figure S10: A) Loss across training epochs for a dendPLRNN ($M = 9$ states, $B = 2$ bases) trained on a biophysical neuron model in a limit cycle (spiking) regime. Red dots indicate training epochs just before and after a loss jump for which time graphs are given in C and D. B) Bifurcation diagram of the dendPLRNN as a function of training epoch, with all state space locations of stable (filled circles) and unstable (open circles) objects projected onto the first principle component. The loss jump in A is accompanied by a bifurcation from fixed point to cyclic behavior. C) Time series of the voltage variable ($x_1$) of the biophysical model (gray) and that predicted by the dendPLRNN (black) before the bifurcation event indicated in B. D) Same directly after the bifurcation event.

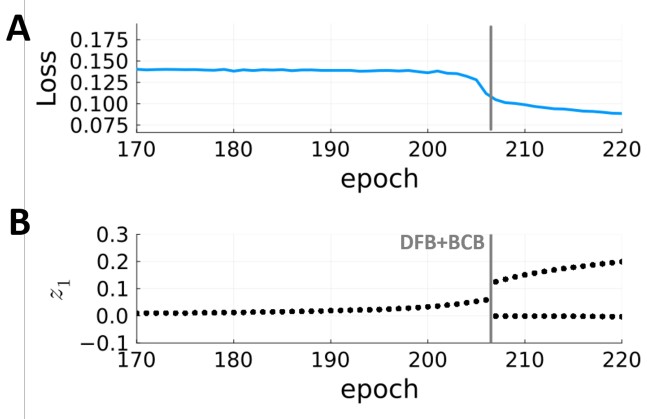

Figure S11: Loss jump induced by a degenerate flip bifurcation (DFB). A) Loss during a training run of a PLRNN ($M = 5$) on a 2-cycle. The gray line indicates a loss jump corresponding to a DFB and a simultaneously occurring border collision bifurcation (BCB). B) Bifurcation diagram of the PLRNN, with the DFB and BCB leading to the destruction of the fixed point and the emergence of a 2-cycle as indicated by the gray line.

### A.3.3 Dealing with bifurcations in RNN training

Here are some additional thoughts on how RNN training algorithms could possibly be modified to deal with bifurcations. If the algorithm finds itself during training in a parameter regime which does not exhibit the right topological structure, it does not make sense to further dwell within that regime, or possibly anywhere within the vicinity of the current parameter estimate. Unlike standard SGD, the algorithm should therefore perhaps take large leaps in parameter space as soon as it gets stuck in a non-suitable dynamical regime. One possibility to implement this is through a 'look-ahead' mechanism that probes for topological properties of regions not visited so far. While fully fleshing out this idea is beyond this paper, a proof of concept that this may speed up convergence is provided in Fig. S12. Along similar lines, if we knew the model's full bifurcation structure in parameter space ahead of time, we could simply pick a parameter set which corresponds to the right dynamics

describing patterns in the data best. While of course it will in general not be feasible to chart the whole bifurcation structure before training (this is in a sense the whole point of a training algorithm), it may be possible to design smart initialization procedures based on this insight, e.g. probing topological regimes at randomly selected points in parameter space before starting training and initializing with parameters that produce a desired type of dynamics (e.g., cyclic behavior) to begin with.

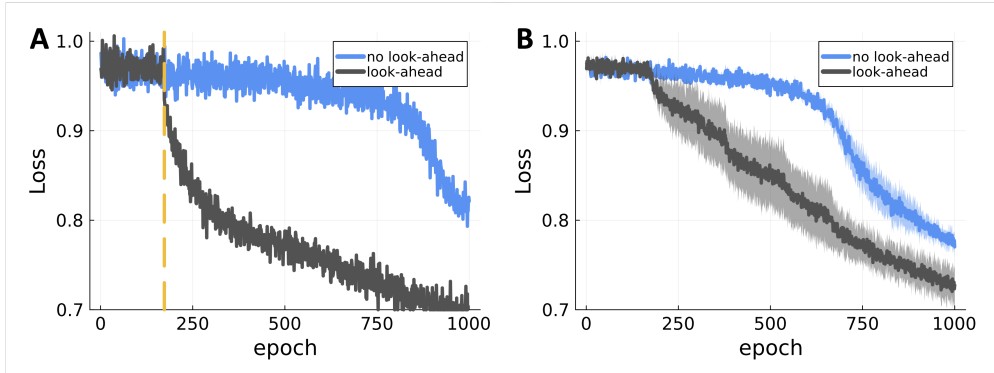

Figure S12: A) Example loss curves for RNNs trained on electrophysiological recordings by BPTT without (blue) vs. with (black) 'look-ahead' (the look-ahead function checks whether there would be a bifurcation away from a stable fixed point when taking $10\times$ the current gradient step). Dashed yellow line indicates the epoch at which the look-ahead step was executed. B) Average across 6 loss curves of RNNs trained without (blue) vs. with (black) look-ahead. Error bands = SEM.

