$$A = \frac{1}{2 + \|R\| + \epsilon} \, \text{diag}(R),$$

$$W = \frac{1}{2 + \|R\| + \epsilon} \left( R - \text{diag}(R) \right). \tag{82}$$

Then

$$\|A\| = \frac{\|\text{diag}(R)\|}{2 + \|R\| + \epsilon} < \frac{1}{2 + \|R\| + \epsilon},$$

$$\|W\| = \frac{\|R - \text{diag}(R)\|}{2 + \|R\| + \epsilon} \leq \frac{\|R\| + \|\text{diag}(R)\|}{2 + \|R\| + \epsilon} < \frac{1 + \|R\|}{2 + \|R\| + \epsilon}, \tag{83}$$

and so $\|A\| + \|W\| < 1$. Therefore

$$\forall t \quad \|W_{\Omega(t)}\| = \|A + W D_{\Omega(t)}\| \leq \|A\| + \|W\| \|D_{\Omega(t)}\| \leq \|A\| + \|W\| < 1, \tag{84}$$

and so

$$\forall t \quad \rho(W_{\Omega(t)}) \leq \|W_{\Omega(t)}\| < 1. \tag{85}$$

In this case, for any $n \in \mathbb{N}$, we also have

$$\left\| \prod_{i=1}^{n} W_{\Omega(t_i)} \right\| \leq \prod_{i=1}^{n} \|W_{\Omega(t_i)}\| \leq \left( \|A\| + \|W\| \right)^n < 1. \tag{86}$$

This ensures the stability of all fixed points and $k$-cycles of the system.

According to (85), we have

$$\left( I - W_{\Omega(t_1^*)} \right)^{-1} = \sum_{n=0}^{\infty} W_{\Omega(t_1^*)}^n = I + W_{\Omega(t_1^*)} + W_{\Omega(t_1^*)}^2 + \cdots. \tag{87}$$

Hence, all the elements of $\left( I - W_{\Omega(t_1^*)} \right)^{-1}$ are positive, and so $z_{m,t_1^*} > 0$ for every $t_1^*$. This implies that all true and virtual fixed points exist within a singular sub-region. Additionally, only one fixed point is true, while all the other fixed points are virtual.

**Case (II)** : Let $h = (h_1, h_2, \cdots, h_M)^\mathsf{T}$ be a random vector with all $h_m$ uniformly distributed in $(0, 1]$ and

$$\beta_{min} = \min\left\{ h_m : h_m \in h, 1 \leq m \leq M \right\} > 0, \qquad 0 < \beta_{min} \leq 1,$$

$$\beta_{max} = \max\left\{ h_m : h_m \in h, 1 \leq m \leq M \right\} > 0, \qquad 0 < \beta_{max} \leq 1. \tag{88}$$

Assume further that $\boldsymbol{R}_1$ is a randomly generated matrix with uniformly distributed entries in the interval $(-1, 0]$, and for $M \geq 2$

$$\boldsymbol{W} = \frac{\beta_{min}}{M + \|R_1\| + \epsilon} \left( \boldsymbol{R}_1 - \text{diag}(\boldsymbol{R}_1) \right). \tag{89}$$

Consider