# OpenReview forum: "Bifurcations and loss jumps in RNN training"
_NeurIPS.cc/2023/Conference — NeurIPS 2023 spotlight_

### Official Review · Reviewer_v6FA · 2023-07-03

**Soundness:** 2 fair
**Presentation:** 2 fair
**Contribution:** 3 good
**Rating:** 6
**Confidence:** 3

**Summary:**

The paper studies the dynamical systems properties of piecewise-linear recurrent neural networks (PLRNNs). Specifically:
- Section 3.1 (and its accompanying Appendix A.1-A.5) analytically categorizes the fixed points of 2-dimensional PLRNNs, and analyzes its bifurcations by explaining how the fixed points (and cycles) change as a function of the network parameters. The analysis involves computing the Jacobian of the recurrent unit and classifying fixed points by using its spectrum. The paper specifically identifies instances of multi-stability, where where multiple attractor regions exist.
- Theorems 1 and 2 in Section 3.2 show that a PLRNN whose unit undergoes a degenerate transcritical bifurcation (DTB) or a border collision bifurcation (BCB) has gradients trending towards infinite and zero respectively as the bifurcation occurs in parameter space.
- Section 4 introduces the SCYFI algorithm for identifying fixed points and cycles in higher dimensional PLRNNs. SCYFI is a heuristic algorithm that finds k-cycles by using exploring the exponentially large space of linear regions from a randomly initialized space of regions. They find empirically that the algorithm finds fixed points much faster than brute-force algorithms that search naively search each subregion.
- Section 5 applies SCYFI and their bifurcation analysis to the prediction of a neuron spiking with a PLRNN and the resulting detection of a stable 39-cycle that predicts the spike. They observe a bifurcation in the training procedure where the stable 39-cycle is eradicated due to a DTB that causes the loss to spike (as predicted by Theorem 1's analysis of exploding gradients.)

**Strengths:**

As far as the reviewer is aware, the paper provides novel analysis of bifurcations on PLRNNs and identifies innovative applications of their techniques through their neuroscience application. The analysis appears to be correct. The theoretical results effectively draw connections between bifurcation theory and gradient-based training and are validated experimentally with the loss jump coinciding the the DTB bifurcation in Section 5.

**Weaknesses:**

As of now, the SCYFI algorithm as written is highly ambiguous, and I cannot recommend acceptance as long as that holds. (These ambiguities are noted in the next part of the review.) I am happy to reconsider if these concerns and concerns about presentation are satisfactorily addressed, given the innovative nature of the results.

**Note: the score of the paper was increased from 4 to 6, because the authors rewrote the algorithm more precisely.**

The results of the paper appear to be novel from a dynamical systems perspective, but their current presentation of the results may be inaccessible for the ML community of NeurIPS. Given the relative novelty of work on the intersection of dynamical systems and ML, clear explanation of dynamical systems concepts and their relevance to ML is key.
- In particular, the paper could more explicitly trace the importance of being able to identify k-cycles and bifurcations when training neural networks. The application in Section 5 is interesting and it could be discussed more prominently in the intro to motivate the study of k-cycles for ML researchers looking for more justification.
- The paper would benefit from more rigorously defining the bifurcations under consideration, like BCBs and DTBs, as well as having a visualization in the parameter space of a toy example.
- The plots in Figure 1 may be unintuitive to NeurIPS readers with a limited dynamical systems background, especially as Figure 1B is not discussed in the text. This applies more broadly to figures in the main body: including more discussion of what concretely is represented in each plot (while potentially pushing some plots to the appendix) would make for a clearer read.

Minor typos and notes:
- $\Gamma$ is notationally overloaded: In 3.1, it refers the spectrum of a fixed point. In 3.2, it is a $k$-cycle.
- On l238, the code's location is referred to by "[PLACEHOLDER]."
- The first paragraph of Section 5 refers to Figure 4, but likely means Figure 3.

**Questions:**

The paper chooses to ignore external inputs $s_t$ to its RNNs, which seems like a very significant omission, given that the primary applications of RNNs in the past have been to NLP and other sequence-learning tasks. Since [some results](http://proceedings.mlr.press/v97/hanin19a/hanin19a.pdf) suggest that adding random noise RNNs can dramatically change the structure of fixed points, are the results expected to hold in regimes with noisy or structured inputs $s_t$ included?

As written, the SCYFI algorithm is somewhat ambiguous to the reviewer:
- $z_n$ and $D_n$ terms are ambiguous. $z_n $ appears to be a known cycle ahead of time (line 1), component obtained by solving equation (7) (line 8), and a cycle already stored in $\mathcal{L}_{k - m}$ (line 9). Which of these sets are being compared in the RHS of line 9?
- Why does the algorithm start out by presuming to know some $z_n$ and $D_n$, or is that just setting up notation for future lines? (line 1)
- When solving equation (7), what precisely is being solved for?  The equation relates $z_k^*$, $W_{\Omega(k-r)}$, and $h$. What is known ahead of time, and what must be solved for? Also, how do we solve the equation? If we are solving for $D_n$ by solving for $W_{\Omega(n)}$, then it appears that the equation is not linear.
- Does the algorithm only work if every $\mathcal{L}_n$ for $n < k$ is exactly correct? While it's difficult to prove convergence results due to the searching of an exponential space, some kind of theoretical guarantee about what is necessary for the algorithm to be successful would be useful, even if just in an appendix.


**Limitations:**

Besides the omission of $s_t$ from the analysis, limitations of the paper appear to be well-documented by the authors.

---

> ### Author Rebuttal · Authors · 2023-08-09
>
> We thank the referee for her/his constructive comments and for bringing up a couple of important points we agree need to be addressed.
>
> **Weaknesses**
>
> *”Improving presentation”*: We will 1) take up the referee’s suggestions and add a pg. to the Introduction which primes more clearly for the importance of detecting bifurcations and cycles, 2) revise sect. 3.1 to introduce these concepts more succinctly while deferring some of the mathematical details to the Appx.
>
> *”Toy example \& better discussion of BCB/DTB, Fig. 1 unintuitive”*: Yes, fair point. We addressed this now by thoroughly reworking Fig. 1 (see Fig. R2 in included PDF file) that now specifically features BCBs and DTBs to bring it into closer agreement with the main results. We will also rewrite sect. 3.1 (see above) and add math. precise def. of BCBs and DTBs. Some of the other material in sect. 3.1, which may have been more tangential to the main messages of our paper, will instead be deferred to the Appx. Figs. 2-4 are all important though, we believe, but we will add further clarification to the legends and the discussion in the text.
>
> **Questions**
>
> *Q1 (“Ignoring inputs”)*: The presence of inputs does not change any of our results. Since inputs do not affect the Jacobians in eqn. (71), (72) and (77), they do not change our two theorems (even if they would affect the Jacobians, Theorem 2 would be unaltered, and Theorem 1 could be amended in a straightforward way). Furthermore, since we are addressing bifurcations that occur during model *training*, from this angle inputs may simply be treated as either additional parameters (if piecewise constant) or states of the system (without changing any of the mathematical derivations). In fact, mathematically, any non-autonomous dynamical system (RNN with inputs) can always and strictly be reformulated as an autonomous system (RNN without inputs), see (Alligood et al., 1996; Perko, 2001; Zhang et al., 2009, Controlling Chaos, Springer).
> We will make this very explicit now in sects. 3-5 of the revised paper. We also thank the referee for the interesting reference!
>
> *Q2 (“SCYFI algorithm ambiguous”)*: Thank you for pointing out, we fully agree the pseudo-code could have been presented in a much clearer way, and include a thoroughly updated version in the general rebuttal (this will be formatted more nicely in the paper, the openreview boxes were a bit limited here).
> In brief (lines refer to original version): Line 1 was simply meant to introduce the notation, the cycle is not known a priori. In eq. 7 (line 8) we obtain a *candidate cycle*, not a true cycle yet. And in line 9, the set may not include that specific cycle yet, but there may be other cycles of order $k$ already detected in previous iterations (there are rare instances where the algorithm may stumble across the same cycle via a different route, and we found it’s more cheaply just to check for it again rather than avoiding this to begin with). Furthermore, in line 9 then a consistency check is performed: The candidate cycle obtained by solving eg. 7 is only a *true cycle* 1) if its corresponding set of \\(D^\*\_l\\) matrices agrees with \\(D_\{init}\\), and 2) if it is not a superset of an already detected lower-order cycle (or otherwise already in the list).
> We hope this becomes all much clearer from our revised pseudo-code as provided in the general response to all referees further above.
>
> *Q3*: As noted above, in line 1 we just set up the notation and do not presume to know any specific $z_n$ or $D_n$ ahead of time.
>
> *Q4*: In eq. 7 we solve for a *candidate cycle* \\( z^\*\_k \\) on the l.h.s. (now indicated also with a \* in the pseudo-code), given all the terms on the r.h.s.: The PLRNN parameters $A$, $W$ and $h$ are provided as inputs, and the set of matrices $D_n$ needed to solve this eq. is the initial guess $D_{init}$, which determines $W_{\Omega(n)}$. The next line then checks whether the initial guess $D_{init}$ is indeed consistent with the signs of $z^*_k$ in the respective linear subregions (so that the solution in eq. 7 is indeed self-consistent and the cycle is a true one).
> This should (hopefully) all be much clearer now from our revised pseudo-code that can be found at the top in the general rebuttal.
>
> *Q5*: Because the cycle candidates are determined analytically and the next step just checks consistency, the solutions will in fact always be exact (down to computer precision).
>
> *Q6 (convergence results)*: Obtaining rigorous theoretical guarantees is challenging in the current case (and sometimes, historically, theoretical understanding of convergence and other properties of an algorithm lagged behind by many years, eg. Simplex or various training tweaks in DL like drop-out).
> However, in the meantime we made some progress here: We can now strictly prove that the algorithm will converge with probability almost 1 under some general conditions on the PLRNN parameters, and we will add the theorem and proof to our revision. Furthermore, we can add some numerical results on this topic: As shown in Fig. R1 in the included PDF file, when we initiate SCYFI in different randomly selected linear subregions, it converges to the subregions including the dynamical objects of interest *exponentially fast*. This numerical observation further helps to explain its efficiency.
> The scaling behavior demonstrated in Fig. 2B further alludes to that: Except for quite artificial and extreme situations that we never encountered in a large database of PLRNNs trained on empirical data (see Appx. sect. A.7), the scaling is always well behaved. We will make this more explicit in sect. 4 where Fig. 2 is discussed.
>
> *Limitations*: See our response to this point above.
>
> **Minor points \& typos**:
>
> 1) Will be changed for clarity.
> 2) Was meant as a placeholder for our github site where we will make the code publicly available, not yet included to prevent deanonymization.
> 3) Thanks for spotting this, will be corrected!

---

> > ### Comment · Reviewer_v6FA · 2023-08-15
> >
> > I thank the authors for their detailed response to my concerns and for their commitment to addressing the issues pointed out by myself and other reviewers. I especially appreciate the changes to the description of the SCYFI algorithm. The improvements in clarity are much appreciated as well. I am happy to amend my score of the paper, based on these changes.
> >
> > Regarding the convergence results, would it be possible for the authors to share any new theoretical results with myself and the other reviewers? I would appreciate the chance to review the proofs and theorem statements to ensure that they are correct and relevant to the rest of the paper.

---

> > > ### Author Response · Authors · 2023-08-17
> > > **Proofs**
> > >
> > > We thank the referee very much for the kind response and appreciation of our work!
> > >
> > > Typing in longer theorem proofs in jax in the limited comment boxes is a bit of a challenge (note that we are not allowed to upload another pdf!). We will try to spread the proof across several comments, if that is possible.
> > >
> > > We can offer two theoretical results. The first states that SCYFI almost surely converges (i.e., with probability tending to 1). The proof of this is relatively straightforward, here is the idea: Drawing uniformly from the $2^N$ different $D$-matrices (linear subregions) for initialization (see revised algorithm), the probability that a particular subregion has not been drawn after $k$ repetitions is $p=\left(1 - \frac{1}{2^N}\right)^k$. Hence, in order to ensure that all $2^N$ subregions have been visited with probability $1-\epsilon$ we need  $k \geq [\frac{\ln(\epsilon)}{\ln\left(1 - \frac{1}{2^N}\right)}] $ iterations. Choosing $N_{out}=k$, we can thus ensure that SCYFI was initialized in each subregion with probability almost 1, and thus, in the limit, will have probed all subregions for dynamical objects.
> > >
> > > However, we also managed to obtain a more specific and stronger theoretical result about SCYFI’s convergence speed under certain conditions on the parameters (which agrees nicely with numerical observations we made, cf. Fig. 2). It rests on the observation that SCYFI is designed to move only among subregions containing virtual or actual fixed points based on the fact that it is always reinitialized with the next virtual fixed point in case the consistency check fails. The result can be stated as follows:
> > >
> > > **Theorem.** Consider a PLRNN of the form (2) with parameters \\( \boldsymbol{\theta}= \\{A, W, h \\} \\). Under certain conditions on \\( \boldsymbol{\theta} \\) (for which \\( ||A||+||W|| <1) \\), SCYFI will converge in at most linear time.
> > >
> > > **Proof**
> > > We examine the convergence and scaling of SCYFI for fixed points. Similar arguments apply to \\(k\\)-cycles where \\(k > 1\\).
> > > \
> > > \
> > > Let \\(\mathbf{z}\_{1}^*\\) be a fixed point of the system, i.e.
> > > \
> > > \
> > > \\(
> > > \mathbf{z}\_{1}^* \\, = \\, (\mathbf{I} - \mathbf{W}\_{\Omega(t\_{1}^*)}  )^{-1} \\, \mathbf{h},\hspace{1cm} (\mathbf{1})
> > > \\)
> > > \
> > > \
> > > \\(\mathbf{z}\_{1}^*\\) is a true fixed point iff
> > > \
> > > \
> > > \\(
> > > (d\_m(t\_{1}^*) - a) \cdot z\_{m, t\_{1}^*} \\, > \\, 0 \hspace{1cm} \forall \\, m \in \{1, 2, \cdots, M \},\hspace{1cm} (\mathbf{2})
> > > \\)
> > > \
> > > \
> > > where \\(\mathbf{D}\_{\Omega(t\_{1}^*)} = \text{diag}(d\_1(t\_{1}^*), d\_2(t\_{1}^*), \cdots, d\_M(t\_{1}^*))\\), and \\(0<a<1\\) is a positive real constant.
> > > \
> > > \
> > > We will consider two specific scenarios for conditions on the parameters \\(\theta\\) in the following.
> > >
> > >
> > > **Case (I)**  Let \\(\mathbf{R}\\) be a randomly generated matrix with uniformly distributed entries in the interval \\([0,1)\\), and \\(\mathbf{h} \neq 0\\) be a random vector with all its components being non-negative. For an arbitrary \\(\epsilon >0\\), we set
> > > \
> > > \
> > > \\(
> > > \mathbf{A} = \frac{1}{2+\|\|\mathbf{R}\|\|+\epsilon} \\, \text{diag}(\mathbf{R}),
> > > \\)
> > > \
> > > \
> > > \\(
> > > \mathbf{W}  = \frac{1}{2+\|\|\mathbf{R}\|\|+\epsilon} \\, (\mathbf{R}-\text{diag}(\mathbf{R}) ). \hspace{1cm} (\mathbf{3})
> > > \\)
> > > \
> > > \
> > > Then
> > > \
> > > \
> > > \\(
> > > \|\|\mathbf{A}\|\|  = \frac{\|\|\text{diag}(\mathbf{R})\|\| }{2+\|\|\mathbf{R}\|\|+\epsilon}  \\, < \\, \frac{1}{2+\|\|\mathbf{R}\|\|+\epsilon},
> > > \\)
> > > \
> > > \
> > > \\(
> > > \|\|\mathbf{W}\|\|  = \frac{\|\|\mathbf{R}-\text{diag}(\mathbf{R})}{2+\|\|\mathbf{R}\|\|+\epsilon}  \\, \leq \\, \frac{\|\|\mathbf{R}+ \|\|\text{diag}(\mathbf{R}) }{2+\|\|\mathbf{R}\|\|+\epsilon} \\, < \\, \frac{1+\|\|\mathbf{R}\|\|}{2+\|\|\mathbf{R}\|\|+\epsilon},\hspace{1cm} (\mathbf{4})
> > > \\)
> > > \
> > > \
> > > and so  \\(\|\|\mathbf{A}\|\| +\|\|\mathbf{W} \|\|  < 1\\). Therefore
> > >
> > >
> > > \\( \forall t \\, \\, \\, \|\|\mathbf{W}\_{\Omega(t)} || = \|\|\mathbf{A} +\mathbf{W} \mathbf{D}\_{\Omega(t)}\|\| \leq \|\|\mathbf{A}\|\|+\|\|\mathbf{W}\|\|\|\|\mathbf{D}\_{\Omega(t)}\|\| \leq \|\|\mathbf{A}\|\|+\|\|\mathbf{W}\|\| < 1,\hspace{1cm} (\mathbf{5})\\)
> > > \
> > > and so
> > > \
> > > \
> > > \\(
> > > \forall t \\, \\, \\, \\, \\,  \rho(\mathbf{W}\_{\Omega(t)}) \leq \|\|\mathbf{W}\_{\Omega(t)}\|\| < 1.\hspace{1cm} (\mathbf{6})
> > > \\)
> > > \
> > > In this case, for any \\(n \in \mathbb{N}\\), we also have
> > > \
> > > \
> > > \\(
> > > \|\|\prod\_{i=1}^{n} \mathbf{W}\_{\Omega(t\_{i})}\|\| \\, \leq \\, \prod\_{i=1}^{n} \|\|\mathbf{W}\_{\Omega(t\_{i})}\|\| \\, \leq \\, (\|\|\mathbf{A}\|\|+\|\|\mathbf{W}\|\|)^n  \\, < \\, 1.\hspace{1cm} (\mathbf{7})
> > > \\)
> > > \
> > > \
> > > This guarantees the stability of all fixed points and \\(k\\)-cycles of the system.
> > > \
> > > According to 6, we have
> > > \
> > > \
> > > \\(
> > > (\mathbf{I} - \mathbf{W}\_{\Omega(t\_{1}^*)} )^{-1} = \sum\_{n=0}^{\infty} \mathbf{W}\_{\Omega(t\_{1}^*)}^n  \\, = \\, \mathbf{I} + \mathbf{W}\_{\Omega(t\_{1}^*)} + \mathbf{W}\_{\Omega(t\_{1}^*)}^2 + \cdots.\hspace{1cm} (\mathbf{8})
> > > \\)

---

> > > > ### Author Response · Authors · 2023-08-17
> > > > **Proof continued**
> > > >
> > > > Hence all the elements of \\((\mathbf{I} - \mathbf{W}\_{\Omega(t\_{1}^*)} )^{-1}\\) are positive, and so \\(z\_{m, t\_{1}^*} > 0\\) for every \\(t\_{1}^*\\). This implies that all true and virtual fixed points exist within a single sub-region. Additionally, only one fixed point is true, while all the other fixed points are virtual. Since, by construction, during the first iteration SCYFI will solve for a virtual or a true fixed point, it will land in the right subregion after only 1 iteration and obtain the true fixed point after at most 2 iterations.
> > > >
> > > > **Case (II)**: Let \\(\mathbf{h}  = (h\_1, h\_2, \cdots, h\_M)^T\\) be a random vector with all \\(h\_m \\) uniformly distributed in \\((0, 1]\\) and
> > > > \
> > > > \
> > > > \\(
> > > > \beta\_{\text{min}}  = \min \\{ h\_{m} \\, : \\, h\_{m} \in \mathbf{h}, \\, 1 \leq m \leq M  \\} > 0, \hspace{1cm}  0 < \beta\_{\text{min}}  \leq 1,
> > > > \\)
> > > > \
> > > > \
> > > > \\(
> > > > \beta\_{\text{max}}  = \max \\{ h\_{m} \\, : \\, h\_{m} \in \mathbf{h}, \\, 1 \leq m \leq M  \\} > 0. \hspace{1cm}  0 < \beta\_{\text{max}}  \leq 1\hspace{1cm} (\mathbf{9})
> > > > \\)
> > > > \
> > > > \
> > > > Assume further that \\(\mathbf{R}\_1\\) is a randomly generated matrix with uniformly distributed entries in the interval \\((-1, 0]\\), and for \\(M \geq 2\\)
> > > > \
> > > > \
> > > > \\(
> > > > \mathbf{W}  = \frac{\beta\_{\text{min}}}{M+\|\|\mathbf{R}\_1\|\|+\epsilon} \\, (\mathbf{R}\_1-\text{diag}(\mathbf{R}\_1) ).\hspace{1cm} (\mathbf{10})
> > > > \\)
> > > > \
> > > > \
> > > > Consider
> > > > \
> > > > \
> > > > \\(
> > > > \alpha\_{\text{max}}  = \max \\{ |w\_{ij}| \\, : \\, w\_{ij} \in \mathbf{W}  \\}, \hspace{1cm}  0 \leq  \alpha\_{\text{max}} < \frac{\beta\_{\text{min}}}{M+\|\|\mathbf{R}\_1\|\|+\epsilon}\hspace{1cm} (\mathbf{11})
> > > > \\)
> > > > \
> > > > \
> > > > and \\(S \subset \\{1, 2, \cdots, M \\} = I \\, \\) such that \\(K = 2^{M-\text{card}(S)} \ll 2^M\\). Suppose that \\(\mathbf{R}\_2 = \text{diag} (r\_1, \cdots, r\_M)\\) is a randomly chosen diagonal matrix with \\(r\_m\\) uniformly distributed in  (\\(-1, 1\\)) for \\(m \in I \setminus S \\), and the other elements(\\(m \in S \\)) uniformly distributed in  (\\(r^{\*} -1, 0\\))  where \\(r^{\*} = \frac{(M-1)\alpha\_{\text{max}}\\, \beta\_{\text{max}}}{ \beta\_{\text{min}}}\\). Since
> > > > \
> > > > \
> > > > \\(
> > > > 0 \leq \frac{(M-1)\alpha\_{\text{max}}\\, \beta\_{\text{max}}}{ \beta\_{\text{min}}} \\, < \\, \frac{(M-1)\\, \beta\_{\text{max}}}{ M+\|\|\mathbf{R}\_1\|\|+\epsilon} \\, \leq \\, \frac{(M-1)}{ M+\|\|\mathbf{R}\_1\|\|+\epsilon} \\, < \\, \frac{(M-1)}{ M} < 1,\hspace{1cm} (\mathbf{12})
> > > > \\)
> > > > \
> > > > \
> > > > so  \\(  -1 \leq r^{\*}-1 < 0\\).
> > > > \
> > > > \
> > > > If
> > > > \
> > > > \
> > > > \\(
> > > > \mathbf{A} = \frac{1}{2+\|\|\mathbf{R}\_1\|\|+\epsilon} \\, \mathbf{R}\_2,\hspace{1cm} (\mathbf{13})
> > > > \\)
> > > > \
> > > > \
> > > > then
> > > > \
> > > > \
> > > > \\(
> > > > \|\|\mathbf{A}\|\|  = \frac{\|\|\mathbf{R}\_2} {2+\|\|\mathbf{R}\_1\|\|+\epsilon}  \\, < \\, \frac{1}{2+\|\|\mathbf{R}\_1\|\|+\epsilon},
> > > > \\)
> > > > \
> > > > \
> > > > \\(
> > > > \|\|\mathbf{W}\|\|  = \frac{\beta\_{\text{min}}\|\|\mathbf{R}\_1-\text{diag}(\mathbf{R}\_1)}{M+\|\|\mathbf{R}\_1\|\|+\epsilon}  \\, \leq \\, \frac{\|\|\mathbf{R}\_1+ \|\|\text{diag}(\mathbf{R}\_1) }{M+\|\|\mathbf{R}\_1\|\|+\epsilon} \\, < \\, \frac{1+\|\|\mathbf{R}\_1\|\|}{2 +\|\|\mathbf{R}\_1\|\|+\epsilon},\hspace{1cm} (\mathbf{14})
> > > > \\)
> > > > \
> > > > \
> > > > which implies
> > > > \
> > > > \
> > > >  \\(\|\|\mathbf{A}\|\| + \|\|\mathbf{W} \|\| < 1\\).
> > > > \
> > > > \
> > > > We set \\(\\,\epsilon >0\\) large enough to satisfy the condition
> > > > \
> > > > \
> > > > \\(
> > > > (\mathbf{I} - \mathbf{W}\_{\Omega(t\_{1}^{\*})} )^{-1} = \sum\_{n=0}^{\infty} \mathbf{W}\_{\Omega(t\_{1}^{\*})}^n  \\, \approx \\, \mathbf{I} + \mathbf{W}\_{\Omega(t\_{1}^{\*})} \hspace{1cm} \forall \\, t\_{1}^{\*}.\hspace{1cm} (\mathbf{15})
> > > > \\)
> > > > \
> > > > \
> > > > On the other hand, for any \\(t\\), we have
> > > > \
> > > > \
> > > > \\(
> > > > \mathbf{W}\_{\Omega(t)}\\, = \\, \mathbf{A} + \mathbf{W} \mathbf{D}\_{\Omega(t)} =\begin{pmatrix}
> > > > a\_{11} & w\_{12}d\_2 (t) & w\_{13}d\_3(t) & \cdots & w\_{1M}d\_M(t) \\\ w\_{21}d\_1(t)  & a\_{22} & w\_{23}d\_3(t)  & \cdots & w\_{2M}d\_M(t)  \\\ w\_{31}d\_1(t)  & w\_{32}d\_2(t)  & a\_{33} & \cdots & w\_{3M}d\_M(t) \\\ \vdots & \vdots & \vdots & \ddots & \vdots \\\ w\_{M1}d\_1(t)  & w\_{M2}d\_2(t)  & w\_{M3}d\_3(t)  & \cdots & a\_{MM} \\\ \end{pmatrix}.\hspace{1cm} (\mathbf{16})
> > > > \\)
> > > > \
> > > > \
> > > > Hence
> > > > \
> > > > \
> > > > \\(
> > > > z\_{m, t\_{1}^*} \\, = \\, (1 + a\_{mm}) \\, h\_m + \underset{j \neq m}{\sum\_{j=1}^{M}} w\_{mj} \\, d\_j(t\_{1}^*) h\_j \\, = \\, (1 + a\_{mm}) \\, h\_m - \underset{j \neq m}{\sum\_{j=1}^{M}} | w\_{mj} | \\, d\_j(t\_{1}^*) h\_j.\hspace{1cm} (\mathbf{17})
> > > > \\)
> > > > \
> > > > \
> > > > Since, for every \\(t\_{1}^*\\)
> > > > \
> > > > \
> > > > \\(
> > > > \underset{j \neq m}{\sum\_{j=1}^{M}} | w\_{mj} | \\, d\_j(t\_{1}^*) h\_j \\, \leq \\, \underset{j \neq m}{\sum\_{j=1}^{M}} | w\_{mj} | \\,  h\_j,\hspace{1cm} (\mathbf{18})
> > > > \\)
> > > > \
> > > > \
> > > > so
> > > > \
> > > > \
> > > > \\(
> > > > z\_{s, t\_{1}^*} \\, \geq \\, (1 + a\_{ss} )\\, h\_s - \underset{j \neq s}{\sum\_{j=1}^{M}} | w\_{sj} | \\,  h\_j  \hspace{1cm} \forall \\, s \in S.\hspace{1cm} (\mathbf{19})
> > > > \\)
> > > > \

---

> > > > > ### Author Response · Authors · 2023-08-17
> > > > > **Proof continued further**
> > > > >
> > > > > Moreover \\(\\, a\_{ss} \in (r^*-1, 0)\\), for every \\(s \in S\\), and thus
> > > > > \
> > > > > \
> > > > > \\(
> > > > > a\_{ss}+ 1 \\, > \\, \frac{(M-1) \alpha\_{\text{max}} \beta\_{\text{max}}}{\beta\_{\text{min}}}\\, = \\, \frac{\underset{j \neq s}{\sum\_{j=1}^{M}} \alpha\_{\text{max}} \beta\_{\text{max}}}{\beta\_{\text{min}}} \\, \geq \\, \frac{\underset{j \neq s}{\sum\_{j=1}^{M}} | w\_{sj} | \\,  h\_j}{h\_s}.\hspace{1cm} (\mathbf{20})
> > > > > \\)
> > > > > \
> > > > > \
> > > > > Therefore, due to (19) and (20), \\(z\_{s, t\_{1}^*} >0\\) for every \\(t\_{1}^*\\) and \\(s \in S\\). This means that all true and virtual fixed points only exist within a relatively small number of sub-regions, denoted as \\( K = 2^{M-\text{card}(S)} \ll 2^M\\).
> > > > > \
> > > > > \
> > > > > Given our specific initialization of \\( \boldsymbol{\theta} \\), there is a set of \\( K \\) different sub-regions, each associated with a unique \\( \mathbf{D}_{\Omega(t)} \\) matrix. We refer to the entire set of these matrices as
> > > > > \
> > > > > \\(
> > > > > \mathcal{D}_K = \\{ \mathbf{D}_1, \ldots, \mathbf{D}_K \\}.\hspace{1cm} (\mathbf{21})
> > > > > \\)
> > > > > \
> > > > > \
> > > > > SCYFI, by construction, only moves between subregions that harbor a virtual or a true fixed point, continuing until it discovers a true fixed point (or gets stuck in a virtual cycle). Thus, it can iterate between \\( J \leq K \\) sub-regions
> > > > > \
> > > > > \
> > > > > \\(
> > > > > \mathcal{D}_J = \\{ \mathbf{D}_1, \ldots, \mathbf{D}_J \\} \subseteq  \mathcal{D}_K,\hspace{1cm} (\mathbf{22})
> > > > > \\)
> > > > > \
> > > > > \
> > > > > or within the set of virtual fixed points
> > > > > \
> > > > > \\(
> > > > > \mathcal{Z}_L = \\{ \mathbf{z}_1, \ldots, \mathbf{z}_L \\}.\hspace{1cm} (\mathbf{23})
> > > > > \\)
> > > > > \
> > > > > \
> > > > > Suppose SCYFI follows the virtual/true fixed point structure within these \\(K\\) sub-regions. To discover within this set the true fixed point with high probability \\(1 - \epsilon\\), it needs
> > > > > \
> > > > > \
> > > > > \\(
> > > > > N \geq \left\lfloor \frac{\ln(\epsilon)}{\ln\left(1 - \frac{1}{2^{M - \text{card}(S)}}\right)} \right\rfloor = \left\lfloor \frac{\ln(\epsilon)}{\ln\left(1 - \frac{1}{K}\right)} \right\rfloor,\hspace{1cm} (\mathbf{24})
> > > > > \\)
> > > > > \
> > > > > \
> > > > > iterations. Since \\(1 \leq \text{card}(S) \leq M - 1\\), so \\(K \geq 2\\) and \\(\ln\left(1 - \frac{1}{K}\right) \approx \frac{-1}{K}\\). For \\(\epsilon^* \geq \epsilon\\), let \\(N = \left\lfloor \frac{\ln(\epsilon^*)}{\ln\left(1 - \frac{1}{K}\right)} \right\rfloor \geq  \left\lfloor \frac{\ln(\epsilon)}{\ln\left(1 - \frac{1}{K}\right)} \right\rfloor\\), then
> > > > > \
> > > > > \
> > > > > \\(
> > > > > N = \left\lfloor \frac{\ln(\epsilon^* )}{\ln\left(1 - \frac{1}{K}\right)} \right\rfloor \leq \frac{\ln(\epsilon^* )}{\ln\left(1 - \frac{1}{K}\right)} \approx \ln\left(\frac{1}{\epsilon^*} \right) K =: cK,\hspace{1cm} (\mathbf{25})
> > > > > \\)
> > > > > \
> > > > > \
> > > > > which implies the number of iterations is bounded from above. If, for every \\(M\\), we choose \\(K\\) small enough, then the upper bound will stay within a linear growth.

---

### Official Review · Reviewer_T1NJ · 2023-07-06

**Soundness:** 4 excellent
**Presentation:** 3 good
**Contribution:** 2 fair
**Rating:** 7
**Confidence:** 4

**Summary:**

The paper analyses a class of ReLU-RNNs (PLRNNs) from the perspective of bifurcations in dynamical systems. This gives rise to new insights into the training process of RNNs (as specific bifurcations are associated with exploding or vanishing gradients -- a major issue in training RNNs on long-term dependency data), but also into the behavior of the trained RNN. The paper further proposes a novel heuristic algorithm to detect bifurcation manifolds in PLRNNs. Finally, the paper provides empirical evidence demonstrating the theoretical findings.


**Strengths:**

The paper is very well written and easy to follow. The idea of applying bifurcation analysis in such a rigorous and broad manner is novel up to my knowledge. Moreover, the authors manage to leverage this perspective to gain new and interesting insights into the training behavior of a broad class of RNNs. The reviewer further appreciates that the paper does not over-claim but rather truthfully demonstrates and discusses its contributions.

**Weaknesses:**

There are not many weaknesses. The paper is novel and interesting. However, one might argue that besides a novel theoretical perspective into the training process (restricted to the exploding and vanishing gradient problem, which has been studied in RNNs for several decades now), there is very little significance of this work. Although one can find some theoretical justification using the proposed SCYFI algorithm on why the training exhibits loss jumps, this has no influence on the training itself. More concretely, even though one might know a-priori that the RNN will run into a bifurcation associated with exploding or vanishing gradients, this does not change the training procedure.
Therefore, in order to argue that the proposed method is not only novel, interesting and well explained, the reviewer would highly appreciate if the authors can provide an experiment or application, where the newly gained insights into the training process has an influence on how the RNN is trained in the end.

**Questions:**

As already stated in the weaknesses: is there a scenario, where the gained insights can positively change/influence the training procedure of the PLRNN?

Since deep feedforward neural networks are also dynamical systems (simply without forcing), can the provided theory also be applied to (residual) feedforward neural networks with ReLU activations?

**Limitations:**

As already explained above, the only limitation/weakness is the limited significance of this work. If the authors can convince me of its significance, I'm happy to further increase my score.

---

> ### Author Rebuttal · Authors · 2023-08-09
>
> We thank the referee for the very constructive and fair feedback, and the supportive response!
>
> **Weaknesses**
>
> *”Limited significance for design of training algorithms”*: Although the exploding or vanishing gradient problem has been studied for a while now, viewing the training process from the perspective of bifurcation theory adds to it and can, in our minds, indeed have significant implications for designing novel training algorithms as it highlights the importance of topological features for guiding training. There are at least three lines along which improvements are conceivable, along two of them we can already provide some new results:
>
> 1) If the algorithm finds itself during training in a parameter regime which does not exhibit the right topological structure, it does not make sense to dwell within that regime or anywhere within the vicinity of the current parameter estimate. An implication of this observation is that, unlike standard SGD procedures, we should take large leaps in parameter space right away as soon as we find ourselves within a non-proper dynamical regime, possibly by building in a ‘look-ahead’ mechanism that probes for topological properties of regions not visited so far. While fully fleshing out these ideas is beyond a single paper, for this latter strategy we now provide a proof of concept in Fig. R4 of the uploaded PDF file, which shows that a topological look-ahead mechanism can lead to much faster convergence to the desired dynamical regime.
>
> 2) Another potentially important insight is that if we knew the model’s full bifurcation structure in parameter space ahead of time, we could simply pick, right away, a parameter set which corresponds to the right dynamics describing patterns in the data best. While of course it will in general not be feasible to chart the whole bifurcation structure before training (this is in a sense the whole point of a training algorithm), we may be able to design smart initialization procedures based on this insight, e.g. probing topological regimes at randomly selected points in parameter space before even starting training, or testing for bifurcations along maximum eigendirections of the system. In fact, the very recently proposed technique of generalized teacher forcing (GTF) often appears to quickly pull the RNN into the right topological regime:
>
> 3) We discovered that a recently proposed technique for training RNNs on chaotic dynamical systems, dubbed ‘generalized teacher forcing (GTF)’ by the authors (Hess et al. 2023 ICML), can be adjusted such as to avoid degenerate transcritical (and possibly other) bifurcations (DTB) during training altogether. Specifically, we can strictly prove (Theorem stated below, proof will be included in revised paper) that a DTB will never occur in training if we adjust the GTF parameter $0< \alpha <1$. This is of course just one example, but it demonstrates that we can actually harvest or amend training procedures such as to avoid specific types of bifurcation. More generally, we observed that GTF tends to circumvent bifurcations, as illustrated in new Fig. R6 in the PDF, leading to much faster convergence to suitable parameter regimes. The way this works is that GTF, by trading off RNN latent states with actual observations according to a specific annealing schedule during training, tends to pull the RNN states directly into the right dynamical regime. Our techniques can explain how this works, and possibly improve on it.
>
> Equally important, however, in our minds, although we showcase SCYFI on bifurcations during training problems, its applications are much wider and general: In most scientific and medical applications of RNN algorithms we are not only interested in obtaining good predictions, but in understanding exactly how and why the RNN model works. By finding all fixed points and cycles (up to some order), as well as bifurcation manifolds, SCYFI adds a level of *interpretability* and *explainability* to trained RNNs that is difficult to obtain otherwise (as, e.g., reflected in Fig. 4). It allows for in depth analysis of trained models and for dissecting their inner workings that helps to understand mechanisms and limitations in trained models. We believe this is a very important application, in particular in medical and scientific contexts. We will clarify and work this out much more in our revision.
>
> *Proposition 1*. Consider a PLRNN of the form (2) with parameters $\boldsymbol{\theta}=\\{A, W, h\\}$. Assume that it has a stable fixed point or $k$-cycle $\Gamma_k$ $(k \geq 1)$ that undergoes a degenerate transcritical bifurcation (DTB) for some parameter value $\theta=\theta_0 \in \boldsymbol{\theta}$. \
> (i) If $||A + W || \leq   1$, then for any GTF parameter $0 < \alpha < 1$, GTF controls the system, avoiding DTB and so gradient divergence at $\theta_0$. \
>  (ii) If $||A + W||  =   r > 1$, then for any $ 1 - \frac{1}{r} < \alpha < 1 $, GTF prevents DTB and so gradient divergence at $\theta_0$.
>
> **Questions**
>
> *Q1*: Yes, see above.
>
> *Q2*: This is an interesting idea, and the answer should be yes: As parameters of a DNN smoothly vary, there may be bifurcations in the forward dynamics across multiple layers that are significant for the training process. We will comment on this in our Discussion.
>
> **Limitations**
>
> See above: We now provide several lines, and provide preliminary results in the included PDF, along which bifurcation theory can help to improve training algorithms.

---

> > ### Comment · Reviewer_T1NJ · 2023-08-17
> > **Thanks for the detailed rebuttal**
> >
> > I thank the authors for their detailed rebuttal. In particular, I appreciate their thorough response to my point regarding the limited significance. The new result regarding the look-ahead mechanism is very interesting and changes my perspective on the significance. I thus increase my rating.

---

> > > ### Author Response · Authors · 2023-08-17
> > > **Thanks!**
> > >
> > > We are happy to hear we could address your concern about significance, and thank you once again for the very supportive and helpful feedback!

---

### Official Review · Reviewer_1SGy · 2023-07-10

**Soundness:** 3 good
**Presentation:** 3 good
**Contribution:** 3 good
**Rating:** 7
**Confidence:** 4

**Summary:**

The main focus of the paper is piecewise-linear RNNs, that is ReLU-based RNNs called PLRNNs, and how certain bifurcations that take place during their training phase will have an adverse effect on their gradients and hence will cause sudden jumps in the loss functions. This helps shed light onto the difficulty of training RNNs and yields new perspectives for the problem of exploding/vanishing gradients.

Moreover, the paper offers a novel way for locating fixed points and k-cycles in PLRNNs which can be helpful in visualizing and exploring the properties of these networks.

The main technical contributions are Theorem 1 and Theorem 2 which provide sufficient conditions under which the loss gradient will go to infinity or will vanish. These sufficient conditions are directly linked to the bifurcation properties of the PLRNN.

**Strengths:**

+the paper presents much-needed theory on a topic that is interesting both for theoreticians and practitioners
+conceptually, the work is very interesting and helps clarify an important problem in the literature of RNNs
+the distinction between the two cases of exploding and vanishing gradient through the lens of bifurcation diagrams is interesting
+their algorithm for discovering fixed points and k-cycles, though heuristic in nature, seems to be performing well.

**Weaknesses:**

-the algorithm for finding fixed points doesn't come with guarantees but that's perhaps not the most important weakness.

The main weakness is in presentation and in the literature review that make the paper hard to read in certain parts. The topic on training RNNs and the difficulty that arises has been studied a lot, but more recently there have been efforts to explain certain phenomena through the lens of dynamical systems.

In terms of presentation, I believe the paper would benefit by instantiating the theorems 1 and 2, on specific simple examples. You do sth related in lines 142-184 but I would create a small subsection "Simple Examples" to help illustrate the points. I believe Figure 1 is hard to understand and would perhaps break it into 2 different figures.

In terms of comparison to related works, recently we have seen several works that analyze behavior of the networks at the edge of stability or using the notion of k-cycles and chaos to illustrate why RNNs are hard to train. See papers below:

[1] On Scrambling Phenomena for Randomly Initialized Recurrent Networks by Chatziafratis, Panageas, Sanford, Stavroulakis

[2] "Better Depth-Width Trade-offs for Neural Networks through the lens of Dynamical Systems" by Chatziafratis, Nagarajan, Panageas,

[3] Understanding Edge-of-Stability Training Dynamics with a Minimalist Example by Xingyu Zhu, Zixuan Wang, Xiang Wang, Mo Zhou, Rong Ge

[4] Understanding Gradient Descent on Edge of Stability in Deep Learning by Sanjeev Arora, Zhiyuan Li, Abhishek Panigrahi

Your paper studies low-dimensional settings, especially in Fig.4 you provided a 2d toy example that helps illustrating the tight association between the loss landscape and bifurcation curves, whereas [1] and [2] studies the 1-dimensional setting giving characterizations from Li-Yorke chaotic systems, a standard notion from the dynamical systems literature. Papers [3] and [4] study the dynamics of GD and establish new connections with the notion of "sharpness"; while the emphasis is more on deep learning, the papers are very related as they analyze the dynamics through eigenvalues analyses.

**Questions:**

Q1: Is there a simple 1-d example that is even simpler that the 2d example you provide? And that helps illustrate your two main theorems?
Q2: In more applied settings, RNNs often operate in large dimensions. Could you comment on how your results can be helpful in these setting as well?

**Limitations:**

No significant limitations.

---

> ### Author Rebuttal · Authors · 2023-08-09
>
> We thank the referee for the very supportive and constructive feedback!
>
> **Weaknesses**
>
> *”Theoretical guarantees for algorithm”*: Obtaining rigorous theoretical guarantees is challenging in the current case (and sometimes, historically, theoretical understanding of convergence and other properties of an algorithm lagged behind by many years, e.g. Simplex or various training tweaks in DL).
> However, in the meantime we made some progress here: We can now strictly prove that the algorithm will converge with probability almost 1 under some general conditions on the PLRNN parameters, and will add this to our revision. Furthermore, we can add some numerical results on this topic: As shown in Fig. R1 in the included PDF file, when we initiate SCYFI in different randomly selected linear subregions, it converges to the subregions including the dynamical objects of interest *exponentially fast*. This numerical observation further helps to explain its efficiency.
>
> *”Presentation”*: Yes, this is a fair point. We have now substantially reworked Fig. 1 to exactly demonstrate the kind of situations our theorems are dealing with, see Fig. R2 in included PDF file. We will furthermore rewrite sect. 3.1, moving some of the current material (like that on multistability) to the Appx. and putting a stronger focus on the bifurcations and situations our paper specifically deals with.
>
> *”Literature covered”*: We thank the Referee for providing additional relevant literature (some of which we are indeed familiar with) which we will incorporate into our Related Work section, apologies for this omission!
>
> **Questions**
>
> *Q1*: In principle yes, one can construct 1d cases which essentially boil down to the skew tent map. See Fig. R3 in the included PDF where we provide one such example for different bifurcations in this 1d case. However, strictly, the smallest type of PLRNN which really makes sense is 2d (e.g., because of the off-diagonal nature of the connectivity matrix $\textbf{W}$ in the original formulation). Also, a 2d representation allows us to bring across certain concepts, like multi-stability in state space or the association with loss landscapes in Fig. 3, a bit better we think. While we are happy to include 1d examples as in the attached PDF, our preference therefore would be to stick mainly with 2d examples.
>
> *Q2*: Our theorems in fact apply to systems of any dimension. Low-dimensional examples in Figs. 1 \& 3 were chosen mainly for purposes of illustration and visualization, but the insights gained from them remain valid in higher-dimensional settings as well. We will make this more explicit in our revision. Also note that Fig. 4 indeed provides one example of a much higher-dimensional setting (likewise Fig. S6 in the Appx.). Its state space and bifurcation graph were only projected into 2d or 1d, respectively, for purposes of visualization.

---

> > ### Comment · Reviewer_1SGy · 2023-08-19
> > **Thank you, score updated**
> >
> > Given that the authors addressed my concerns, and given that they will include the above discussions and examples in the final version, I decided to update my score from 6 to 7.

---

> > > ### Author Response · Authors · 2023-08-19
> > > **Thanks!**
> > >
> > > Thank you very much again for your helpful feedback and suggestions for improvement!
> > > Yes, as stated in our rebuttal, we will make sure to include the new material, clarifications, and suggested literature in our revision.

---

### Official Review · Reviewer_qxJb · 2023-08-02

**Soundness:** 3 good
**Presentation:** 4 excellent
**Contribution:** 3 good
**Rating:** 6
**Confidence:** 3

**Summary:**

In this paper, the authors provide a thorough analysis of the bifurcation structure of RNNs and its relationship with discontinuities of the loss function during the training process. The authors also present a novel algorithm for detecting fixed points and k-cycles in ReLU-based RNNs, providing a new tool for understanding the computational and dynamical properties of these networks.

**Strengths:**

This paper nicely connects dynamical systems theory (DST) and machine learning by studying the problem of RNN training. This subject is known to be a challenging one, and the authors apply DST techniques to prove a rigorous result about the connection between bifurcations and jumps in the loss function during training, a question I would see as quite natural but rather unexplored in this context.

The paper is nicely written and accessible to the broad audience. Results are clearly stated and novel.

**Weaknesses:**

From the theoretical standpoint, the authors limit their discussions to two types of bifurcations. A more general and comprehensive discussion would be a nice addition to the paper.

The proposed algorithm appears to be purely heuristic and clashes a bit with the more rigorous first part of the paper.

The numerical experiments are a bit limited. The 2d example only qualitatively fits with the stated results: the bifurcation points do not seem to be exactly the points where the phase transition occurs. Is this due to numerical approximation?

**Questions:**

Can the authors say something about other kinds of bifurcations and their connection to loss function behavior during training?

Other questions are listed in the "weaknesses" sections

**Limitations:**

The authors should address the limitations more openly, both in terms of the presented algorithm and in terms of the scope of their theoretical results.

---

> ### Author Rebuttal · Authors · 2023-08-09
>
> We thank the referee for the very supportive and overall positive assessment of our work!
>
> **Weaknesses**
>
> *“Limited discussion of bifurcation types”*: Yes, true, originally there were two types of bifurcation for which we could offer rigorous proofs, so we focused on them in our presentation. To obtain general results for all types of bifurcations is virtually impossible within the scope of such a paper.
> However, we now managed to derive another proof for a degenerate flip bifurcation (DFB), which we will add to the revised manuscript. In the case of a DFB, gradients also vanish at the bifurcation point (Theorem stated below, proof will be added to ms.). We can also provide another experimental example of this during training, see Fig. R5 in the uploaded PDF, as well as new results on center bifurcations (CB). Based on these new results and observations, we are happy to expand our discussion of other bifurcations in sects. 3.1 and 6 and provide a more general perspective on this topic.
>
> *Proposition*. Assume that the PLRNN (2) has a stable fixed point $\Gamma_1$ with $B_{\Gamma_1}$ as its basin of attraction. If $\Gamma_1$ undergoes a degenerate flip bifurcation (DFB) for some parameter value $\theta=\theta_0 \in \boldsymbol{\theta}$, then  $\lim_{\theta\to\theta_0} || \frac{\partial L_t}{\partial \theta} || \to 0$ for every $Z_1 \in B_{\Gamma_1}.$
>
> *”Algorithm heuristic”*: Yes, this is true, at the time of submission the algorithmic approach rested on a heuristic, yet a powerful one. We also would like to note that historically there were many powerful algorithms invented (like Simplex or various training tweaks in deep learning) where a more theoretical understanding sometimes lagged behind by (many) years.
> However, in the meantime we made some progress here: We can now strictly prove that the algorithm will converge with probability almost 1 under some general conditions on the PLRNN parameters, and will add this to our revision. We also can empirically demonstrate that the algorithm converges exponentially fast to those particular subregions in state space which harbor the dynamical objects of interest (fixed points or cycles), see Fig. R1 in the provided PDF document.
>
> *”Only qualitative fits/ examples limited”*: We will add further examples on bifurcations (DFB and CB) to the Appx., see above and Fig. R5 in the provided PDF.
> And yes, in the 2d example in Fig. 3 (which we assume the referee is referring to) the apparent mismatch is a purely numerical (and visualization) issue, partly due to the fact that we cannot apply SCYFI for each and every BPTT updating step (note that here this was done mainly for purposes of illustration, while usually one would use SCYFI only for once trained models such that this issue would not occur).
>
> **Question**
>
> Yes, see above, we now have a proof also for Degenerate Flip Bifurcations (DFBs), and will extend our discussion and examples on other types of bifurcation.
>
> **Limitations**
>
> Some of the limitations we think we had discussed, but we agree there are other ones that were perhaps less explicit. So we will add a more explicit pg. on current limitations of the algorithm and approach to the ‘Conclusions’ sect. (incl. the current gaps in theoretical understanding, and the limitation to certain classes of bifurcations).

---

> > ### Comment · Reviewer_qxJb · 2023-08-16
> > **Response to rebuttal**
> >
> > I thank the authors for their response and congratulate them on the extended results. I would be happy to raise my score if the proof of the proposition, the discussion on the limitations and an updated, more precise (if possible) figure 3 could be somehow displayed (or at least a sketch of the first two could be given as a response to this comment).

---

> > > ### Author Response · Authors · 2023-08-17
> > > **Additional material**
> > >
> > > We thank the referee for the encouraging response!
> > >
> > > Please find a proof of the proposition on the gradient behavior across DFBs further below.
> > >
> > > We will add the following pg. on limitations to the Discussion:
> > >
> > > “While we discovered numerically that SCYFI for empirically relevant scenarios converges surprisingly fast, deriving strict theoretical guarantees is hard, and so far we could establish stronger theoretical results on its convergence properties only under specific assumptions on the RNN parameters. Further theoretical work is therefore necessary to precisely understand why the algorithm works so effectively. We also would like to note that there are numerous other types of bifurcations (e.g., center bifurcations, Hopf bifurcations etc.) that are likely to impact gradients during training, only for a subset of which we could provide formal proofs here. As we have demonstrated with two examples, understanding the topological and bifurcation landscape of RNNs could help improve training algorithms and provide insights into their working. Hence, a more general understanding of how various types of bifurcation affect the training process in a diverse range of RNN architectures is a promising future avenue not only for our theoretical understandings of RNNs, but also for guiding future algorithm design.“
> > >
> > > Regarding Fig. 3, please note that it is not possible at this stage to upload a further pdf or updated figure. We would like to remark, however, that even if we had SCYFI evaluations after each single BPTT updating step, there would likely still be some mismatch since an upcoming bifurcation will already start impacting the gradients when the RNN parameters get close to it.
> > >
> > > \
> > > **Proposition**. Assume that the PLRNN (2) has a stable fixed point \\( \Gamma_1\\) with  \\(B_{\Gamma_1} \\) as its basin of attraction. If \\(\Gamma_1 \\\) undergoes a degenerate flip bifurcation (DFB) for some parameter value \\(\theta=\theta_0 \in \boldsymbol{\theta}\\), then  \\(\lim_{\theta\to\theta_0} || \frac{\partial L_t}{\partial \theta} || \to 0 \\) for every \\(Z_1 \in B_{\Gamma_1}\\).
> > >
> > > **Proof**. For \\(M=2 \\), let \\(h_1 \neq 0\\). Then, the DFB curves of the fixed point \\( \Gamma_1\\) coincide with the BCB curves of the 2-cycle \\(\mathcal{O}\_{\mathcal{R L}}\\) of the form
> > > \
> > > \\[ \mathcal{F}\_{1} = \xi^1\_{\mathcal{R L}} =
> > > \\{ (h_1, h_2, a_{11}, a_{22})  |  1 + a_{11}+a_{22}+ a_{11} a_{22}  = 0 \\}, \\] \
> > > or   \
> > > \\[  \mathcal{F}\_{2} = \xi^2_{\mathcal{R L}}= \\{ (h_1, h_2, a_{11}, w_{11}, w_{21},  a_{22}) \big|   1+ a_{11} + w_{11} +a_{22}+ ( a_{11}+w_{11} )a_{22} = 0 \\}.
> > > \\]
> > >
> > > For \\(M>2 \\), assume that \\( \Gamma_1 = \\{ \mathbf{z}\_{1}^* \\} \\) is a fixed point of the system, i.e.
> > > \
> > > \\(
> > > \mathbf{z}\_{1}^* \\, = \\, (\mathbf{I} - \mathbf{W}\_{\Omega(t\_{1}^*)}  )^{-1} \\, \mathbf{h} = \frac{adj(\mathbf{I} - \mathbf{W}\_{\Omega(t\_{1}^*)})}{P\_{\mathbf{I} - \mathbf{W}\_{\Omega(t\_{1}^*)}}(1)} \mathbf{h},
> > > \\)
> > > \
> > > where \\( P\_{\mathbf{I} - \mathbf{W}\_{\Omega(t\_{1}^*)}}(1) \\) is the characteristic polynomial of \\( \mathbf{I} - \mathbf{W}\_{\Omega(t\_{1}^*)}\\) at 1. Let us denote the first row of  the adjoint matrix of  \\(\mathbf{I} - \mathbf{W}\_{\Omega(t\_{1}^*)} \\) by   \\(adj(\mathbf{I} - \mathbf{W}\_{\Omega(t\_{1}^*)})_1\\). If  \\(adj(\mathbf{I} - \mathbf{W}\_{\Omega(t\_{1}^*)})_1 \\, \mathbf{h}  \neq 0\\), then we can analogously demonstrate that the DFB curves of the fixed point align with the BCB curves of the 2-cycles. This implies that, in accordance with Theorem 2, DFBs of fixed points will also lead to vanishing gradients in the loss function.

---

> > > > ### Comment · Reviewer_qxJb · 2023-08-18
> > > >
> > > > Thank you for your response. I have updated my score.

---

### Author Rebuttal · Authors · 2023-08-09

We thank all four referees for their generally positive evaluation and supportive response! We appreciate the detailed reading of our manuscript, and the many helpful comments pointing out where our paper needs clarification or extension of results. Besides the detailed reply to all issues raised in our responses below, we have included a PDF with further material and figures.

In brief, here we

- provide a new theorem and numerical results on SCYFI’s convergence properties
- extend the bifurcation results by providing a new theorem and examples for two other important bifurcations (degenerate flip bifurcation (DFB) and center bifurcation (CB))
- updated the pseudo-code for SCYFI for clarification (see below)
- completely reworked Fig. 1 for clarification and to bring it into closer relation with the rest of the text
- provide an example of bifurcations in a related 1-d system
- provide a new theorem and numerical illustrations to demonstrate the implications of bifurcation theory for designing novel training algorithms

We hope these new results together with the point-by-point replies below sufficiently cover all the remaining concerns.

Here is a clarified version of the SCYFI pseudo-code:

**************************
## Algorithm 2 SCYFI
**************************
The algorithm is iteratively run with \\( k = 1 \ldots K_{max} \\), where \\( K_{max} \\) is the max. order of cycles tested \
**Input:** PLRNN parameters \\( A \\), \\( W \\), \\( h \\);
\
&emsp;&emsp; \\( \mathcal{L} = \\{ \mathcal{L}\_n \\}\_{n=1}\^{k-1} \\): collection of all sets \\( \mathcal{L}\_n = \\{ \\{ z\^{(m) }\_{l} \\}\_{l=1}\^{n} \\}\^{M_n}\_{m=1} \\) of all lower order \\( n \\)-cycles discovered so far, where \\( M_n \\) is the number of found cycles \\( \\{ z\^{(m)}\_{l} \\}\_{l=1}\^{n} \\) of order \\( n \\), with corresponding ReLU derivative matrices \\( \\{ D\^{(m)}\_{l} \\}\_{l=1}\^{n} \\).

**Parameters:**
\
&emsp;&emsp;  \\( N_{out} \\): max. number of random initializations; \
&emsp;&emsp;  \\( N_{in} \\): max. number of iterations

**Output:** \\( \mathcal{L} \cup \mathcal{L}\_k \\); \\( \mathcal{L}_k \\): set of all discovered \\( k \\)-cycles

1: \\( \mathcal{L}\_k = \\{ \\} \\) \
2: \\( i \rightarrow 0 \\) \
3: **while** \\( i < N_{out} \ldots \\)  do \
4: &emsp; Select \\( k \\) subregions \\( D_{init} = \\{ D_1, D_2, \ldots, D_{k} \\} \\) at random with replacement \
5:  &emsp;  \\( c \rightarrow 0 \\) \
6:  &emsp;  **while** \\( c < N_{in} \ldots \\)  do \
7: &emsp; &emsp; Solve eq. (7) for a cycle candidate \\( \\{ z^*_{l} \\}\_{l=1}^{k} \\) with \\( W_{\Omega(k-r)} \\) as defined in equation (2) based on \\( D_{init} \\) \
8: &emsp; &emsp; Determine \\( \\{ D^*_{l} \\}\_{l=1}^{k} \\) based on the signs of the corresponding components of \\( \\{ z^*_{l} \\}\_{l=1}\^{k} \\) \
9: &emsp; &emsp; **if**  \\( D_{init} = \\{ D^*_{l} \\}\_{l=1}^{k} \\) (*self-consistency*) &  \\( \forall 1 \leq s \leq k, \forall \\{ z^{(m)}\_{l} \\}\_{l=1}\^{k-s+1} \in \mathcal{L}\_{k-s+1}: \\{ z^{(m)}_{l} \\}\_{l=1}\^{k-s+1} \not\subseteq  \\{ z^*_l \\}\_{l=1}\^{k} \\) **then**

10: &emsp; &emsp; &emsp; \\( \mathcal{L}\_k \rightarrow \mathcal{L}\_k  \cup \\{ \\{ z^*_{l} \\}\_{l=1}\^{k} \\} \\)  \
11:  &emsp; &emsp; &emsp;  \\( i \rightarrow c \rightarrow 0 \\) \
12:   &emsp; &emsp;  **else** \
13:  &emsp; &emsp; &emsp; \\( D_{init} \rightarrow \\{ D^*_{l} \\}\_{l=1}\^{k} \\) \
14:   &emsp; &emsp;  **end if** \
15:  &emsp; &emsp;  \\( c \rightarrow c+1 \\) \
16:   &emsp; **end while** \
17: &emsp;  \\( i \rightarrow i+1 \\) \
18:  **end while**
**************************************

---

### Decision · Program_Chairs · 2023-09-21

**Decision:**

Accept (spotlight)

**Comment:**

All reviewers agree that the paper discusses an interesting and challenging problem and brings new insights and a significant contribution to the problem of learning RNNs based on dynamical systems theory. Also during the discussion phase, the authors were able to provide critical feedback to the comments given by the reviewers, which was helpful in addressing their technical concerns. As a whole, I recommend acceptance (splotlight) of the paper for this submission.